# Reducing Symmetry Increase in Equivariant Neural Networks

**Ning Lin, Jiacheng Cen, Anyi Li, Wenbing Huang**[*]**, Hao Sun**[*]
Gaoling School of Artificial Intelligence, Renmin University of China, Beijing, China
{ninglin00, jiacc.cn, li_anyi}@outlook.com,
{hwenbing, haosun}@ruc.edu.cn

## Abstract

Equivariant Neural Networks (ENNs) have empowered numerous applications in scientific fields. Despite their remarkable capacity for representing geometric structures, ENNs suffer from degraded expressivity when processing symmetric inputs: the output representations are invariant to transformations that extend beyond the input's symmetries. The mathematical essence of this phenomenon is that a symmetric input, after being processed by an equivariant map, experiences an increase in symmetry. While prior research has documented symmetry increase in specific cases, a rigorous understanding of its underlying causes and general reduction strategies remains lacking. In this paper, we provide a detailed and in-depth characterization of symmetry increase together with a principled framework for its reduction: (i) For any given feature space and input symmetry group, we prove that the increased symmetry admits an infimum determined by the structure of the feature space; (ii) Building on this foundation, we develop a computable algorithm to derive this infimum, and propose practical guidelines for feature design to prevent harmful symmetry increases. (iii) Under standard regularity assumptions, we demonstrate that for *most* equivariant maps, our guidelines effectively reduce symmetry increase. To complement our theoretical findings, we provide visualizations and experiments on both synthetic datasets and the real-world QM9 dataset. The results validate our theoretical predictions.

## 1 Introduction

Equivariant Neural Networks (ENNs) have become a cornerstone of modern machine learning, empowering numerous applications in scientific fields ranging from molecular dynamics to materials design (Bronstein et al., 2021; Huang & Cen, 2026). By building in physical symmetries, these models achieve remarkable data efficiency and generalization capabilities when representing complex geometric structures.

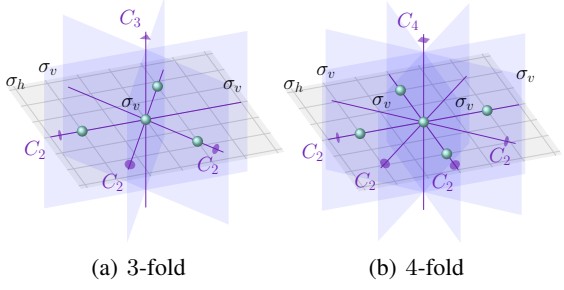

(a) 3-fold      (b) 4-fold

Figure 1: $k$-fold structures.

Despite their success, ENNs exhibit a critical vulnerability when processing symmetric inputs: their expressivity can degrade, leading to a loss of information. This phenomenon, which we term **symmetry increase**, occurs when the output representation becomes invariant to transformations that are not symmetries of the original input itself. A canonical example arises when processing $k$-fold symmetric structures. These objects, visualized in Fig. 1, possess a specific dihedral symmetry, yet an ENN will map their distinct rotated versions to an identical feature, erasing their orientation.

This type of degradation has been documented in previous work. Empirically, it has been observed that the degradation depends on the feature space, particularly for symmetric inputs (Joshi et al., 2023). Theoretically, research on ENNs has shown that for $k$-fold symmetries, selecting only low-degree

---

[*]Corresponding authors.

features can cause all rotated inputs to collapse into a single representation (Cen et al., 2024). This ENN-specific issue is a modern manifestation of a general phenomenon observed in other fields. It has been linked to physical principles such as Curie's Principle (Smidt et al., 2021) and described using the concept of orbit types (Kaba & Ravanbakhsh, 2023).

However, existing analyses provide an incomplete picture and lack a predictive framework. In our analysis, this degradation of $k$-fold caused by symmetry increase can be categorized into three distinct types: *full degeneration*, *axial degeneration*, and *half degeneration* (see Fig. 2)[1]. The work of Joshi et al. (2023), while empirically important, does not theoretically explore the cause of the degradation. The *collapse-to-zero* theory proposed by Cen et al. (2024) addresses only full degeneration, which is the most extreme case identified by our analysis. The broader principles discussed by Smidt et al. (2021) and Kaba & Ravanbakhsh (2023) lack a rigorous mathematical description, and the solutions proposed, such as in Kaba & Ravanbakhsh (2023), often involve relaxing the equivariance constraint itself, rather than providing a solution within the equivariant framework.

In this paper, we fill this gap by providing a comprehensive mathematical characterization of symmetry increase. Our main contributions are briefly listed as follows:

- In § 3, we prove for any given feature space and input symmetry group, that the increased symmetry is bounded from below by a unique symmetry infimum, which is determined entirely by the algebraic structure of the feature space.
- In § 4, we develop a computable algorithm to derive this infimum by analyzing the orbit types. This provides practical guidelines for predicting and controlling potential symmetry increases, thereby preventing harmful symmetry increases in feature design.
- In § 5, we demonstrate that under regularity conditions, such as the manifold hypothesis for data, our method can fully reduce symmetry increase. Specifically, for *most* equivariant maps or for ENNs with sufficient approximation capabilities, the output symmetry will be precisely this predictable infimum, preventing orientational information loss.
- In § 6, we complement our theoretical findings with empirical evidence. We provide visualizations to illustrate the proposed concepts and present experimental results on both synthetic datasets and the real-world QM9 dataset, which validate our theoretical predictions and demonstrate the practical effectiveness of our framework.

## 2 PRELIMINARIES

**Group action and representation.** Consider the action of a group $G$ on a set $X$, denoted by $\rho_X$. This action is a map that assigns to each element $g \in G$ a transformation $\rho_X(g) : X \to X$, such that $\rho_X(g_1 g_2) = \rho_X(g_1)\rho_X(g_2)$. We call such a set $X$ a $G$-set. In particular, if $X$ is a vector space and $\rho_X(g)$ is a linear transformation for all $g \in G$, we call $X$ a $G$-representation[2].

**Equivariant map.** For maps between two $G$-sets $X$ and $Y$, an equivariant map is one that respects the group action, meaning that the output transforms accordingly when the input is transformed. Formally, a map $f : X \to Y$ is equivariant if for all $g \in G$ and $x \in X$:

$$f(\rho_X(g)(x)) = \rho_Y(g)(f(x)). \tag{1}$$

**Example 2.1** (Equivariant Encoding of Point Clouds). *The symmetry group is $G = H \times S_n$, where $H$ is typically the special orthogonal group $SO(3)$ or the orthogonal group $O(3)$, and $S_n$ is the permutation group. We consider features that are invariant to both permutation and translation. To achieve translation invariance, the input representation $X$ is the space of centered point clouds, where $H$ acts on the coordinate of each point and $S_n$ permutes the points. To achieve permutation invariance, the final output are designed to be invariant with respect to $S_n$. This means the feature representation of interest is a direct sum of specified irreducible representations of $H$. The task of equivariant encoding is then to learn an equivariant map $f : X \to Y$.*

**Data symmetry.** For a $G$-set $X$, the action partitions the space into orbits, $G(x) := \{g(x) \mid g \in G\}$. The subgroup that fixes a point is its **isotropy subgroup**, $G_x := \{g \in G \mid g(x) = x\}$. The conjugacy

---

[1]Although we choose $k$-fold as an illustrative example, our theories are applicable to general cases.
[2]Unless otherwise specified, all groups are assumed to be compact Lie groups, and all vector spaces are finite-dimensional real vector spaces.

class $(G_x)$ of this subgroup is the **orbit type** of $x$. The set of all points with orbit type $(H)$ is $X_{(H)}$ and the set of points fixed by all elements of a subgroup $H$ is the fixed-point set $X^H$.

These concepts distinguish between the global symmetry of the space and the intrinsic symmetry of an object. The group $G$ represents the global symmetry, transforming the object between different reference frames. An orbit $G(x)$ thus represents a single physical object in all of its possible orientations. This object possesses its own intrinsic symmetry, mathematically defined by the isotropy subgroup $G_x$ for any point $x$ on its orbit. While the specific subgroup is frame-dependent, its structure up to conjugation is constant. The orbit type $(G_x)$ therefore serves as a reference-frame independent identifier that precisely describes the physical object's intrinsic symmetry. The set of all possible orbit types, $\mathcal{O}_G(X)$, thus catalogs all distinct symmetries that objects in the space can possess.

**Example 2.2.** *Now apply the setup from Ex. 2.1. Consider the action of $G = O(3) \times S_{k+1}$ on centered point cloud space $X$ for $k > 2$. Let $x \in X$ be the set of vertices of a $k$-fold in the $xOy$-plane:*

$$x = (x_0, x_1, \ldots, x_k), \quad \text{where } x_i = (\cos(2i\pi/k), \sin(2i\pi/k), 0) \text{ for } i > 0. \quad (2)$$

*with $x_0$ at the origin. The generators of $G_x$ include:(1) A rotation about the $z$-axis combined with a cyclic permutation of $x_1, \ldots, x_k$. (2) A reflection across the $xOz$-plane combined with a product of transpositions. (3) A reflection across the $xOy$-plane combined with the identity.*

*Considering the projection map $\pi_X((g, \sigma)) = g$, where $(g, \sigma)$ is a pair consisting of a geometric transformation $g \in O(3)$ and a permutation $\sigma \in S_k$, we find that geometric symmetry $\pi_X(G_x)$ is the dihedral group with horizontal reflection of order $4k$, denoted by the Schoenflies symbol $D_{kh}$.*

The symmetry of data can be altered by an equivariant map. The following theorem shows that an equivariant map does not decrease symmetry.

**Theorem 2.3** (Curie's principle, Kaba & Ravanbakhsh (2023), Thm. 1)**.** *Let $f : X \to Y$ be a $G$-equivariant map. For $x \in X$, the isotropy subgroup of $x$ is contained in that of its image $f(x)$, i.e.,*

$$G_x \subseteq G_{f(x)}. \quad (3)$$

Such an increase in symmetry becomes unavoidable if the feature space $Y$ cannot support the input's symmetry. If the orbit type $(G_x)$ is not present in $Y$ (i.e., $(G_x) \notin \mathcal{O}_G(Y)$), then the equality $G_x = G_{f(x)}$ is impossible (otherwise, $G_{f(x)}$ becomes an isotropy subgroup, implying $(G_x) \in \mathcal{O}_G(Y)$, which leads to a contradiction), and the symmetry must therefore strictly increase. This increase leads to a degeneration, as any transformation $g$ in the larger group $G_{f(x)}$ that is not in $G_x$ will map the distinct inputs $x$ and $g(x)$ to the same output, since $f(g(x)) = g(f(x)) = f(x)$. Fig. 2 illustrates three possible types of such degenerations for the $k$-fold structure from Ex. 2.2.

## 3 THE INFIMUM OF SYMMETRY

This section establishes that symmetry increase is governed by an infimum determined by the feature space. We prove the existence of this infimum and show that its coincidence with the input symmetry is a necessary condition for an equivariant map to preserve symmetry.

Figure 2: Three types of degeneration of $k$-fold.

### 3.1 THE INFIMUM OF SYMMETRY

The symmetry of data can increase after being transformed by an equivariant map. This increase can be an intentional design choice, or it can arise from subtle properties of the feature space. For instance, in the task from Ex. 2.1, the requirement for permutation-invariant features means that the permutation group $S_n$ is naturally introduced into the isotropy subgroup of any output. This type of designed, unavoidable symmetry increase is formalized by the kernel of the group action on the feature space. We define the **kernel** of the action $\rho_X$ as the set of group elements that fix every point in $X$: $\ker\rho_X := \{g \in G \mid g(x) = x, \forall x \in X\}$

Distinguishing between intentional and unintended symmetry increase is crucial. We begin by assuming that the group action is faithful, i.e., it has a trivial kernel with $\ker\rho_Y = \{e\}$. The case of a nontrivial kernel will be discussed in the next subsection.

To characterize the behavior of symmetry increase within a representation $X$, we first establish a partial order to compare different symmetries. An orbit type $(H_1)$ is considered greater than or equal to another, $(H_2)$, written as $(H_1) \geq (H_2)$, if $H_1$ contains a conjugate of $H_2$. This ordering reflects that a larger orbit type corresponds to a higher symmetry.

With this framework, we are interested in the lower bound of orbit types that can be reached from a point $x$ with a specific isotropy group $H = G_x$. The analysis can be framed around any closed subgroup $H$, as the set of all possible isotropy subgroups is precisely the set of all closed subgroups of $G$ (see, e.g., Field (2007); Mostow (1957)). The fixed-point space $X^H$ contains points of all higher orbit types. This leads to the following powerful theorem, which guarantees that the lower bound on symmetry increase is unique, corresponding to an isotropy subgroup that is unique up to conjugation.

**Theorem 3.1** (Uniqueness of Minimal Type)**.** *Let $X$ be a representation of a compact Lie group $G$. For any closed subgroup $H \subseteq G$, a unique minimal orbit type exists among the points in the fixed-point subspace $X^H$. In particular, if $(H) \in \mathcal{O}_G(X^H)$, then $(H)$ is the minimal orbit type within that subspace.*

The uniqueness guaranteed by this theorem allows us to define the **symmetry infimum**, denoted by $I_G(X, H)$, as this unique minimal orbit type. In the context of symmetry increase, we are concerned with the relationship between $I(Y, G_x)$ and $(G_{f(x)})$. An unexpected symmetry increase occurs if $(G_{f(x)}) > I_G(Y, G_x)$. The desired behavior for an equivariant map is captured by the following definition. For a map between $G$-sets $X$ and $Y$, we define an **isovariant map** as one that strictly preserves symmetry for all $x \in X$:

$$G_x = G_{f(x)}. \tag{4}$$

Using the concept of the symmetry infimum, we can provide a necessary condition for the existence of isovariant maps. In § 5.1, we will see that this condition is in fact not sufficient for equivariant maps between representations, even when we assume a trivial kernel $\ker\rho_Y = \{e\}$.

**Theorem 3.2.** *A necessary condition for the existence of an isovariant map between $G$-sets $X$ and $Y$ is that $\mathcal{O}_G(X) \subseteq \mathcal{O}_G(Y)$. When $X$ and $Y$ are representations of a compact Lie group $G$, this is equivalent to the condition that $I_G(Y, H) = (H)$ for all $(H) \in \mathcal{O}_G(X)$.*

### 3.2 Equivariance with Non-trivial Kernels

When the feature space $Y$ is restricted to have a non-trivial kernel, i.e., $\ker \rho_Y \neq \{e\}$, the definition of an isovariant map becomes too restrictive. Since every isotropy group $G_y$ in $Y$ contains the kernel, any subgroup $H$ not containing $\ker \rho_Y$ cannot occur as an isotropy subgroup. Consequently, for any map $f : X \to Y$, an input isotropy subgroup $G_x$ that does not contain the kernel must increase.

To formalize this unavoidable symmetry increase, we introduce an operator $p_Y$. In the discussion following Ex. 2.2, the presence of the $S_{k+1}$ kernel forces the input isotropy subgroup $G_x$ to become at least $D_{kh} \times S_{k+1}$ in the feature space. We generalize this observation. The operator $p_Y(H)$ is defined as the smallest subgroup containing $H$ that is compatible with the action on $Y$, given by the projection $p_Y = \pi_Y^{-1} \circ \pi_Y$, where $\pi_Y : G \to G/\ker\rho_Y$ is the natural projection. This operator is idempotent ($p_Y^2 = p_Y$) and maps any isotropy subgroup in $Y$ to itself. These properties reveal why increasing is unavoidable. For any equivariant map $f$, the relation $G_x \subseteq G_{f(x)}$ must hold. Applying the $p_Y$ operator to this inclusion gives:

$$G_x \subseteq p_Y(G_x) \subseteq p_Y(G_{f(x)}) = G_{f(x)}. \tag{5}$$

This unavoidable increasing from $G_x$ to $p_Y(G_x)$ means our goal is not to preserve $G_x$ itself, but to ensure no *additional* symmetry is introduced beyond $p_Y(G_x)$. This leads to a more practical definition. We say a map $f : X \to Y$ is an **isovariant map relative to** $Y$ if for all $x \in X$ it satisfies

$$p_Y(G_x) = G_{f(x)} \quad \Longleftrightarrow \quad \rho_Y(G_x) = \rho_Y(G_{f(x)}). \tag{6}$$

When the kernel is trivial, this definition reduces to that of a standard isovariant map. With this refined goal, we can state a necessary condition for the existence of such maps.

**Theorem 3.3.** *A necessary condition for the existence of an isovariant map relative to $Y$ from a $G$-set $X$ to a $G$-set $Y$ is that $(p_Y(H)) \in \mathcal{O}_G(Y)$ for every $(H) \in \mathcal{O}_G(X)$. When $X$ and $Y$ are representations, this is equivalent to the condition that $I_G(Y, H) = (p_Y(H))$ for all $(H) \in \mathcal{O}_G(X)$.*

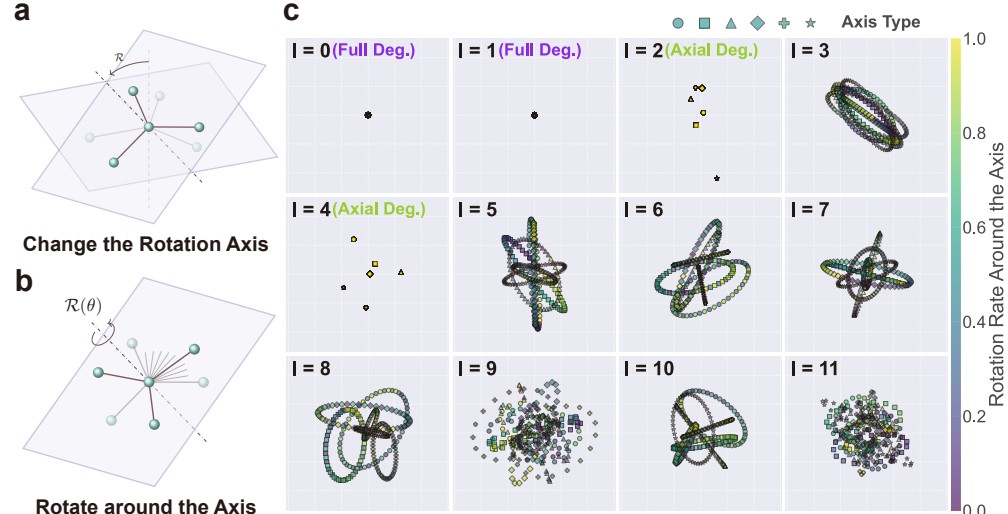

Figure 3: Visualization of representation spaces. (a) A $k$-fold structure is reoriented onto multiple planes. (b) Each is further rotated about the perpendicular axis. (c) All structures are embedded and projected into 2D. Marker shapes denote rotation axes, and colors denote rotation rates. Full degeneration appears at $l = 0, 1$, and axial degeneration at $l = 2, 4$.

## 4 COMPUTATION OF ORBIT TYPES

The computation of orbit types is a classical problem, with most established results focusing on irreducible representations, often in the context of bifurcation theory. In representation learning tasks, however, we utilize feature spaces containing high multiplicities of these representations, which requires us to supplement the existing computational frameworks.

### 4.1 ORBIT TYPES OF HIGH-MULTIPLICITY REPRESENTATIONS

For a compact Lie group $G$, any representation space $X$ can be uniquely decomposed into a direct sum of its irreducible components. For $G = SO(3)$, this decomposition is written as

$$X \cong \bigoplus_{l_0=0}^{\infty} V_{l=l_0}^{\oplus m(X, V_{l=l_0})}, \tag{7}$$

where $V_{l=l_0}$ is the irreducible representation corresponding to the space of spherical harmonics of degree $l_0$, and $m(X, V_{l=l_0})$ is its multiplicity. For $G = O(3)$, the decomposition is similar, but the irreducible representations must also be distinguished by parity, denoted $V_{l=l_0^+}$ and $V_{l=l_0^-}$.

We begin with the foundational criterion for identifying isotropy subgroups, first established as a necessary condition by Michel (1980).

**Theorem 4.1** (Michel's Criterion, Michel (1980), App. A). *Let $V$ be a representation of a group $G$. A necessary condition for a closed subgroup $H$ to be an isotropy subgroup in $V$ is that for any adjacent closed subgroup $H' \supsetneq H$, the dimension of the fixed-point subspace strictly decreases:*

$$\dim V^{H'} < \dim V^H. \tag{8}$$

While this condition is not sufficient for all representations, its sufficiency can be guaranteed under certain common conditions. We define a representation $V$ of a group $G$ as a high-multiplicity representation if for every non-zero isotypic component corresponding to an irreducible representation $V_i$, its multiplicity $m(V, V_i)$ is greater than $\dim G$.

**Proposition 4.2.** *For a high-multiplicity representation $V$, the necessary condition stated in Thm. 4.1 is also sufficient.*

This criterion is particularly powerful for two reasons. First, it offers a computationally convenient method in the form of a chain recursion, which only requires checking adjacent subgroups. The dimensions of the necessary fixed-point spaces can be calculated via the trace formula (Golubitsky

**Algorithm 1:** Orbit Type Test for High-Multiplicity Representations

**Data:** Symmetry group $G$;
      Closed subgroup $H \subset G$;
      High-Multiplicity Rep. $V$ of $G$
**Result:** is_in$((H), \mathcal{O}_G(V))$.
1 Let set $S = \{H_i\} \subset G$ to be all adjacent closed supergroups of $H$ in $G$;
2 $\mathcal{O} \leftarrow \varnothing$;

3 $d_H \leftarrow \dim V^H$;

4 **for** $H_i$ *in S* **do**
5     $d_{H_i} \leftarrow \dim V^{H_i}$;
6     **if** $d_H - d_{H_i} = 0$ **then**
7         **return** False;
8     **end**
9 **end**
10 **return** True;

**Algorithm 2:** Symmetry Infimum Calculation

**Data:** Symmetry group $G$;
      Closed subgroup $H \subset G$;
      Rep. $V$ of $G$.
**Result:** Symmetry inf. $I_G(V, H)$.
1 **if** is_in$((H), \mathcal{O}_G(V))$ **then**
2     **return** $(H)$;
3 **end**
4 Let set $S = \{H_i\}$ to be all closed supergroups of $H$ in $G$;
5 $\mathcal{O} \leftarrow \varnothing$;
6 **for** $H_i$ *in S* **do**
7     **if** is_in$((H_i), \mathcal{O}_G(V))$ **then**
8         Add $(H_i)$ to $\mathcal{O}$;
9     **end**
10 **end**
11 **return** $\min(\mathcal{O})$;

et al., 1988). Second, the sufficiency condition is frequently met in our applications, as it holds for all finite groups and for feature spaces with a high number of channels. Based on the result of Prop. 4.2, we design an orbit type test Algo. 1 and a symmetry infimum calculation Algo. 2. Using the two algorithms described above, we have characterized all instances of symmetry increase for the closed subgroups of $SO(3)$ or $O(3)$ in the representations $V_{l=l_0}^{\oplus r}$ and $V_{l=l_0^{\pm}}^{\oplus r}$ for $r > 3$, respectively, see § C.4.

We now illustrate our algorithms with a simple example.

**Example 4.3.** *We illustrate our algorithms by calculating the orbit type and symmetry infimum for the geometric symmetry $D_{kh}$ ($k > 2$) of the $k$-fold from Ex. 2.2, considered as a subgroup of $O(3)$. The calculation is performed in the high-multiplicity representation space $Y = V_{l=l_0}^{\oplus r}$ ($r > 3, l_0 > 0$). Here we provide only a sketch of the derivation, the full procedure is provided in § C.3.*

*First, we apply the orbit type test from Algo. 1. This involves comparing the dimension of the fixed-point space of $D_{kh}$ with that of its adjacent supergroups (e.g., $D_{pk,h}$ and, for $k = 4$, $O_h$). The analysis shows that $(D_{kh})$ is an orbit type if and only if $l_0 \geq k$ and $l_0, k$ have the same parity. Next, we apply the symmetry infimum calculation from Algo. 2. This requires identifying the minimal orbit type among all supergroups of $D_{kh}$, including non-adjacent ones like $D_{\infty h}$ and $O(3)$. The final results are summarized in Table 1.*

Table 1: The symmetry infimum $I_{O(3)}(V_{l=l_0}^{\oplus r}, D_{kh})$ for $k > 2, r > 3, l_0 > 0$.

| | $l_0 < k$ | | $k \leq l_0 < 2k$ | | $l_0 \geq 2k$ | |
| | $l_0$ is even | $l_0$ is odd | $l_0$ is even | $l_0$ is odd | $l_0$ is even | $l_0$ is odd |
|---|---|---|---|---|---|---|
| $k$ is even | $(D_{\infty h})$ | $(O(3))$ | $(D_{kh})$ | $(O(3))$ | $(D_{kh})$ | $(O(3))$ |
| $k$ is odd | $(D_{\infty h})$ | $(O(3))$ | $(D_{\infty h})$ | $(D_{kh})$ | $(D_{2kh})$ | $(D_{kh})$ |

The analysis in Ex. 4.3 predicts three types of degeneration for the $k$-fold inputs from Ex. 2.1:

- **Half Degeneration:** The symmetry infimum of $G_x$ is $(D_{2kh} \times S_{k+1})$. The feature cannot distinguish the $k$-fold from itself rotated by $\pi/k$ around the $z$-axis.

- **Axial Degeneration:** The symmetry infimum of $G_x$ is $(D_{\infty h} \times S_{k+1})$. The feature cannot distinguish the $k$-fold from itself rotated by *any* angle around the $z$-axis.

- **Full Degeneration:** The symmetry infimum of $G_x$ is $(O(3) \times S_{k+1})$. The feature cannot distinguish the $k$-fold from itself rotated by *any* angle around *any* axis.

We consider encoding $k$-fold point clouds using equivariant neural networks and visualize the resulting embeddings. The three degenerations are experimentally verified in our visualizations, with Fig. 3 showing full and axial degeneration, and Fig. 4 showing axial and half degeneration. Although derived assuming high multiplicity ($r > 3$) in the feature representation, these predictions are identical for the single representation case ($r = 1$), see § C.4.

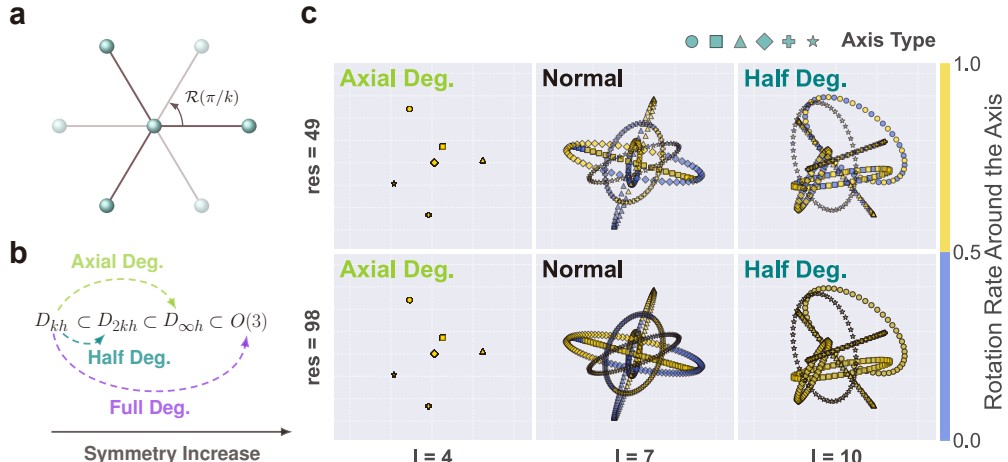

Figure 4: Visualization of representation spaces. (a) A $k$-fold ($k$ is odd) structure is rotated $\pi/k$ about the perpendicular axis. (b) The path of symmetry increase. (c) Half degeneration appears at $l = 10$. At $\mathrm{res} = 98$ and $\mathrm{res} = 49$, the overall shape is identical, but the yellow data points (*i.e.* the second half of the rotation) completely cover the blue data points.

## 4.2 GUIDELINES FOR MANAGING SYMMETRY INCREASE

It is a general property of representations that the orbit types of a direct sum are related to those of its components by $\mathcal{O}_G(V_1) \cup \mathcal{O}_G(V_2) \subseteq \mathcal{O}_G(V_1 \oplus V_2)$, and $I_G(V_1 \oplus V_2, H) \leq I_G(V_i, H)$ for $i = 1, 2$, with equality conditions discussed in § C.5. These properties provide a direct mechanism for controlling the symmetry increase of an equivariant feature, that is to choose components whose symmetry infimum (computed as described in § C.4 for $G = SO(3)$ or $O(3)$) align with the desired behavior for task-relevant symmetries. Regarding the selection of the feature space $Y$ in equivariant representation learning, this principle translates into two guidelines.

For **orientation-dependent tasks** (e.g., § 6.2), when considering the kernel of feature space, it is crucial to avoid non-trivial symmetry increase (i.e. ensuring the map is relative isovariant) since such increases can lead to the accidental loss of orientational information. Therefore, for a given input symmetry ($H$), one should select feature components that contain the orbit type ($p_Y(H)$).

For **general tasks** (e.g., § 6.3), certain forms of symmetry increase must be avoided, as the output symmetry reflects the dimensionality of the fixed-point subspace where the equivariant features lie (see *Remark* of Prop. C.2). In this context, one should generally avoid components where the symmetry infimum indicates a severe compression of the fixed-point subspace. Specifically, one must be cautious of components corresponding to non-trivial representations where the symmetry increases to the full group, as this causes the component to be annihilated and lose all discriminative power.

## 5 DENSITY OF (ALMOST) ISOVARIANT MAPS

We now connect the preceding theory to a practical machine learning context by introducing models for the data distribution and for the parameterized map. We show that the necessary conditions for isovariance established previously become sufficient under a relaxed definition of isovariance.

## 5.1 THE MANIFOLD HYPOTHESIS

Motivated by the manifold hypothesis and the broader considerations summarized in § A.3, we model the data distribution as being supported on a finite union of smooth, compact submanifolds $M = \bigcup_j M_j$ embedded in $X$. When a group $G$ acts on $X$, this action equips each $M_j$ with a natural $G$-manifold structure.

The central question is: when does an isovariant map $f : M \to Y$ exist? The existence is a non-trivial issue. The counterexample in Cex. D.3 demonstrates that the necessary condition of orbit type

inclusion is not sufficient. Specifically, it shows that an isovariant map can fail to exist precisely because the multiplicities of the irreducible representations in the feature space are insufficient.

The non-existence of perfectly isovariant maps motivates a more practical, relaxed definition: a map that is isovariant almost everywhere. To formalize this, we equip $M$ with the $d$-dimensional Hausdorff measure $\mu_M$, where $d = \max_j\{\dim M_j\}$. This allows us to identify subsets of *measure zero* as negligible. Note that $M_{(H)}$ is a finite union of submanifolds, then $f$ is **almost isovariant relative to** $Y$ if for every orbit type $(H)$ in the data support, the isovariance condition

$$\rho_Y(G_x) = \rho_Y(G_{f(x)}), \tag{9}$$

holds for all points $x \in M_{(H)}$ except for a subset of $\mu_{M_{(H)}}$-measure zero. This ensures that any undesired increase in symmetry occurs only on a negligible portion of the data.

## 5.2 GENERICITY OF (ALMOST) ISOVARIANT MAPS

For Ex. 2.1, we select TFN (Thomas et al., 2018), a classic ENN based on tensor products, as our parameterized model. We provide the complete formulation to § D.2. An important property of TFN parameterizations $\mathcal{F}_{\text{TFN}}$ is that they satisfy a universal approximation theorem (Dym & Maron, 2021). In topology, this is equivalent to $\mathcal{F}_{\text{TFN}}$ being dense in equivariant function space $C_G(X, Y)$ with respect to the $C^0$ topology. In fact, we can establish a stronger approximation theorem.

**Theorem 5.1.** *In Ex. 2.1, the function families $\mathcal{F}_{\text{TFN}}$ with smooth activation function are $C^\infty$-dense in the space of smooth equivariant maps $C_G^\infty(X, Y)$. That is, for any integer $r \geq 0$, any map $f \in C_G^\infty(X, Y)$, any compact set $K \subset X$, and any $\epsilon > 0$, there exists a function $g \in \mathcal{F}_{\text{TFN}}$ such that*

$$\max_{x \in K} \left\| D^k f(x) - D^k g(x) \right\| < \epsilon, k \leq r. \tag{10}$$

Here, $D^k$ denotes the $k$-th order total derivative operator. A significant portion of maps within a dense parameterization reflects the *generic* properties of the mapping space. For equivariant maps, a key generic property, closely related to almost isovariance, is that the dimension of the set of points where the orbit type is increase from $(H)$ to $(H')$ by a map $f$ is constrained for a generic map. The following theorem shows that for expressive models with $C^\infty$ approximation capabilities, such as the TFN discussed, almost isovariance is a generic property, and full relative isovariance can be achieved by increasing representation multiplicity. As shown in Cex. D.3, this requirement is tight.

**Theorem 5.2.** *Let $\mathcal{F}$ be a equivariant parametrization with $C^\infty$ approximation capability. If for every $(H) \in \mathcal{O}_G(M)$ we have $(p_Y(H)) \in \mathcal{O}_G(Y)$, then for any finite union of compact, smooth $G$-submanifolds $M \subset X$, any $f \in C_G^\infty(X, Y)$, any integer $r \geq 0$, and any $\epsilon > 0$, there exists a map $g \in \mathcal{F}$ such that*

$$\max_{x \in M} \|D^k f(x) - D^k g(x)\| < \epsilon, k \leq r, \tag{11}$$

*and $g|_M$ is almost isovariant relative to $Y$. Furthermore, if the feature space $Y$ contains a representation $\tilde{Y}^{\oplus r}$ for an integer $r > \max_j\{\dim M_j\}$, where $\tilde{Y}$ itself satisfies the condition $(p_{\tilde{Y}}(H)) \in \mathcal{O}_G(\tilde{Y})$, then the approximating map $g|_M$ can be chosen to be isovariant relative to $Y$.*

## 6 EXPERIMENT

We validate our theoretical analysis through three experiments: representation-space visualizations in § 6.1, a geometric graph discrimination task in § 6.2, and a molecular isotropic polarizability prediction task on QM9 (Ramakrishnan et al., 2014) in § 6.3. Across all experiments, we consider two equivariant architectures, TFN (Thomas et al., 2018) and HEGNN (Cen et al., 2024): TFN is used in the first two experiments, whereas HEGNN is employed in the last two. All detailed experimental settings are provided in § F.

## 6.1 VISUALIZATION OF REPRESENTATION SPACE

To provide a clearer illustration of our theory, we present visualizations of the representation spaces of different degrees, obtained from the 3-fold structure.

**Dataset.** We first construct a $k$-fold structure lying in the $xOy$ plane (here $k = 3$), and then apply random rotations to place it on $m$ distinct planes (here $m = 6$), as illustrated in Fig. 4(a).

Subsequently, as shown in Fig. 4(b), each structure is further rotated about the axis perpendicular to its plane. The rotation angle $\theta \in [0, 2\pi/k)$ is uniformly discretized into res candidate values, defined as $\{2\pi i/(k \cdot \text{res})\}_{i=0}^{\text{res}-1}$. Unless otherwise stated, we use res = 49 to verify the half-degeneration. We also consider the doubled resolution res = 98.

**Embeddings.** For the resulting $m \cdot \text{res}$ candidate structures, we compute the graph-level features via randomly initialized single-layer TFN (Thomas et al., 2018) with detailed setting in § F.1. For $l_0 = 0$ the feature dimension is 1; for visualization we set the second coordinate to zero. For $l_0 \geq 1$ the feature dimension is $2l_0 + 1 > 2$; we reduce dimensionality via random projection and then rescale all features to a common range so that visualizations are comparable. Data points are plotted in ascending order of plane index; for structures on the same plane they are ordered by increasing rotation angle about the axis.

**Results.** Detailed experimental results are presented in Figs. 3 and 4. The former shows the input symmetry with $l_0 \leq 11$ increase to $(O(3) \times S_{k+1})$ (Full degeneration, $l_0 = 0, 1$), $(D_{\infty h} \times S_{k+1})$ (Axial degeneration, $l_0 = 2, 4$), or remains non-degenerate. The latter shows the symmetry increase to $(D_{2kh} \times S_{k+1})$ at $l_0 = 10$. These experimental results are consistent with Ex. 4.3.

## 6.2 EXPRESSIVITY ON SYMMETRIC GRAPHS

To experimentally validate our theoretical conclusion established , we design a more comprehensive experiment following Joshi et al. (2023).

**Dataset.** We construct four symmetric $k$-fold structures ($k \in 2, 3, 4, 6$), each centered at the origin. For each structure $\mathcal{G}_0$ we apply a random rotation to obtain $\mathcal{G}_1$, ensuring that $\mathcal{G}_1$ does not coincide with the original $\mathcal{G}_0$. The goal is to evaluate whether different ENNs can distinguish $\mathcal{G}_0$ from $\mathcal{G}_1$. To probe different aspects of our theory, we treat 2D and 3D rotations separately; in the 3D setting we additionally require that $\mathcal{G}_1$ is not coplanar with $\mathcal{G}_0$.

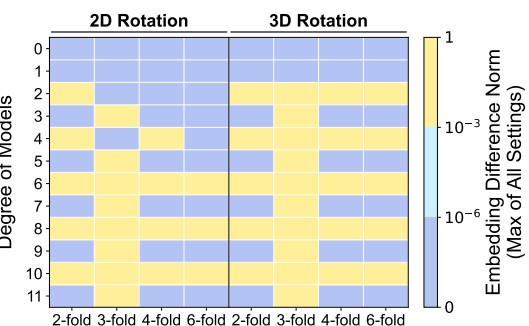

Figure 5: Heatmap of Emb. Diff. Norm.

**Embeddings.** We employed both TFN (Thomas et al., 2018) and HEGNN (Cen et al., 2024) to compute the norm of the embedding difference across 12 configurations for each, varying by the number of irrep channels (1, 4, 16) and layers (1-4). The extracted $l_0$-degree embeddings are evaluated via the norm of the difference between the embeddings of $\mathcal{G}_0$ and $\mathcal{G}_1$ as in Cen et al. (2024) with detailed setting in § F.2.1. When this norm approaches zero, the two embeddings are numerically indistinguishable, and hence the corresponding geometric figures cannot be told apart by the model [3].

**Results.** The maximum value was selected for each configuration and visualized in a heatmap, as shown in Fig. 5. The results exhibit a clear binary pattern: the values are either greater than $10^{-3}$ or less than $10^{-6}$ (due to numerical error), with a difference of more than $10^3$ times. This suggests that values exceeding $10^{-3}$ indicate distinguishable structures, while those below $10^{-6}$ correspond to indistinguishable structures. These findings align precisely with our theoretical predictions. Furthermore, as the maximum value was chosen, with all norms being less than $10^{-6}$, this phenomenon is shown to be independent of the model choice, the number of channels or layers.

## 6.3 MOLECULE PROPERTY PREDICTION WITH PRETRAINED EQUIVARIANT FEATURES

To illustrate the guiding significance of the theory presented in this paper for practical applications, we designed experiments on the QM9 dataset (Ramakrishnan et al., 2014) for verification.

**Dataset.** We choose to predict the molecular isotropic polarizability $\alpha$. It is worth noting that the QM9 dataset (Ramakrishnan et al., 2014) contains many highly symmetric structures spanning 22

---

[3] In Joshi et al. (2023), the embeddings are directly fed to a vanilla classifier. To address issues such as imperfect classifier training and numerical error, we slightly modify this experimental setup. We also reproduce their original experiment in § F.2.2, and the results remain consistent with our theory.

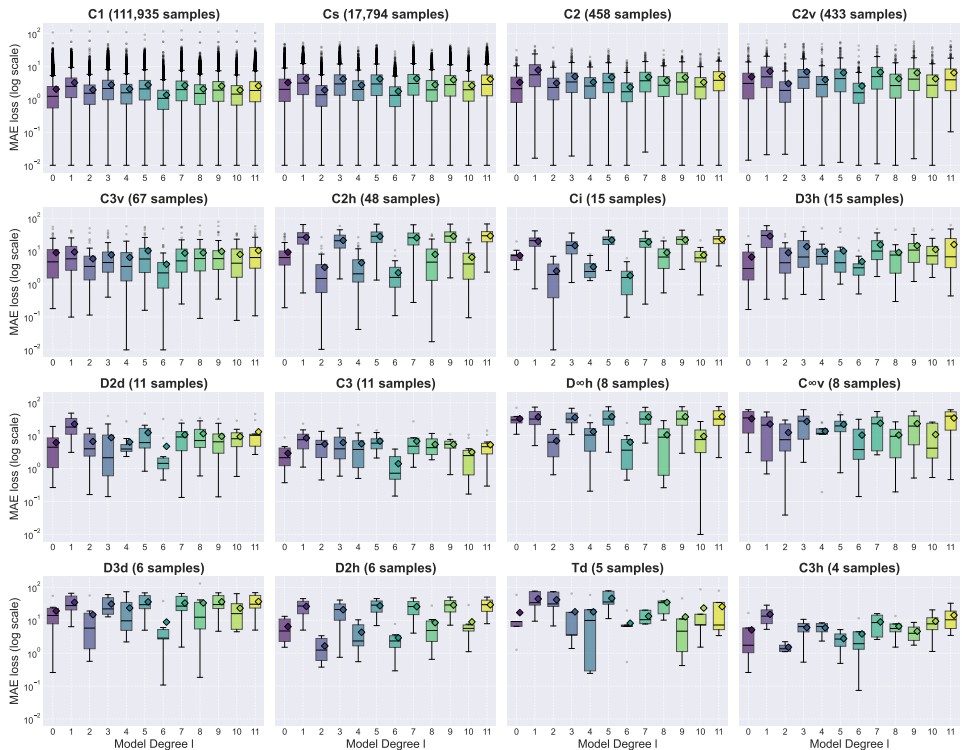

Figure 6: MAE loss (in units of $a_0^3$) for isotropic polarizability prediction with degree $l = l_0$ across molecules from the top-16 point groups by molecular count. Each boxplot shows the distribution of errors at a given degree, while diamond markers denote the corresponding mean MAE.

molecular symmetry groups [4], with Fig. 6 reporting sample counts for the 16 most frequent groups; the remaining seven are $D_3$ with 2 samples and $C_4$, $D_2$, $D_{6h}$, $O_h$, $S_4$, each appearing only once.

**Embeddings.** We adopt HEGNN (Cen et al., 2024) as our backbone. Specifically, we first pretrain on features of $l \leq 11$ to obtain a shared equivariant feature encoder, ensuring that all subsequent configurations operate on the same embedding space. We then consider two fine-tuning strategies: (a) using only features of $l = l_0$, and (b) using all features of $l \leq l_0$, yielding 12 distinct prediction heads for each. The detailed experimental setup is given in § F.3.1.

**Results.** The results are shown in Figs. 6 and 7, with a summary of all symmetry increases in Table 22. We note that for the majority of samples, different feature components contribute similarly to the prediction. However, as illustrated in Fig. 6, for non-trivial feature components where molecular symmetry increase to $O(3)$, the prediction loss is substantially higher. Furthermore, the results in Fig. 7 show that for symmetries causing full degeneration in 1-degree features, including additional 1-degree features may not provide significant improvement to the model's prediction performance. This validates the design guidelines in § 4.2. Detailed case studies are provided in § F.3.2.

## 7 CONCLUSION

In this work, we presented a rigorous mathematical framework to address the critical issue of symmetry increase in ENNs. We introduced the concept of the symmetry infimum, a computable lower bound for any increase in symmetry determined by the feature space. Our central contribution is to show that this infimum can be used to precisely predict and control the expressive degradation of ENNs. The framework successfully explains phenomena in settings like those of Joshi et al. (2023), which could not be fully accounted for by prior theories such as the collapse-to-zero model from Cen et al. (2024). Our findings provide both a robust theoretical understanding and practical guidelines for designing more reliable ENNs.

---

[4]We use the QM9 dataset as provided in PyG (Fey & Lenssen, 2019) and apply the PointGroup library (Carreras, 2025) to pre-compute and manually post-process the point groups of all molecules. As a result, our statistics may differ slightly from those reported in previous works like Zeng et al. (2025).

## ACKNOWLEDGMENTS

This work was supported by the National Natural Science Foundation of China (Nos. 62276269, 92270118, and 62376276), the Beijing Natural Science Foundation (No. 1232009), and the Beijing Nova Program (No. 20230484278).

## AUTHOR CONTRIBUTIONS

Ning Lin organized this project. Ning Lin led the theoretical development in § 2–§ 5 and was responsible for the theoretical proofs of the corresponding part. Ning Lin and Jiacheng Cen jointly led the experimental studies in § 6. Jiacheng Cen was responsible for model implementation and code development. Anyi Li was responsible for data processing. Jiacheng Cen and Anyi Li jointly contributed to figure preparation and visualization. Wenbing Huang and Hao Sun jointly supervised and guided the project. All authors participated in writing and revising the manuscript.

## USAGE OF LARGE LANGUAGE MODELS

We only use Large Language Models to polish our writing.

## ETHICS STATEMENT

This work is a theoretical contribution in the domain of equivariant neural networks, focusing on mathematical properties of symmetry preservation and transformation under equivariant mappings. The research does not involve human subjects, personal data, or real-world deployments, and therefore does not raise concerns related to privacy, fairness, bias, or potential misuse.

We affirm that this work adheres to the ICLR Code of Ethics. In particular, we have ensured honesty in representing our contributions, accuracy in reporting our findings, and proper attribution of prior work. As this is a theoretical study, there are no conflicts of interest, sponsorship influences, or applications with foreseeable societal harms to disclose. We support responsible stewardship of machine learning research and believe that foundational advances such as ours contribute positively to the scientific community by enhancing understanding and enabling future trustworthy systems.

## REPRODUCIBILITY STATEMENT

We are committed to ensuring the reproducibility of both the theoretical and experimental components of our work. To support the reproducibility of our theoretical results, we provide complete and self-contained proofs for all main theorems and propositions in the appendix, including detailed derivations and necessary mathematical background. These proofs clarify all assumptions and logical steps required to verify our claims. For the experimental component, we have made our implementation code publicly available at `https://github.com/GLAD-RUC/SymInc`. The codebase contains clear documentation and scripts that fully reproduce the reported results. All experimental settings, hyperparameters, and data generation procedures are described in detail within the supplementary materials. Together, the comprehensive theoretical appendices and open-sourced code ensure that our findings can be rigorously verified and built upon by the research community.

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

CONTENTS OF APPENDIX

# A  BACKGROUNDS

## A.1  EQUIVARIANT NEURAL NETWORKS

Equivariant Neural Networks (ENNs) have emerged as a cornerstone of modern machine learning, enabling a wide range of applications across the sciences (Han et al., 2025; Wu et al., 2023; Li et al., 2025b; Wu et al., 2025; Li et al., 2025c; Feng et al., 2026; Cao et al., 2026; Wu et al., 2026; Li et al., 2025a). Most mainstream ENNs adopt tensor product operators to design the message passing architecture (Thomas et al., 2018). In particular, since scalarization operations (*e.g.*, norm and inner product) can substantially reduce computational cost, Cartesian vector-based networks (Satorras et al., 2021) and spherical scalarization networks (Cen et al., 2024; Aykent & Xia, 2025) have become particularly popular. More specifically for asymmetric graph structures, Cen et al. (2025) point out that networks employing scalarization operations already possess sufficient expressive power, i.e., universal approximation.

However, additional subtleties arise in the presence of input symmetries. The interaction between architectural components and symmetric structures can lead to nontrivial representational effects. For example, techniques such as global virtual nodes (Zhang et al., 2024; 2025) and reference frames (Puny et al., 2022; Duval et al., 2023), though effective in improving model capacity, may exhibit unintended behaviors when applied to symmetric inputs. In particular, their use can potentially induce symmetry increase. In this work, we systematically analyze these phenomena under symmetric structures for general equivariant neural network architectures.

## A.2  CLOSED SUBGROUPS OF $SO(3)$ OR $O(3)$

Before proceeding, we briefly review the classification of the closed subgroups of $SO(3)$ or $O(3)$. The closed subgroups of $O(3)$, also known as point groups, are classified up to conjugacy as follows. Throughout this paper, we use the Schoenflies notation to denote these groups.

- **Finite Subgroups**: These are divided into axial and polyhedral groups. The axial groups include the Abelian subgroups ($C_k, S_{2k}, C_{kh}$) and the non-Abelian subgroups ($C_{kv}, D_k, D_{kh}, D_{kd}$). The polyhedral groups are all non-Abelian and comprise seven families ($T, T_d, T_h, O, O_h, I, I_h$). Among these, the groups $C_k, D_k, T, O, I$ consist solely of pure rotations and are subgroups of $SO(3)$.

- **Infinite Subgroups**: These include the cylindrical groups ($C_\infty, C_{\infty h}, C_{\infty v}, D_\infty, D_{\infty h}$), which arise as limits of the axial groups, and the spherical groups, which are $K = SO(3)$ and $K_h = O(3)$.

To facilitate the identification of input symmetries, we now provide a brief overview of the elemental structure of the key finite subgroups. The identification of cylindrical symmetries, which are infinite, can be achieved by taking the limits of these finite axial groups.

ABELIAN AXIAL GROUPS

- $C_k$ **(Cyclic Group):** Generated by a rotation $c_k$ of angle $2\pi/k$, with $k$ elements:

    (1) Rotations $c_k^j$ about the principal axis.

- $S_{2k}$ **(Rotation-Reflection Group):** Generated by adding a rotation-reflection element $c_{2k}\sigma_h$ to $C_k$, with $2k$ elements:

    (1) Rotations $c_k^j$ about the principal axis.

    (2) Rotation-reflections $c_{2k}^{2j+1}\sigma_h$ about the principal axis.

- $C_{kh}$ **(Cyclic Groups with Horizontal Reflection):** Generated by adding a horizontal reflection plane $\sigma_h$ to $C_k$, with $2k$ elements:

    (1) Rotations $c_k^j$ about the principal axis.

    (2) Rotation-reflections $c_k^j \sigma_h$ about the principal axis.

Non-Abelian Axial Groups

- $C_{kv}$ **(Cyclic Groups with Vertical Reflections):** Generated by adding a vertical reflection plane $\sigma_v$ to $C_k$, with $2k$ elements:

  (1) Rotations $c_k^j$ about the principal axis.

  (2) Reflections $c_k^j \sigma_v$ about the vertical plane.

- $D_k$ **(Dihedral Groups):** Generated by adding a 2-fold rotation axis $u_2$ perpendicular to the principal axis of $C_k$, with $2k$ elements:

  (1) Rotations $c_k^j$ about the principal axis.

  (2) 2-fold rotations $c_k^j u_2$ about horizontal axes.

- $D_{kh}$ **(Dihedral Groups with Horizontal Reflection):** Generated by adding a horizontal reflection plane $\sigma_h$ to $D_k$, with $4k$ elements:

  (1) Rotations $c_k^j$ about the principal axis.

  (2) 2-fold rotations $c_k^j u_2$ about horizontal axes.

  (3) Rotation-reflections $c_k^j \sigma_h$ about the principal axis.

  (4) Reflections $c_k^j \sigma_v$ about the vertical plane.

- $D_{kd}$ **(Dihedral Groups with Dihedral Reflections):** Generated by adding a diagonal reflection plane $\sigma_d = c_{2k}\sigma_v$ to $D_k$, with $4k$ elements:

  (1) Rotations $c_k^j$ about the principal axis.

  (2) 2-fold rotations $c_k^j u_2$ about horizontal axes.

  (3) Rotation-reflections $c_{2k}^{2j+1}\sigma_h$ about the principal axis.

  (4) Reflections $c_{2k}^{2j+1}\sigma_v$ about the diagonal plane.

Polyhedral Groups

- $T$ **(Tetrahedral Group):** The rotational symmetry group of a tetrahedron, with 12 elements.

- $T_d$ **(Full Tetrahedral Group):** The full symmetry group of a tetrahedron, with 24 elements.

- $T_h$ **(Pyritohedral Group):** Generated by adding an inversion center to $T$, with 24 elements.

- $O$ **(Octahedral Group):** The rotational symmetry group of a cube or octahedron, with 24 elements.

- $O_h$ **(Full Octahedral Group):** The full symmetry group of a cube or octahedron, generated by adding an inversion center to $O$, with 48 elements.

- $I$ **(Icosahedral Group):** The rotational symmetry group of an icosahedron, with 60 elements.

- $I_h$ **(Full Icosahedral Group):** The full symmetry group of an icosahedron, generated by adding an inversion center to $I$, with 120 elements.

The subgroup relations among point groups, as well as the dimensions of fixed-point spaces of $O(3)$ representations, can be readily determined. In particular, they can be derived from the minimal subgroup relations provided in § E.1 and from the dimension table for fixed-point spaces of irreducible $O(3)$ representations established in § E.2.

### A.3  Manifolds and Basic Data Assumption

Following the definition in Milnor (1990), a $C^r$ manifold $M$ is mathematically defined as a subset of some linear space $\mathbb{R}^d$. For every point $x$ in this subset, there must exist a neighborhood $W$ in $\mathbb{R}^d$ such that $W \cap M$ is $C^r$-diffeomorphic to an open set in another linear space $\mathbb{R}^l$. The integer $l$ is known as the dimension of $M$. An alternative, intrinsic definition that does not depend on an embedding space can also be found in Hirsch (1976).

Machine learning often assumes that the data distribution is supported on a submanifold of the input space. The dimension of this manifold characterizes the directions in which the data can vary within

the input space. However, the term manifold is often used loosely in this context and does not perfectly align with its mathematical counterpart (Goodfellow et al., 2016). First, the data manifold may have self-intersections, causing the local dimension of data variation to differ at various points. Second, data belonging to different classes or clusters may possess different structures and potentially different dimensions. Lastly, stochastic factors such as observational noise can prevent the data from forming a strict surface in the input space. We will not address this last factor, instead, we assume that the interference from noise is minimal and ignorable.

To address the first two issues, we can relax the manifold hypothesis by assuming that the data is supported on a finite union of submanifolds within the input space. Since a manifold with multiple connected components can itself be viewed as a disjoint union of connected manifolds, we can assume, without loss of generality, that each of these submanifolds is connected. For theoretical convenience, we further assume that these manifolds are bounded and closed. By the Weierstrass theorem, this is equivalent to assuming they are compact. We briefly outline the justification for these assumptions below:

- **Finite Union of Manifolds:** This assumption has been partially verified experimentally on computer vision datasets (Brown et al., 2022). Theoretically, it encompasses classic cases of self-intersection. A data manifold may arise as the image of a map. However, the image of an immersion or submersion of a manifold can have self-intersections. The image of such a map from a compact manifold is, however, a finite union of compact manifolds.

- **Boundedness of Manifolds:** This assumption stems from the natural assumption that the data distribution itself is bounded.

- **Closedness of Manifolds:** This assumption covers scenarios where the data is defined by a set of well-behaved constraints. For example, the set of points satisfying $n$ independent, differentiable constraint equations in the input space forms a closed submanifold of codimension $n$.

In equivariant representation learning, the data possesses symmetries, meaning that data corresponding to the same physical object can be transformed by symmetry operations to represent different reference frames. These symmetry transformations typically form a group, and since the transformed data is still valid data, the data manifold must be closed under these group transformations, that is, the data manifold must be invariant under the group action. Considering the group action for any given input, we summarize the preceding discussions into the following assumptions about the data:

- **Lie Group Assumption:** The transformation group $G$ is a compact Lie group.
- **Manifold Assumption:** The input space $X$ is a linear space equipped with a linear action $\rho_X$ of the group $G$. That is, there exists a map $\rho_X : G \to GL(X)$ such that $\rho_X(g_1 g_2) = \rho_X(g_1)\rho_X(g_2)$, and the identity element of the group is mapped to the identity transformation. The data manifold $M$ is a finite union of compact, connected, smooth, and $G$-invariant submanifolds of $X$.

The smoothness assumption is added for theoretical convenience and the connectedness assumption does not impose additional restrictions, since each compact smooth submanifold admits only finitely many connected components.

## B  PROOFS OF THE INFIMUM OF SYMMETRY (§ 3)

### B.1  PROOF OF THM. 3.1

**Lemma B.1** (Azzi et al. (2023), Cor. 2.3). *For a compact Lie group $G$ and a representation $V$, let $v \in V$ be any point. There exists a neighborhood $U$ of $v$ such that for any point $u \in U$, we have $(G_u) \le (G_v)$.*

**Lemma B.2** (Azzi et al. (2023), Cor. 2.20). *For a reductive group $G$ and an affine algebraic variety $X$, let $x \in X$ be a point with a closed orbit $G(x)$. There exists a Zariski neighborhood $U$ of $x$ such that for any point $u \in U$, we have $(G_u) \le (G_x)$.*

We use the definition of the complexification from Prop. 3.3 of Azzi et al. (2023). For a real vector space $V$, its complexification is $V^{\mathbb{C}} := \mathbb{C} \otimes V$. Considering an orthogonal action of $G$ on $V$, with $\mathfrak{g}$ being the Lie algebra of $G$, the complexification of the group is $G^{\mathbb{C}} := \{g \exp(iX) \mid g \in G, X \in \mathfrak{g}\}$. By the above complexification, we obtain the following lemma.

**Lemma B.3.** *For a compact Lie group $G$, consider the inclusion map $\iota : G \hookrightarrow G^{\mathbb{C}}$ into its complexification. Let $H$ and $K$ be subgroups of $G$ and let $g_0 \in G$ be an element such that $g_0 H g_0^{-1} \subseteq K$. Then,*

$$\iota(g_0) H^{\mathbb{C}} \iota(g_0)^{-1} \subseteq K^{\mathbb{C}}. \tag{12}$$

*Proof.* Let $\mathfrak{h}$ and $\mathfrak{k}$ be the Lie algebras of $H$ and $K$, respectively. Any element of $H^{\mathbb{C}}$ can be expressed as $h_0 \exp(iX_0)$ for $h_0 \in H$ and $X_0 \in \mathfrak{h}$. Its conjugation by $g_0$ is

$$g_0 \big(h_0 \exp(iX_0)\big) g_0^{-1} = (g_0 h_0 g_0^{-1}) \exp(i\mathrm{Ad}_{g_0} X_0). \tag{13}$$

Since $g_0 H g_0^{-1} \subseteq K$, the term $g_0 h_0 g_0^{-1}$ belongs to $K$. The expression thus belongs to $K^{\mathbb{C}}$ provided that $\mathrm{Ad}_{g_0} X_0 \in \mathfrak{k}$.

This condition on the Lie algebra is obtained by differentiating the subgroup inclusion $g_0 H g_0^{-1} \subseteq K$ at the identity element, which gives $\mathrm{Ad}_{g_0}(\mathfrak{h}) \subseteq \mathfrak{k}$. As $X_0 \in \mathfrak{h}$, it follows immediately that $\mathrm{Ad}_{g_0} X_0 \in \mathfrak{k}$, which completes the proof. $\square$

**Proposition B.4.** *Let $V$ be a representation of a compact Lie group $G$. For any closed subgroup $H$ of $G$, the set of points in the fixed-point subspace $V^H$ that have the minimal orbit type is a dense and open subset of $V^H$.*

*Proof.* The proof strategy is similar to that of Prop. 2.10 in Azzi et al. (2023). We first show that the set of points with a minimal orbit type is open, then use complexification to show it is Zariski-open, which implies density.

First, to prove openness, let $(K)$ be a minimal orbit type in $V^H$ and let $x \in V^H$ be a point with $(G_x) = (K)$. By Lem. B.1, there exists a neighborhood $U$ of $x$ such that for any $u \in U$, $(G_u) \le (K)$. Since $(K)$ is minimal in $V^H$, any point in the open set $U \cap V^H$ must have orbit type $(K)$. Thus, openness is established.

Next, we consider the complexifications $G^{\mathbb{C}}$ and $V^{\mathbb{C}}$. For a point $x \in V$ with orbit type $(K)$, by Lem. 3.13 of Azzi et al. (2023), we have $(G^{\mathbb{C}})_x = (G_x)^{\mathbb{C}} = K^{\mathbb{C}}$. By Prop. 3.14 of Azzi et al. (2023), the orbit $G^{\mathbb{C}}(x)$ is closed. Therefore, by Lem. B.2, there exists a Zariski neighborhood $U$ of $x$ in $V^{\mathbb{C}}$ such that any point in this neighborhood has an isotropy type less than or equal to $(K^{\mathbb{C}})$. The set $U' = U \cap (V^{\mathbb{C}})^{H^{\mathbb{C}}}$ is Zariski-open in $(V^{\mathbb{C}})^{H^{\mathbb{C}}}$. By Cor. A.7 of Azzi et al. (2023), its real part $U'' = U' \cap V$ contains a non-empty real Zariski-open subset of $V^H$.

We now show that for any $y \in U''$, its orbit type is $(G_y) = (K)$. Since $y \in V^H$, we have $(K) \le (G_y)$, which implies there exists $g_1 \in G$ such that $g_1 K g_1^{-1} \subseteq G_y$. By Lem. B.3, this gives $g_1 K^{\mathbb{C}} g_1^{-1} \subseteq (G_y)^{\mathbb{C}}$. Since $y \in U'' \subset U$, we have $((G^{\mathbb{C}})_y) \le (K^{\mathbb{C}})$, which means there exists $g_2 \in G^{\mathbb{C}}$ such that $(G_y)^{\mathbb{C}} \subseteq g_2 K^{\mathbb{C}} g_2^{-1}$. Combining these inclusions yields

$$g_1 K^{\mathbb{C}} g_1^{-1} \subseteq (G_y)^{\mathbb{C}} \subseteq g_2 K^{\mathbb{C}} g_2^{-1}. \tag{14}$$

By Lem. A.3 of Azzi et al. (2023), this implies that the real subgroups are conjugate, i.e., $(G_y) = (g_1 K g_1^{-1}) = (K)$. This completes the proof of density. $\square$

**Theorem 3.1** (Uniqueness of Minimal Type)**.** *Let $X$ be a representation of a compact Lie group $G$. For any closed subgroup $H \subseteq G$, a unique minimal orbit type exists among the points in the fixed-point subspace $X^H$. In particular, if $(H) \in \mathcal{O}_G(X^H)$, then $(H)$ is the minimal orbit type within that subspace.*

*Proof of Thm. 3.1.* The density and openness of the set of points corresponding to any minimal orbit type is established by Prop. B.4. The uniqueness then follows from the fact that two distinct open dense sets must have a non-empty intersection.

$\square$

### B.2 PROOF OF THM. 3.3

**Theorem 3.3.** *A necessary condition for the existence of an isovariant map relative to $Y$ from a $G$-set $X$ to a $G$-set $Y$ is that $(p_Y(H)) \in \mathcal{O}_G(Y)$ for every $(H) \in \mathcal{O}_G(X)$. When $X$ and $Y$ are representations, this is equivalent to the condition that $I_G(Y, H) = (p_Y(H))$ for all $(H) \in \mathcal{O}_G(X)$.*

*Proof of Thm. 3.3.* The equivalence for representations follows directly from applying the operator $p_Y$ to the known partial order $(H) \leq I_G(Y, H) \leq (p_Y(H))$. Since $p_Y$ is order-preserving and fixes isotropy subgroups in $Y$, the inequality collapses to an equality. $\square$

## C  SYMMETRY INCREASE FOR $G = SO(3)$ OR $O(3)$ IN § 4

### C.1  GENERAL ORBIT TYPE CRITERION

To derive Prop. 4.2, we need more general results require criteria for identifying orbit types of general representation. Given the sufficiency of the Ihrig-Golubitsky Criterion (Prop. C.1), we demonstrate that for high-multiplicity representations, the conditions of the Michel Criterion imply the conditions of the Ihrig-Golubitsky Criterion. This, in turn, establishes the sufficiency of the Michel Criterion.

Let $G$ be a compact Lie group and $X$ be the input space. The normalizer of a subgroup $H \subseteq G$ is $N_G(H) := \{g \in G \mid gHg^{-1} = H\}$. The normalizer of $H$ relative to a supergroup $H' \supset H$ is $N_G(H, H') := \{g \in G \mid H \subset gH'g^{-1}\}$. Based on the above definitions, we recall the following general criterion.

**Proposition C.1** (Ihrig-Golubitsky Criterion, Ihrig & Golubitsky (1984), Prop. 5.3). *Let $V$ be a faithful representation of a group $G$. A sufficient condition for a closed subgroup $H$ to be an isotropy subgroup in $V$ is that for every orbit type $(H')$ in $V$ with $(H') > (H)$, the following inequality holds:*

$$\dim V^{H'} + \alpha_G(H, H') := \dim V^{H'} + \dim N_G(H, H') - \dim N_G(H') < \dim V^H, \quad (15)$$

*where $\alpha_G(H, H')$ is the Ihrig-Golubitsky correction term.*

*Remark.* Ihrig & Golubitsky (1984) states that the condition is also necessary for $G = SO(3), O(3)$.

Compared to Prop. C.1, the Linehan-Stedman Criterion (Linehan & Stedman, 2001) is more convenient due to its structure, which only requires checking adjacent subgroups. Since the associated theoretical results are not needed in our proofs, we do not elaborate on them here.

### C.2  PROOF OF PROP. 4.2

**Proposition 4.2.** *For a high-multiplicity representation $V$, the necessary condition stated in Thm. 4.1 is also sufficient.*

*Proof of Prop. 4.2.* We show that for representations where each irreducible component has multiplicity $r > \dim G$, the necessary condition from Michel's Criterion (Thm. 4.1) becomes sufficient by proving it implies the condition in Prop. C.1. We may assume without loss of generality that the representation is faithful. For unfaithful representations, we proceed by factoring out the kernel from $G$ and invoking Prop. C.1, as the procedure remains unchanged.

It follows from the condition that for any closed subgroup $H'$ of $G$ containing $H$, we have $\dim V^{H'} < \dim V^H$. The condition implies that for at least one irreducible component $V_i$, we must have $\dim V_i^{H'} < \dim V_i^H$. Since dimensions are integers, this is equivalent to $\dim V_i^{H'} + 1 \leq \dim V_i^H$. Given that the multiplicity $m(V_i, V) > \dim G$, it follows that

$$m(V_i, V) \dim V_i^{H'} + \dim G < m(V_i, V) \dim V_i^H. \quad (16)$$

Summing this strict inequality for one such component with the non-strict inequalities for all other components yields

$$\dim V^{H'} + \dim G < \dim V^H. \quad (17)$$

The term from Prop. C.1 is bounded by

$$0 \leq \alpha_G(H, H') = \dim N_G(H, H') - \dim N_G(H') \leq \dim G. \quad (18)$$

The sufficiency of Michel's condition follows from combining these two results:

$$\dim V^{H'} < \dim V^H \quad \Longrightarrow \quad \dim V^{H'} + \dim G < \dim V^H \quad (19)$$

$$\Longrightarrow \quad \dim V^{H'} + \dim N_G(H, H') - \dim N_G(H') < \dim V^H, \quad (20)$$

where the final implication uses the upper bound from Eq. (18). This shows that Michel's condition implies the sufficient condition from Prop. C.1. $\square$

### C.3 DETAILED CALCULATION IN EX. 4.3

We now conduct the orbit-type test and symmetry infimum calculation for the geometric symmetry $D_{kh}$ ($k > 2$) of the $k$-fold from Ex. 2.2. Consider $D_{kh}$ as a closed subgroups of $O(3)$, the calculation is performed in the representation space $Y = V_{l=l_0}^{\oplus r}$ for $r > 3$ and $l_0 > 0$. We will use the pre-computed subgroup relations from Table 6, the dimensions of the fixed-point spaces for $D_{kh}$ and $O_h$ in $V_{l=l_0}$ from Table 8, and the orbit type test results for $(D_{\infty h})$ and $(O_h)$ in $V_{l=l_0}^{\oplus r}$.

First, we perform the orbit type test. Consider the case where $k \neq 4$. In this situation, the adjacent supergroups of $D_{kh}$ are of the form $D_{pk,h}$, where $p$ is a prime number. We classify the discussion into 12 cases based on the parity of $k$ and $l_0$, and the ranges $l_0 < k$, $k \leq l_0 < 2k$, and $l_0 \geq 2k$, and calculate the dimensions of the fixed-point spaces of $V_{l=l_0}$ for $D_{kh}$ and $D_{pk,h}$. The calculated dimensions are shown in Table 2. For the case $k = 4$, we must additionally consider the supergroup $O_h$. Here, $\dim V_{l=l_0}^{D_{4h}} = \dim V_{l=l_0}^{O_h}$ only when $l_0$ is odd, which aligns with the results from Table 2. In all other cases, $O_h$ does not affect the orbit type test of $(D_{kh})$. Thus, we find that $(D_{kh})$ is an orbit type when $l_0 \geq k$ and when $l_0$ and $k$ have the same parity.

Table 2: Dimension of fixed-point space for supergroups of $D_{kh}$ ($k > 2$) in $V_{l=l_0}^{\oplus r}$ ($r > 3$), organized by the parity of $k$ and $l_0$.

|  | $l_0 < k$ | | $k \leq l_0 < 2k$ | | $l_0 \geq 2k$ | |
|---|---|---|---|---|---|---|
|  | $l_0$ is even | $l_0$ is odd | $l_0$ is even | $l_0$ is odd | $l_0$ is even | $l_0$ is odd |
| $k$ is even | | | | | | |
| $D_{kh}$ | 1 | 0 | 2 | 0 | $d$ | 0 |
| $D_{pkh}$ | 1 | 0 | 1 | 0 | $< d$ | 0 |
| $k$ is odd | | | | | | |
| $D_{kh}$ | 1 | 0 | 1 | 1 | $d$ | $d$ |
| $D_{2kh}$ | 1 | 0 | 1 | 0 | $d$ | 0 |
| $D_{p^*kh}$ | 1 | 0 | 1 | 0 | $< d$ | $< d$ |

Next, we proceed with the symmetry infimum calculation. Again, we first consider the case where $k \neq 4$. In this situation, the supergroups of $D_{kh}$ are $D_{pk,h}$, $D_{\infty h}$, and $O(3)$, where $O(3)$ is always an orbit type. Using the orbit type test results just calculated and those pre-computed, we analyze the cases based on the parity of $k$ and $l_0$ as before, with the results summarized in Table 3. For $k = 4$, the additional supergroup $O_h$ does not affect the final result. This is because the supergroup $O_h$ is never the minimal isotropy supergroup, as it only becomes an orbit type when $(D_{4h})$ already is. This leads us to the symmetry infimums shown in Table 1.

Table 3: Isotropy conditions for supergroups of $D_{kh}$ ($k > 2$) in $V_{l=l_0}^{\oplus r}$ ($r > 3$), organized by the parity of $k$ and $l_0$. A ✓ indicates the condition is satisfied, and ✗ that it is not. In the subgroup notation, $p$ denotes a prime number and $p^*$ denotes an odd prime number.

|  | $l_0 < k$ | | $k \leq l_0 < 2k$ | | $l_0 \geq 2k$ | |
|---|---|---|---|---|---|---|
|  | $l_0$ is even | $l_0$ is odd | $l_0$ is even | $l_0$ is odd | $l_0$ is even | $l_0$ is odd |
| $k$ is even | | | | | | |
| $D_{kh}$ | ✗ | ✗ | ✓ | ✗ | ✓ | ✗ |
| $D_{pkh}$ | ✗ | ✗ | - | ✗ | - | ✗ |
| $D_{\infty h}$ | ✓ | ✗ | - | ✗ | - | ✗ |
| $k$ is odd | | | | | | |
| $D_{kh}$ | ✗ | ✗ | ✗ | ✓ | ✗ | ✓ |
| $D_{2kh}$ | ✗ | ✗ | ✗ | - | ✓ | - |
| $D_{p^*kh}$ | ✗ | ✗ | ✗ | - | - | - |
| $D_{\infty h}$ | ✓ | ✗ | ✓ | - | - | - |

### C.4 CALCULATION OF SYMMETRY INFIMUM

We calculate the orbit types for the $SO(3)$ representation $V_{l=l_0}^{\oplus r}$ and the $O(3)$ representation $V_{l=l_0^\pm}^{\oplus r}$. As the calculation procedure is highly similar to that in Ex. 4.3, we omit the detailed steps here. For the case of $r = 1$, the results can be found in Table B.1 and Table B.2 of Linehan & Stedman (2001).

The orbit types are calculated using the procedure in Algo. 1. This calculation requires the dimensions of the fixed-point spaces for the closed subgroups of $SO(3)$ or $O(3)$ in the representations $V_{l=l_0}$ and $V_{l=l_0^\pm}$, respectively, which are provided in § E.2.

According to Prop. 6.2 from Ihrig & Golubitsky (1984), the Ihrig-Golubitsky correction term $\alpha(H, H') = 0$ for all subgroups except $H = C_k$ in $SO(3)$, and for all subgroups except $H = C_k, S_{2k}, C_{kh}$ in $O(3)$. Therefore, for all subgroups other than $H = C_k, S_{2k}, C_{kh}$, our results are identical to those calculated by Linehan & Stedman (2001) for $r = 1$. We present only the results of whether $(C_k) \in \mathcal{O}_{SO(3)}(V_{l=l_0}^{\oplus r})$ on Table 4 and whether $(C_k), (S_{2k}), (C_{kh}) \in \mathcal{O}_{O(3)}(V_{l=l_0^-}^{\oplus r})$ on Table 5.

The reason for omitting the discussion of $V_{l=l_0^+}^{\oplus r}$ is consistent with the explanation in the header of Table B.2 in Linehan & Stedman (2001). This is because the corresponding conclusions can be found in the orbit type table for the $SO(3)$ representation $V_{l=l_0}^{\oplus r}$ via the mappings $D_\infty \to D_{\infty h}$, $C_\infty \to C_{\infty h}, I \to I_h, O \to O_h, T \to T_h$, as well as $D_k \to D_{kh}$ and $C_k \to C_{kh}$ for even $k$, and $D_k \to D_{kd}$ and $C_k \to S_{2k}$ for odd $k$.

Table 4: Modified isotropy subgroups for the multiple irreducible representation $V_{l=l_0}^{\oplus r}, r > 3$ of $SO(3)$ obtained via the Michel criterion.

| $H$ | Condition of $l_0$ |
|---|---|
| $C_k$ | $l_0 \geq k$ |

Table 5: Modified isotropy subgroups for the multiple irreducible representation $V_{l=l_0^-}^{\oplus r}, r > 3$ of $O(3)$ with nontrivial reflection, obtained via the Michel criterion.

| $H$ | Condition of $l_0$ |
|---|---|
| $C_k$ | $l_0 \geq k$ |
| $S_{2k}$ ($k$ is even) | $l_0 \geq k$ |
| $C_{kh}$ ($k$ is odd) | $l_0 \geq k$ |

The complete set of orbit type calculation results can also be found in the tables for the symmetry infimum. This is because $(H)$ is an orbit type of a representation $V$ if and only if the symmetry infimum of $H$ in $V$ is $(H)$ itself. These non-degenerate cases are highlighted in green in the tables.

Using the calculated orbit types for the $SO(3)$ representation $V_{l=l_0}^{\oplus r}$ and the $O(3)$ representation $V_{l=l_0^\pm}^{\oplus r}$, we can now compute the symmetry infimum for all subgroups of $SO(3)$ or $O(3)$ in these representations. When calculating the symmetry infimum for all subgroups, the algorithm differs slightly from that in Ex. 4.3. We reduce the need to enumerate all supergroups by leveraging the symmetry infimums of adjacent supergroups. An improved version of Algo. 2 is detailed in Algo. 3.

To reduce the computational load, this algorithm employs a top-down calculation strategy. First, we compute the results for the infinite groups, followed by the polyhedral groups. Finally, we calculate the results for the axial groups in the order $D_{kh}, D_{kd}, D_k, C_{kv}, C_{kh}, S_{2k}, C_k$. Due to the exceptional subgroup relations shown in Table 6, special consideration for additional supergroups is required for certain cases where $k \in \{1, 2, 3, 4, 5\}$. These cases are handled last.

The results for $SO(3)$ are presented in § E.3, and the results for $O(3)$ are in § E.4. Here, we provide a general classification for the observed symmetry increases. For a closed subgroup $H$, an increase

---

**Algorithm 3:** Symmetry Infimum Calculation With Precomputed Results

**Data:** A symmetry group $G$;
       a closed subgroup $H \subset G$;
       a Rep. $V$ of $G$;
       a map $M$ of previously computed symmetry infs.
**Result:** Symmetry inf. $I_G(V, H)$.

1   **if** `is_in`$((H), \mathcal{O}_G(V))$ **then**
2      **return** $(H)$;
3   **end**
4   Let $S_0 = \{H_i\} \subset G$ to be all adjacent closed supergroups of $H$ in $G$;
5   $\mathcal{O} \leftarrow \varnothing$;
6   **for** $H_i$ *in* $S_0$ **do**
7      **if** `is_in`$((H_i), \mathcal{O}_G(V))$ **then**
8          Add $(H_i)$ to the set $\mathcal{O}$;
9      **end**
10      **else if** $H_i$ *is a key in the map* $M$ **then**
11          Add $M[H_i] = I_G(V, H_i)$ to the set $\mathcal{O}$;
12      **end**
13      **else**
14          Let $S_i = \{K_j\}$ to be all adjacent closed supergroups of $H_i$ in $G$;
15          **for** $K_j$ *in* $S_i$ **do**
16              **if** $(K_j) \in \mathcal{O}_G(V)$ **then**
17                  Add $(K_j)$ to the set $\mathcal{O}$;
18              **end**
19          **end**
20      **end**
21   **end**
22   **return** $\min(\mathcal{O})$;

---

to the full group $O(3)$ or $SO(3)$ is termed **full degeneration** and marked in `red` in the tables. An increase to a supergroup of higher dimension than $H$ is termed **continuous degeneration** and marked in `blue`. An increase to a supergroup of the same dimension as $H$ is termed **discrete degeneration** and marked in `yellow` or `light green`. No symmetry increase is **no degeneration** and marked in `green`.

For the subgroup $D_{kh} \subset O(3)$, the classification of degeneration behaviors given after Ex. 4.3 is a special case of this general framework. Specifically, full degeneration is identical to the definition here, axial degeneration corresponds to continuous degeneration, and half degeneration corresponds to discrete degeneration.

We note that for the representation $Y = V_{l=l_0^+}^{\oplus r}$, the action of $G$ has a non-trivial kernel, and thus symmetry increase is inevitable. Let $\pi_Y$ be the natural projection to the quotient $G/\ker\rho_Y$ In this context, the `light green` marks an increase to $\pi_Y^{-1}(\pi_Y(H))$, which, as explained in § 3.2, is the lowest possible symmetry that $H$ can be reduced to when a non-trivial kernel is present. This is therefore a predictable behavior. The `yellow` marks other exceptional cases within discrete degeneration.

## C.5   COMPOSITION PROPERTY OF HIGH-MULTIPLICITY REPRESENTATION

In the tables of § E.3 and § E.4, there exists a special class of subgroups $H$ for which any non-trivial symmetry increase always results in an infimum $(H_0)$ where $H_0$ is an adjacent supergroup. We call $(H_0)$ the **bottleneck** of $H$, denoted by $B_G(H)$, and we say that a subgroup $H$ that possesses a bottleneck satisfies the **bottleneck condition**. These groups that act as bottlenecks satisfy some elegant properties. To demonstrate this, we first prove a structure theorem for the symmetry infimum of high-multiplicity representations.

**Proposition C.2.** *For a high-multiplicity representation $V$, the lowest orbit type in the fixed-point subspace $V^H$ is $(G_{V^H})$, where $G_{V^H} = \bigcap_{x \in V^H} G_x$ is the largest subgroup that leaves $V^H$ invariant. This shows that $I_G(V, H) = (G_{V^H})$.*

*Proof.* Since any element in $V^H$ has at least the symmetry $G_{V^H}$, we only need to prove that $G_{V^H}$ is an isotropy subgroup. Assume, for the sake of contradiction, that $G_{V^H}$ is not an isotropy subgroup. By the sufficiency of the Michel Criterion, there exists a supergroup $K$ such that $G_{V^H} \subsetneq K$ and

$$\dim V^H = \dim V^{G_{V^H}} = \dim V^K \implies V^H = V^{G_{V^H}} = V^K. \tag{21}$$

This means that $K$ leaves all elements of $V^H$ fixed. From this, we derive a contradiction:

$$K \subset \bigcap_{x \in V^H} G_x = G_{V^H}. \tag{22}$$

$\square$

*Remark.* This result implies that for a high-multiplicity representation, the symmetry increase from $(H)$ to $I_G(V, H)$ does not alter the dimension of the fixed-point space. Therefore, for high-multiplicity representations, the dimension of the fixed-point subspace corresponding to the input symmetry group equals that corresponding to the symmetry infimum. Since the fixed-point subspace dimensions for certain closed subgroups exhibit distinct regularities, the behavior of symmetry increase toward these subgroups serves as an indicator of the underlying subspace dimension. For instance, in the cases of $SO(3)$ or $O(3)$, for non-trivial representations, full degeneration corresponds to a $0$-dimensional fixed-point subspace, whereas for finite input subgroups, continuous degeneration corresponds to a $1$-dimensional subspace. However, we must emphasize that when a quantitative assessment of the equivariant feature's expressive capacity is required, relying solely on the symmetry infimum is insufficient, as it yields only coarse, qualitative insights. In such cases, one should directly compute the fixed-point subspace dimension. For calculations related to $SO(3)$ or $O(3)$, see § E.2.

We can prove that these groups acting as bottlenecks satisfy the property that for any irreducible representation $V_0$, if $\dim V_0^H < \dim V_0^{H_0}$, then $\dim V_0^H < \dim V_0^{H'}$ holds for all adjacent supergroups $H'$ of $H$. This is because if we assume there exists an $H'$ such that equality holds, then by the Michel Criterion, $(H)$ must undergo a non-trivial symmetry increase to $(G_{V^H})$ in the high-multiplicity representation $V_0^{\oplus r}$. Since these increases always have the bottleneck $(H_0)$ as their infimum, the inequalities

$$\dim V_0^H \leq \dim V_0^{H_0} \leq \dim V_0^{G_{V^H}} \tag{23}$$

must in fact collapse to $\dim V_0^H = \dim V_0^{H_0}$, which leads to a contradiction.

For any high-multiplicity representation $V$ that satisfies $\dim V^H < \dim V^{H_0}$, there must exist a component corresponding to an irreducible representation $V_0$ in $V$ such that $\dim V_0^H < \dim V_0^{H_0}$. It follows that $\dim V_0^H < \dim V_0^{H'}$ for all adjacent supergroups $H'$ of $H$, which in turn shows that $\dim V^H < \dim V^{H'}$ holds for all such $H'$. Therefore, for high-multiplicity representations, $H_0$ controls the dimension gap of the fixed-point spaces between $H$ and its other adjacent supergroups.

This property allows us to prove a theorem regarding the direct sum of high-multiplicity representations, which in turn establishes a property for the orbit types of such direct sums.

**Theorem C.3.** *Let a subgroup $H$ satisfy the bottleneck condition. For two high-multiplicity representations $V_1$ and $V_2$, we have $(H) \in \mathcal{O}_G(V_1 \oplus V_2)$ if and only if $(H) \in \mathcal{O}_G(V_1)$ or $(H) \in \mathcal{O}_G(V_2)$.*

*Proof.* We have already shown that for any high-multiplicity representation $V$, if $\dim V^H < \dim V^{H_0}$, then $\dim V^H < \dim V^{H'}$ holds for all adjacent supergroups of $H$. Therefore, assuming $(H) \in \mathcal{O}_G(V_1 \oplus V_2)$, the Michel Criterion gives us

$$\dim V_1^H + \dim V_2^H < \dim V_1^{H'} + \dim V_2^{H'}. \tag{24}$$

By taking $H' = H_0 = B_G(H)$, we see that at least one of the following two conditions must be true:

$$\dim V_i^H < \dim V_i^{H_0}, \quad i = 1, 2. \tag{25}$$

This implies that for all adjacent supergroups of $H$, at least one of the following two conditions must hold:

$$\dim V_i^H < \dim V_i^{H'}, \quad i = 1, 2. \tag{26}$$

Therefore, $(H)$ is an orbit type of either $V_1$ or $V_2$. $\qquad\square$

The above theorem shows that when we construct a high-multiplicity representation for which $(H)$ is an orbit type using high-multiplicity components, the only way is to find a high-multiplicity component that already contains $(H)$ as an orbit type.

For $SO(3)$, all closed subgroups satisfy the bottleneck condition. Consequently, for high-multiplicity representations, we have $\mathcal{O}_G(V_1 \cup V_2) = \mathcal{O}_G(V_1 \oplus V_2)$. This means that $I_G(V_1 \oplus V_2, H)$ will be the minimum of $I_G(V_1, H)$ and $I_G(V_2, H)$. Furthermore, the set of representations $\{V_{l=l_0}^{\oplus r}\}$ is sufficient to generate all closed subgroups as orbit types, because for any closed subgroup, there always exists a high-multiplicity representation for which it is an orbit type.

For $O(3)$, the bottleneck condition does not necessarily hold. For example, for $H = C_\infty$, a symmetry increase can result in either $D_\infty$ or $C_{\infty v}$, which shows that no bottleneck group exists. The fact that $C_\infty$ never appears as an orbit type in any representation $V_{l=l_0^\pm}^{\oplus r}$ demonstrates this point precisely.

This introduces a subtle issue when we apply the guideline on § 4.2: for certain orbit types, a representation exhibiting the target orbit type cannot be constructed simply by including the component $V_{l=l_0^\pm}^{\oplus r}$ associated with it. Fortunately, $C_\infty$ is the sole instance of this phenomenon encountered when $G = O(3)$. Regarding the construction of a $O(3)$ representation containing $C_\infty$, following Prop. 4.2, it suffices to simultaneously select components $V_{l=l_0^-}^{\oplus r}$ with both odd and even degrees $l_0 > 0$.

# D  PROOFS OF DENSITY OF (ALMOST) ISOVARIANT MAPS (§ 5)

## D.1  COUNTEREXAMPLE IN § 5.1

**Lemma D.1** (Borsuk-Ulam Theorem, Guillemin & Pollack (1974), Chap. 2, Sec. 6). *For any continuous odd function $g : S^n \to \mathbb{R}^n$, there exists a point $x \in S^n$ such that $g(x) = 0$.*

**Lemma D.2** (Weak Borsuk-Ulam Theorem, Nagasaki (2003), Thm. A). *Let $M$ and $N$ be $G$-spheres in a representation space for a compact group $G$. If there exists an equivariant map from $M$ to $N$, then the following dimensional inequality holds:*

$$\varphi_G(\dim M - \dim M^G) \leq \dim N - \dim N^G, \tag{27}$$

*where $\varphi_G : \mathbb{N} \to \mathbb{N}$ is a non-decreasing function that diverges to infinity.*

**Counterexample D.3.** *For a compact Lie group $G$, consider a representation $\tilde{Y}$ with no trivial component, i.e. $\tilde{Y}^G = \{0\}$. Let $Y = \tilde{Y}^{\oplus r}$ and $X = Y^{\oplus (n_0+1)}$ for some $r, n_0 > \dim G$. By Prop. 4.2, we have $\mathcal{O}_G(X) = \mathcal{O}_G(Y)$, yet no isovariant map exists from the unit sphere in $X$ to $Y$ for a sufficiently large integer $n_0$.*

*In particular, for $G = \mathbb{Z}_2$, if $\tilde{Y}$ is the non-trivial irreducible representation, then for any $X = \tilde{Y}^{\oplus r_1}$ and $Y = \tilde{Y}^{\oplus r_2}$ with $r_1 > r_2$, no isovariant map exists from the unit sphere of $X$ to $Y$.*

*Proof.* For any compact Lie group $G$, consider an equivariant map $f$ from a $G$-sphere $M$ in a vector space $X$ to a $G$-representation $Y$. We assume that the multiplicities of the trivial representation components in both $X$ and $Y$ are zero, i.e., $X^G = \{0\}$ and $Y^G = \{0\}$. Therefore, only the origin in $X$ and $Y$ has the orbit type $(G)$, and the sphere $M$ does not contain any point of orbit type $(G)$. Consequently, if $f$ is an isovariant map, it must have no zeros.

From such a zero-free map $f$, we can define a map $\tilde{f}$ to the $G$-sphere $N$ in $Y$:

$$\tilde{f} : M \to N, \quad \tilde{f}(x) = f(x)/\|f(x)\|_2. \tag{28}$$

Since scaling in a vector space does not change the orbit type, $\tilde{f}$ is also an isovariant map. Therefore, by Lem. D.2, we obtain the dimensional relation:

$$\varphi_G(\dim M - \dim M^G) = \varphi_G(\dim X - 1) \leq \dim N - \dim N^G = \dim Y - 1. \tag{29}$$

Since $\varphi_G$ is a non-decreasing function that diverges to infinity, there must exist an integer $n_0 > \dim G$ such that $\varphi_G(n_0) > \dim Y - 1$. This implies that

$$\varphi_G(\dim(Y^{\oplus(n_0+1)}) - 1) > \dim Y - 1. \tag{30}$$

This shows that no isovariant map can exist from the unit $G$-sphere $M$ in the space $X = Y^{\oplus(n_0+1)}$ to the space $Y$. To complete the counterexample, let $Y = \tilde{Y}^{\oplus r}$, where $r > \dim G$. By Prop. 4.2, we have $\mathcal{O}_G(Y) = \mathcal{O}_G(X)$, yet no isovariant map exists between them.

For the special case of $G = \mathbb{Z}_2$, let $\tilde{Y}$ be the non-trivial irreducible representation. Let $X = \tilde{Y}^{\oplus r_1}$ and $Y = \tilde{Y}^{\oplus r_2}$ with $r_1 > r_2$. A map $f$ from the unit sphere in $X$ to $Y$ is equivariant if and only if it is an odd function. Since the representations have no trivial components, an isovariant map is equivalent to an odd function that has no zeros. However, by Lem. D.1, any odd map from a sphere in a higher-dimensional space to a lower-dimensional space must have a zero. Thus, no such isovariant map exists. $\square$

## D.2  PROOF OF THM. 5.1

TFN is an ENN based on tensor products of hidden features. The TFN architecture is composed of a feature lifting map followed by an equivariant pooling stage. The feature lifting map contains equivariant convolutional layers that update node features by aggregating information from neighbors. These convolutions employ filters built from learnable radial functions parameterized by a Multi-Layer Perceptron (MLP) and real spherical harmonics $Y_{lm}$.

Mathematically, the parameterized map $f_{\text{TFN}} \in \mathcal{F}_{\text{TFN}}$ is defined as a sum over feature channels. Each term in the sum is a composition $f_{\text{pool}} \circ f_{\text{feat}}$, consisting of a feature lifting map followed by

equivariant linear pooling. Let $Z$ represent the hidden features associated with each node. The feature lifting map $f_{\text{feat}}$ takes the input point cloud $X$ to node features $Z \otimes \mathbb{R}^n$. This map is constructed as a composition of layers:

$$f_{\text{feat}} = \pi_{Z \otimes \mathbb{R}^n} \circ (f^{(L)}, \text{id}) \circ \cdots \circ (f^{(1)}, \text{id}) \circ \text{ext} \circ C, \tag{31}$$

where $C$ is a centering operation, $\text{ext}$ is a constant extension map, and each layer $f^{(k)}$ updates the node features $v^{(k-1)}$ to $v^{(k)}$ according to the rule:

$$v^{(k)}_{il_3m_3} = \theta v^{(k-1)}_{il_3m_3} + \sum_{j \neq i} \sum_{l_1,m_1,l_2,m_2} C^{(l_3,m_3)}_{(l_2,m_2),(l_1,m_1)} F^{(l_2)}_{m_2}(x_i - x_j) v^{(k-1)}_{jl_1m_1}. \tag{32}$$

Here, $v_{il_3m_3}$ corresponds to the feature of type $l_3$ at node $i$, and $C^{(l_3,m_3)}_{(l_2,m_2),(l_1,m_1)}$ are the Clebsch-Gordan coefficients. The filter function is defined as $F^{(l)}_m(x) = h_l(\|x\|_2) Y_{lm}(x/\|x\|_2)$, where the radial function $h_l : \mathbb{R}_{\geq 0} \to \mathbb{R}$ is parameterized by a Multi-Layer Perceptron (MLP). For $G = O(3) \times S_n$, the parities of $l_1$ and $l_3$ must also be considered. We denote by $\mathcal{F}_{\text{TFN}}$ the resulting family of TFN filters under MLP-parameterized radial functions. In contrast, when $h_l$ is chosen as polynomial parameterization, we denote the corresponding parameterization by $\mathcal{F}^{\text{poly}}_{\text{TFN}}$.

We use $L(X,Y)$ to denote the vector space of linear maps $X \to Y$, and $P(X,Y)$ the vector space of polynomial maps $X \to Y$. Given the $G$-actions on $X$ and $Y$, we write $L_G(X,Y) \subseteq L(X,Y)$ and $P_G(X,Y) \subseteq P(X,Y)$ for the subspaces of $G$-equivariant maps, i.e., $f(g(x)) = g(f(x))$ for all $g \in G$ and $x \in X$. If $Y$ carries the trivial action, equivariance reduces to invariance.

**Lemma D.4.** *Let $X, Y, Z$ be representations of a group $G$. Consider a subset of $G$-equivariant polynomial maps $S \subseteq P_G(X,Z)$ that satisfies the spanning condition:*

$$P(X,Y) \subseteq \text{span}(L(Z,Y) \circ S) := \text{span}(\{A \circ p \mid A \in L(Z,Y), p \in S\}). \tag{33}$$

*Then, it follows that $P_G(X,Y) = \text{span}(L_G(Z,Y) \circ S)$.*

*In the special case where $G = H_1 \times H_2$, the spanning condition can be relaxed to*

$$P_{H_2}(X,Y) \subseteq \text{span}(L(Z,Y) \circ S). \tag{34}$$

*Remark.* The case for $G = SO(3) \times S_n$ is from Thm. 1 of Dym & Maron (2021). In this context, the action of $S_n$ on the representations is faithful, while we are concerned with features that are invariant under $S_n$.

*Proof.* Let $f \in P_G(X,Y)$. Since $P_G(X,Y) \subseteq P(X,Y)$, by the spanning condition, $f$ can be written as a linear combination of compositions:

$$f = \sum_{i=1}^N A_i \circ p_i, \tag{35}$$

where $p_i \in S$ and $A_i \in L(Z,Y)$. Here, we have absorbed the expansion coefficients into the linear maps $A_i$.

Since $f$ is $G$-equivariant, it is a fixed point of the group averaging operator. Applying this operator to both sides of the equation gives:

$$f(x) = \int_G \rho_Y(g^{-1}) f(\rho_X(g)(x)) \, dg \tag{36}$$

$$= \int_G \rho_Y(g^{-1}) \left( \sum_{i=1}^N A_i(\rho_X(g)(x)) \right) dg \tag{37}$$

$$= \sum_{i=1}^N \left( \int_G \rho_Y(g^{-1}) A_i \rho_Z(g) \, dg \right) p_i(x). \tag{38}$$

In the last step, we used the fact that the maps $p_i \in S$ are themselves $G$-equivariant and moved the integral inward. Let us define the averaged linear maps as

$$(A_G)_i := \int_G \rho_Y(g^{-1}) A_i \rho_Z(g) \, dg. \tag{39}$$

By construction, each $(A_G)_i$ is a $G$-equivariant linear map, i.e., $(A_G)_i \in L_G(Z, Y)$. The expression for $f(x)$ can now be written as $f(x) = \sum_{i=1}^{N} (A_G)_i \circ p_i(x)$. This shows that any map in $P_G(X, Y)$ can be expressed as a linear combination of compositions of maps from $S$ and equivariant linear maps from $L_G(Z, Y)$. Therefore,

$$P_G(X, Y) \subseteq \text{span}(L_G(Z, Y) \circ S). \tag{40}$$

The reverse inclusion, $\text{span}(L_G(Z, Y) \circ S) \subseteq P_G(X, Y)$, is true by definition, since the composition of two equivariant maps is equivariant. Thus, the equality $P_G(X, Y) = \text{span}(L_G(Z, Y) \circ S)$ is established.

The proof for the relaxed condition when $G = H_1 \times H_2$ follows the exact same logic, by taking an $f \in P_G(X, Y) \subseteq P_{H_2}(X, Y)$ and averaging over the group $H_1$. $\qquad \square$

**Lemma D.5.** *Consider the TFN model with polynomial parametric radial function, a input space $X$, a lifted representation space $Z$, and a final output space $Y$. Suppose the final output space $Y = \tilde{Y} \otimes \mathbb{R}^n$, where $\tilde{Y}$ is a representation of $SO(3)$ or $O(3)$, and the symmetric group $S_n$ acts on $Y$ by permuting the coordinates in the $\mathbb{R}^n$ factor. Then the family of polynomial feature maps, $\mathcal{F}_{\text{feat}}^{\text{poly}} \subseteq P_G(X, Z)$, used in TFN satisfies the relaxed spanning condition:*

$$P_{S_n}(X, Y) \subseteq \text{span}(L(Z, Y) \circ \mathcal{F}_{\text{feat}}^{\text{poly}}). \tag{41}$$

*Remark.* Lem. 4 in Dym & Maron (2021) proves the case for $G = SO(3) \times S_n$. The proof for $O(3)$ is similar to that for $SO(3)$ and is therefore not repeated here. It is worth noting that in the $O(3)$ case, the spherical harmonics map can still be used to construct the component-wise map from representation $V_{l=1^-} \cong \mathbb{R}^3$ to symmetric algebra $\text{Sym}_k(V_{l=1^-})$, which in turn is used to build the lifted representation.

**Lemma D.6.** *In Ex. 2.1, the function families $\mathcal{F}_{\text{TFN}}^{\text{poly}}$ contain all equivariant polynomial maps.*

*Proof.* According to Lem. D.4, to prove that a family of equivariant maps contains all equivariant polynomials, it is sufficient to show that it satisfies the (potentially relaxed) spanning condition. We verify this for $\mathcal{F}_{\text{TFN}}^{\text{poly}}$.

By Lem. D.5, for $Y = \tilde{Y} \otimes \mathbb{R}^n$, the family $\mathcal{F}_{\text{feat}}^{\text{poly}}$ satisfies the spanning condition

$$P_{S_n}(X, Y) \subseteq \text{span}(L(Z, Y) \circ \mathcal{F}_{\text{feat}}^{\text{poly}}). \tag{42}$$

We now only need to verify the following spanning condition:

$$P_{S_n}(X, \tilde{Y}) \subseteq \text{span}(L(Z, \tilde{Y}) \circ \mathcal{F}_{\text{feat}}^{\text{poly}}). \tag{43}$$

For any $g \in P_{S_n}(X, \tilde{Y})$, we construct an $S_n$-equivariant polynomial map to $Y$ as

$$\tilde{g} : X \to Y, \quad \tilde{g}(x_1, \ldots, x_n)_j = g(x_1, \ldots, x_n) \quad \text{for } j = 1, \ldots, n. \tag{44}$$

This function is clearly an $S_n$-equivariant polynomial because all $n$ components of its output are identical. Thus, $\tilde{g} \in P_{S_n}(X, Y)$. By the spanning condition, we can write

$$\tilde{g} = \sum_i A_i \circ p_i, \quad \text{where } A_i \in L(Z, Y) \text{ and } p_i \in \mathcal{F}_{\text{feat}}^{\text{poly}}. \tag{45}$$

Applying the averaging operator to both sides yields

$$\frac{1}{|S_n|} \sum_{\sigma \in S_n} (\tilde{g}(\sigma(x)))_{\sigma(j)} = g(x) = \sum_i (A_{S_n})_i \circ p_i(x), \tag{46}$$

where

$$(A_{S_n})_i = \frac{1}{|S_n|} \sum_{\sigma \in S_n} \pi_{\tilde{Y}} \circ \rho_Y(\sigma^{-1}) \circ A_i \circ \rho_Z(\sigma) \in L_G(Z, \tilde{Y}). \tag{47}$$

This establishes the spanning condition

$$P_{S_n}(X, \tilde{Y}) \subseteq \text{span}(L_G(Z, \tilde{Y}) \circ \mathcal{F}_{\text{feat}}^{\text{poly}}). \tag{48}$$

$$\square$$

We consider the higher-order approximation theorem of MLPs, and obtain $C^\infty$-density via higher-order approximations based on polynomial parameterization.

**Lemma D.7** (Higher-Order Approximation Theorem for MLPs, Pinkus (1999), Thm. 4.1). *For any compact set $K \subset \mathbb{R}^n$, any function $f \in C^m(\mathbb{R}^n)$, and any non-polynomial activation function $\sigma \in C^m(\mathbb{R})$, let $\epsilon > 0$. Then there exists an MLP parameterized map $g_\theta$ with activation function $\sigma$ such that*

$$\max_{x \in K} |\partial_{x_1}^{k_1} \ldots \partial_{x_n}^{k_n} f(x) - \partial_{x_1}^{k_1} \ldots \partial_{x_n}^{k_n} g_\theta(x)| < \epsilon \tag{49}$$

*for all non-negative integers $k_1, \ldots, k_n$ with $k_1 + \cdots + k_n < m$.*

**Theorem 5.1.** *In Ex. 2.1, the function families $\mathcal{F}_{\mathrm{TFN}}$ with smooth activation function are $C^\infty$-dense in the space of smooth equivariant maps $C_G^\infty(X, Y)$. That is, for any integer $r \geq 0$, any map $f \in C_G^\infty(X, Y)$, any compact set $K \subset X$, and any $\epsilon > 0$, there exists a function $g \in \mathcal{F}_{\mathrm{TFN}}$ such that*

$$\max_{x \in K} \left\| D^k f(x) - D^k g(x) \right\| < \epsilon, k \leq r. \tag{10}$$

*Proof of Thm. 5.1.* On the domain of analyticity of a function, its Taylor polynomials and their derivatives converge locally uniformly to the function. Therefore, an analytic function and its derivatives can be uniformly approximated by a polynomial and its derivatives on any compact set. This establishes the $C^\infty$-density of polynomial functions in the set of analytic functions. Furthermore, by Chap. 2, Thm. 5.1 in Hirsch (1976), the set of analytic functions is $C^\infty$-dense in the space of smooth functions. Consequently, polynomials are $C^\infty$-dense in the space of smooth functions. By applying the group averaging operator, equivariant polynomials are also $C^\infty$-dense in the space of equivariant functions.

We use mathematical induction to prove the $C^\infty$-density of $\mathcal{F}_{\mathrm{TFN}}$. Let

$$\tilde{f}^{(k)} = (f^{(n)}, \mathrm{id}) \circ \cdots \circ (f^{(1)}, \mathrm{id}) \circ \mathrm{ext} \circ C. \tag{50}$$

We prove that for the TFN, the MLP-based parameterized map $\tilde{f}_{\mathrm{MLP}}^{(n)}$ can approximate the polynomial-based parameterized map $\tilde{f}_{\mathrm{poly}}^{(n)}$. After establishing the approximation property of the feature maps, and noting that the equivariant pooling maps coincide in both settings, we obtain the $C^\infty$-approximation of $\mathcal{F}_{\mathrm{TFN}}^{\mathrm{poly}}$ by $\mathcal{F}_{\mathrm{TFN}}$. The desired approximation result then follows directly from Lem. D.6.

For the base case $n = 1$, there is no difference in the derivatives with respect to the $X$ component corresponding to the identity map. Therefore, we only need to discuss the output of $f^{(1)}$. Since

$$\tilde{f}_{il_3 m_3}^{(1)}(x_1, \ldots, x_n) = \theta + \Delta_{il_3 m_3}^{(1)}, \quad i = 1, 2, \ldots, n, \tag{51}$$

we only need to consider the approximation of the derivatives of $\Delta_{il_3 m_3}^{(1)}$. Expanding $\Delta_{il_3 m_3}^{(1)}$ yields

$$\Delta_{il_3 m_3}^{(1)} = \sum_{l_1, m_1, l_2, m_2} C_{(l_2, m_2), (l_1, m_1)}^{(l_3, m_3)} \sum_{j \neq i} h_{l_2}^{(1)}(\|x_i - x_j\|_2) Y_{l_2 m_2}((x_i - x_j)/\|x_i - x_j\|_2). \tag{52}$$

The difference between $\tilde{f}_{\mathrm{MLP}}^{(k)}$ and $\tilde{f}_{\mathrm{poly}}^{(k)}$ lies in the parameterization of $h_{l_2}$. When taking derivatives of any order of $\Delta_{il_3 m_3}$, each term in the result is a product of derivatives of $h$ of various orders and a fixed function, and the number of terms is finite. By Lem. D.7, these terms can be approximated to any precision on a compact set.

Now, assume the conclusion holds for $n = k - 1$. We will prove that it also holds for $n = k$. Let

$$v_{il_3 m_3}^{(k-1)} = \tilde{f}_{il_3 m_3}^{(k-1)}(x_1, \ldots, x_n). \tag{53}$$

The update rule is given by

$$\tilde{f}_{il_3 m_3}^{(k)}(x_1, \ldots, x_n) = \theta v_{il_3 m_3}^{(k-1)} + \Delta_{il_3 m_3}^{(k)}, \quad i = 1, 2, \ldots, n. \tag{54}$$

By the inductive hypothesis, the first term can be approximated by an MLP model to any precision. We observe the second term. Similarly, expanding $\Delta_{il_3 m_3}^{(k)}$ yields

$$\Delta_{il_3 m_3}^{(k)} = \sum_{l_1, m_1, l_2, m_2} C_{(l_2, m_2), (l_1, m_1)}^{(l_3, m_3)} \sum_{j \neq i} h_{l_2}^{(k)}(\|x_i - x_j\|_2) Y_{l_2 m_2}((x_i - x_j)/\|x_i - x_j\|_2) v_{il_1 m_1}^{(k-1)}. \tag{55}$$

The derivative of this term is a finite sum of products, where each product involves derivatives of various orders of both $h_{l_2}^{(k)}$ and $v_{il_1m_1}^{(k-1)}$.

To show that the error of this term can be controlled, we only need to prove that for a vector space $X$, if for any compact set $K \subset X$, we consider maps $f \in C(X)$ and $g \in C(X)$, and for any $\epsilon_1 > 0$ there exists $f_1 \in R_1$, and for any $\epsilon_2 > 0$ there exists $g_1 \in R_2$ such that

$$\max_{x \in K} \|f(x) - f_1(x)\| < \epsilon_1, \quad \max_{x \in K} \|g(x) - g_1(x)\| < \epsilon_2, \tag{56}$$

then for any $\epsilon > 0$, there exist $f_1 \in R_1, g_1 \in R_2$ such that

$$\max_{x \in K} \|f(x)g(x) - f_1(x)g_1(x)\| < \epsilon. \tag{57}$$

We bound the term as follows:

$$\begin{aligned}
\|f(x)g(x) - f_1(x)g_1(x)\| &\leq \|f(x)\|\|g(x) - g_1(x)\| + \|g_1(x)\|\|f(x) - f_1(x)\| \\
&\leq \|f(x)\|\epsilon_2 + \|g_1(x)\|\epsilon_1 \\
&\leq (\|f_1(x)\| + \epsilon_1)\epsilon_2 + \|g_1(x)\|\epsilon_1 \\
&= \|f_1(x)\|\epsilon_2 + \|g_1(x)\|\epsilon_1 + \epsilon_1\epsilon_2.
\end{aligned} \tag{58}$$

Since a continuous function on a compact set attains its maximum, let $L_1$ be the maximum of $\|f_1(x)\|$ on $K$ and $L_2$ be the maximum of $\|g_1(x)\|$ on $K$. We get

$$\max_{x \in K} \|f(x)g(x) - f_1(x)g_1(x)\| \leq L_2\epsilon_1 + L_1\epsilon_2 + \epsilon_1\epsilon_2. \tag{59}$$

This shows that the error can be made arbitrarily small, completing the proof. $\qquad \square$

### D.3 Some Results on Topology

Here we adopt the definition of $N_G(H)$ and $N_G(H, H')$ from § C.1, and consider the twisted product between $G$-sets as defined in Chap. 2, Sec. 2 of Bredon (1972). We denote $X_H := \{x \in X \mid G_x = H\}$, from which we obtain the following lemma on topology.

**Lemma D.8.** *When the action of a group $G$ on a $G$-manifold $M$ is faithful, the following decompositions hold:*

$$M_{(H')} = M_{H'} \times_{N_G(H')/H'} G/H' \tag{60}$$

*and*

$$M_{(H')}^H = M_{H'} \times_{N_G(H')/H'} N_G(H, H')/H'. \tag{61}$$

*Proof.* The first identity is derived from Thm. 1.31 in Meinrenken (2003), which states the existence of a homeomorphism

$$\sigma : M_{H'} \times_{N_G(H')/H'} G/H' \to M_{(H')}, \quad \text{given by} \quad \sigma([x, gH']) = g(x). \tag{62}$$

We now prove that $\sigma(M_{H'} \times_{N_G(H')/H'} (N_G(H, H')/H')) = M_{(H')}^H$. The second identity then follows by restricting this homeomorphism.

($\subseteq$) Take an element $[x, gH']$ where $x \in M_{H'}$ and $g \in N_G(H, H')$. Its image under $\sigma$ is $g(x)$. The isotropy subgroup is $G_{g(x)} = gG_x g^{-1} = gH'g^{-1}$. Since $g \in N_G(H, H')$, we have $H \subseteq gH'g^{-1}$. This implies $g(x) \in M^H$, and since its orbit type is $(H')$, we have $g(x) \in M_{(H')}^H$.

($\supseteq$) Take any $y \in M_{(H')}^H$. This means $H \subseteq G_y$ and $(G_y) = (H')$. The latter implies there exists a $g_0 \in G$ such that $G_y = g_0 H' g_0^{-1}$. The condition $H \subseteq g_0 H' g_0^{-1}$ implies that $g_0 \in N_G(H, H')$. Let $x = g_0^{-1}(y)$. Then $G_x = H'$, so $x \in M_{H'}$. We can then write $y$ as

$$y = g_0(g_0^{-1}(y)) = g_0(x) = \sigma([x, g_0 H']). \tag{63}$$

Since $x \in M_{H'}$ and $g_0 \in N_G(H, H')$, this shows that $y$ is in the image of the restricted domain. The inclusion is thus proven. $\qquad \square$

From the following proposition onward, we need to invoke stratification theory. For the definitions of Whitney conditions and stratifications, we refer to Chap. 3, Sec. 9 of Field (2007). Going forward, unless otherwise specified, all function spaces in this paper are equipped with the (weak) $C^\infty$ topology. For the topology on function spaces, we refer to Chap. 2, Sec. 1 of Hirsch (1976).

**Proposition D.9.** *Consider smooth manifolds $M$ and $N$. Let $(S_\alpha, S_\beta)$ be a pair of submanifolds in $N$ that satisfies Whitney's condition (a). If a map $f : M \to N$ is transverse to $S_\alpha$ at a point $x \in f^{-1}(S_\alpha)$, then there exists a neighborhood $U$ of $f$ in $C^\infty(M, N)$ and a neighborhood $V$ of $x$ in $M$, such that for any map $g \in U$ and any point $y \in V$, $g$ is transverse to both $S_\alpha$ and $S_\beta$ at $y$.*

*Proof.* The proof is adapted from Prop. 1.3 of Trotman (1976). We proceed by contradiction. Assume the conclusion is false. Then for any neighborhood $V$ of $x$ and any neighborhood $U$ of $f$, there exists a map $g \in U$ for which the condition $g \pitchfork S_\alpha, S_\beta$ on $V$ does not hold.

For a neighborhood $V_1$ of $x$ with radius less than $\epsilon_1$, we can construct a sequence of maps $\{g_n^{(1)}\}$ converging to $f$ such that transversality to $S_\alpha$ or $S_\beta$ fails on $V_1$. Let the sequence of non-transverse points be $\{y_n^{(1)}\}$. We can similarly construct, for neighborhoods $V_t$ of $x$ with radius less than $\epsilon_t$, sequences of maps $\{g_n^{(t)}\}$ converging to $f$ and corresponding sequences of non-transverse points $\{y_n^{(t)}\}$.

Now, consider the diagonal sequence of maps $\{g_n^{(n)}\}$ and the corresponding sequence of non-transverse points $\{y_n^{(n)}\}$. We have $g_n^{(n)} \to f$ and $y_n^{(n)} \to x$. We can partition the sequence $\{y_n^{(n)}\}$ into a subsequence lying in $S_\alpha$ and another lying in $S_\beta$. At least one of these two subsequences must be infinite. It is therefore sufficient to negate the following proposition: There exists a sequence of maps $\{g_n\} \to f$ and a sequence of points $\{y_n\} \to x$ with either $\{y_n\} \subset S_\alpha$ or $\{y_n\} \subset S_\beta$, such that for each $n$, $g_n$ is not transverse to $S_\alpha$ or $S_\beta$ at $y_n$.

This is impossible. Note that for a sufficiently small $\epsilon_1$, the closure of the neighborhood $V_1$ is compact by local compactness of the manifold, and we can consider the control of the neighborhood $U$ over functions on this compact closure in the $C^\infty$ topology of functioin space. Also note that $\mathrm{d}(g_n)_{y_n}(T_{y_n}M)$ converges to $\mathrm{d}f_x(T_xM)$. For the case of $S_\alpha$, by the smoothness of the manifold, $T_{g_n(y_n)}S_\alpha$ converges to $T_{f(x)}S_\alpha$. For the case of $S_\beta$, by Whitney's condition (a), the limit of $T_{g_n(y_n)}S_\beta$ contains $T_{f(x)}S_\alpha$. However, $f$ is transverse to $S_\alpha$ at $x$, meaning

$$\mathrm{d}f_x(T_xM) + T_{f(x)}S_\alpha = T_{f(x)}N. \tag{64}$$

Therefore, due to convergence, the transversality condition for $S_\alpha$ or $S_\beta$ must hold for $g_n$ at $y_n$ for sufficiently large $n$, which is a contradiction. $\square$

**Corollary D.10.** *Under the conditions of the previous proposition, assume that the stratification $S = \{S_\alpha\}$ satisfies Whitney's condition (a). If a map $f$ is transverse to a stratum $S_\alpha$ at a point $x \in f^{-1}(S_\alpha)$, then there exists a neighborhood $V_x$ of $x$ and a neighborhood $U_x$ of $f$ in $C^\infty(M, N)$ such that for any $g \in U_x$ and any $y \in V_x$, $g$ is transverse to the entire stratification $S$ in the neighborhood $V_x$.*

*Proof.* Let $x$ be a point where $f$ is transverse to $S_\alpha$. We consider all strata $S_\beta$ that satisfies $\overline{S_\beta} \cap S_\alpha \neq \varnothing$. Since a stratification is locally finite, there are only a finite number of such adjacent strata. For each such adjacent stratum $S_\beta$, the pair $(S_\alpha, S_\beta)$ satisfies Whitney's condition (a). By Prop. D.9, there exist neighborhoods $U_x^{(\alpha,\beta)}$ of $f$ and $V_x^{(\alpha,\beta)}$ of $x$ where transversality to both $S_\alpha$ and $S_\beta$ holds. We can then construct the desired neighborhoods by taking the finite intersection of these individual neighborhoods labelled by $\beta$:

$$U_x := \bigcap_{\overline{S_\beta} \cap S_\alpha \neq \varnothing} U_x^{(\alpha,\beta)} \quad \text{and} \quad V_x := \bigcap_{\overline{S_\beta} \cap S_\alpha \neq \varnothing} V_x^{(\alpha,\beta)}. \tag{65}$$

Since the intersection is finite, $U_x$ and $V_x$ are still open neighborhoods of $f$ and $x$, respectively. For any $g \in U_x$ and $y \in V_x$, $g$ is transverse to $S_\alpha$ and all its adjacent strata $S_\beta$ at $y$. Therefore, it is transverse to the entire stratification $S$ within the local neighborhood $V_x$.

$\square$

**Lemma D.11.** *For a $G$-manifold $M$ and a $G$-representation $Y$, the collection of submanifolds*

$$\{(M_{(H)} \times Y_{(H')})_{(H)}\}_{(H') \geq (H)} \tag{66}$$

*forms a Whitney stratification of $(M \times Y)_{(H)}$.*

*Proof.* By Prop. 3.7.1 of Field (2007), we obtain that these point sets in the collection are smooth $G$-submanifolds. By Lem. D.8, we have

$$(M_{(H)} \times Y_{(H')})_{(H)} = (M_{(H)} \times Y_{(H')})_H \times_{N_G(H)/H} G/H \tag{67}$$

$$= (M_{(H)} \times Y_{(H')})^H_{(H)} \times_{N_G(H)/H} G/H \tag{68}$$

$$= (M^H_{(H)} \times Y^H_{(H')}) \times_{N_G(H)/H} G/H. \tag{69}$$

By local trivialization, similar to the proof of Prop. 3.9.2 of Field (2007), it is sufficient to prove that the pair $(Y^H_{(H')}, Y^H_{(K)})$ satisfies Whitney's condition (b). By Prop. 3.9.2 of Field (2007), the orbit type stratification for a representation space is a normal orbit type stratification, and therefore the pair of strata $(Y_{(H')}, Y_{(K)})$ for $(H') > (K)$ satisfies Whitney's conditions. We now show that $(Y^H_{(H')}, Y^H_{(K)})$ also satisfies Whitney's conditions.

The condition holds for the pair $(Y_{(H')}, Y_{(K)})$. Therefore, for a point $x \in Y_{(K)} \cap \overline{Y_{(H')}}$, and for sequences $\{y_n\} \subset Y_{(K)} \to x$ and $\{x_n\} \subset Y_{(H')} \to x$ such that the secant lines $\overline{x_n y_n}$ converge to a line $l \subset T_x Y$ and the tangent spaces $T_{y_n} Y_{(K)}$ converge to a linear subspace $E \subset T_x Y$, we have $l \subseteq E$. By restricting the point $x$ to be in $Y^H_{(K)} \cap \overline{Y^H_{(H')}}$, and the sequences $\{x_n\}$ and $\{y_n\}$ to be in $Y^H_{(H')}$ and $Y^H_{(K)}$ respectively, the conclusion $l \subseteq E$ still holds. Therefore, the pair $(Y^H_{(H')}, Y^H_{(K)})$ also satisfies Whitney's condition (b). $\square$

**Lemma D.12.** *Let $M$ and $N$ be topological spaces, with $M$ being compact. Consider a continuous map $f : M \to N$. Let $C$ be a closed set in $N$, and suppose there exists a neighborhood $V$ of the preimage $f^{-1}(C)$.*

*Then there exists a neighborhood $U$ of $f$ in $C(M, N)$ with the $C^0$ topology (compact-open topology) such that for any map $g \in U$, we have $g(M \setminus V) \cap C = \varnothing$.*

*Proof.* Note that $M \setminus V$ is a compact set, and $N \setminus C$ is an open set. By the definition of the compact-open topology, e.g. Definition 43.1 of Willard (1970), the set

$$U := \{h \in C(M, N) \mid h(M \setminus V) \subseteq N \setminus C\} \tag{70}$$

is an open set.

Since $V$ is a neighborhood of $f^{-1}(C)$, we have $f^{-1}(C) \subseteq V$, which implies $f(M \setminus V) \subseteq N \setminus C$. Therefore, the map $f$ itself is an element of $U$, meaning $U$ is an open neighborhood of $f$. For any map $g \in U$, its definition directly implies that $g(M \setminus V) \subseteq N \setminus C$, which is equivalent to $g(M \setminus V) \cap C = \varnothing$. Thus, this set $U$ is the desired neighborhood. $\square$

### D.4 Generic Equivariant Mappings

**Proposition D.13** (Equivariant Smooth Extension). *Let $M$ be a smooth $G$-manifold and let $S \subset M$ be a smooth and compact $G$-submanifold. For any smooth equivariant map $f \in C^\infty_G(S, Y)$ defined on $S$ that maps into a representation space $Y$, there exists a smooth equivariant extension $\tilde{f} \in C^\infty_G(M, Y)$ such that its restriction to $S$ is $f$, i.e., $\tilde{f}|_S = f$.*

*Proof.* The proof strategy is the same as that for the Tietze-Gleason Theorem (see Chap. 1, Thm. 2.3 of Bredon (1972)).

First, we can view the map $f$ as a collection of $\dim Y$ smooth real-valued functions defined on $S$. By the standard smooth extension theorem, e.g., Lem. 5.34 in Lee (2012), there exists a smooth (but not necessarily equivariant) extension $\varphi \in C^\infty(M, Y)$ such that $\varphi|_S = f$.

We then construct an equivariant map from $\varphi$ using the group averaging operator. Define the map $\psi : M \to Y$ by

$$\psi(x) = \int_G \rho_Y(g^{-1})\varphi(\rho_X(g)(x)) \, \mathrm{d}g. \tag{71}$$

Due to the linearity of the integral and the properties of the Haar measure, this map is equivariant. For any $h \in G$:

$$\psi(h(x)) = \int_G \rho_Y(g^{-1})\varphi(g(h(x))) \, \mathrm{d}g \tag{72}$$

$$= \int_G \rho_Y((g'h^{-1})^{-1})\varphi(g'(x)) \, \mathrm{d}g' \tag{73}$$

$$= \rho_Y(h) \int_G \rho_Y((g')^{-1})\varphi(g'(x)) \, \mathrm{d}g' \tag{74}$$

$$= \rho_Y(h)\psi(x). \tag{75}$$

Next, we verify that the restriction of $\psi$ to the submanifold $S$ is equal to the original map $f$. For any $s \in S$:

$$\psi(s) = \int_G \rho_Y(g^{-1})\varphi(g(s)) \, \mathrm{d}g \tag{76}$$

$$= \int_G \rho_Y(g^{-1})f(g(s)) \, \mathrm{d}g \tag{77}$$

$$= \int_G \rho_Y(g^{-1})\rho_Y(g)f(s) \, \mathrm{d}g \tag{78}$$

$$= \int_G f(s) \, \mathrm{d}g = f(s) \int_G 1 \, \mathrm{d}g = f(s). \tag{79}$$

Finally, the smoothness of $\psi$ follows from the smoothness of $\varphi$ and the properties of integration over a compact group. This can be verified by a local coordinate analysis. Thus, $\psi$ is the desired smooth equivariant extension $\tilde{f}$. $\qquad\square$

**Corollary D.14.** *Consider maps from a $G$-manifold $X$ to a representation space $Y$, where $X$ contains a compact, smooth $G$-submanifold $M$. Let $S$ be a subset of $C_G^\infty(M, Y)$ that contains an open dense set. Then the set of maps $f \in C_G^\infty(X, Y)$ whose restriction to $M$ lies in $S$ (i.e., $f|_M \in S$) also contains an open dense subset of $C_G^\infty(X, Y)$.*

*Proof.* We recall that a set contains an open dense subset if and only if for any non-empty open set, its intersection with the set contains a non-empty open subset. Let $A := \{f \in C_G^\infty(X, Y) \mid f|_M \in S\}$. Our goal is to show that for any $f \in C_G^\infty(X, Y)$ and any of its open neighborhoods $U$, there exists a non-empty open set $V \subseteq U \cap A$.

Consider the restriction map $\mathrm{Res}_M : C_G^\infty(X, Y) \to C_G^\infty(M, Y)$, by Prop. D.13, this map is surjective. Furthermore, this map is continuous and open.

Let $U$ be an open neighborhood of an arbitrary map $f$. Since $\mathrm{Res}_M$ is an open map, its image, $\mathrm{Res}_M(U)$, is an open neighborhood of $f|_M$. By our hypothesis on $S$, its intersection with the open set $\mathrm{Res}_M(U)$ must contain a non-empty open subset. Let us call this non-empty open set $V'$. We have

$$V' \subseteq \mathrm{Res}_M(U) \cap S. \tag{80}$$

Now, let's consider the preimage $V := \mathrm{Res}_M^{-1}(V')$. Since $\mathrm{Res}_M$ is continuous and $V'$ is open, $V$ is an open set. Since $\mathrm{Res}_M$ is surjective, the preimage $V$ must also be non-empty. Therefore $U \cap V$ is a non-empty open subset of $U \cap A$, which proves the corollary. $\qquad\square$

For $C^r$ manifolds $X$ and $Y$, $C^r(X)$ is the set of $C^r$ real-valued functions on $X$, and $C^r(X, Y)$ is the set of $C^r$ maps from $X$ to $Y$. For manifolds with a $C^r$ $G$-action, $C_G^r(X, Y)$ is the set of $C^r$ equivariant maps from $X$ to $Y$. We assume these function spaces are endowed with the $C^r$ topology; in our proofs, we always consider the $C^\infty$ topology. For a map $f \in C^1(X, Y)$ and a submanifold $A \subseteq Y$, $f \pitchfork A$ denotes that $f$ is transverse to $A$. For $G$-manifolds, $f \pitchfork_G A$ denotes that $f$ is in

equivariant general position with respect to the $G$-submanifold $A$ defined in Bierstone (1977). For $f \in C^r(X, Y)$, the jet map $j^r f : X \to J^r(X, Y)$ maps a point $x$ to the equivalence class of the first $r$ derivatives of $f$ at $x$. We obtain the following results.

**Proposition D.15** (Bierstone (1977), Thm. 1.3). *Let $M$ and $N$ be smooth $G$-manifolds. If $P$ is a closed $G$-submanifold of $N$ and $K$ is a compact subset of $M$, then the set of maps $f \in C_G^\infty(M, N)$ satisfying $f \pitchfork_G P$ on $K$ forms an open subset of $C_G^\infty(M, N)$ (in the Whitney $C^\infty$ topology).*

**Proposition D.16** (Bierstone (1977), Thm. 1.4). *Let $M, N$ be smooth $G$-manifolds and $P$ be a $G$-submanifold of $N$. The set of maps $f \in C_G^\infty(M, N)$ satisfying $f \pitchfork_G P$ forms a residual subset of $C_G^\infty(M, N)$, i.e., a countable intersection of open dense sets (in the Whitney $C^\infty$ topology).*

*Remark.* For compact $M$, note that the Whitney $C^\infty$ topology coincides with the $C^\infty$ topology.

**Proposition D.17** (Stratumwise Transversality Theorem). *For any orbit type $(H) \in \mathcal{O}_G(M)$, a map $f \in C_G^\infty(M, N)$ with $f \pitchfork_G P$ satisfies the stratumwise transversality property:*

$$f|_{M_H} : M_H \mapsto N^H, f|_{M_H} \pitchfork P^H. \tag{81}$$

*Alternatively, this can be expressed in the language of jets as*

$$j^0 f|_{M_{(H)}} : M_{(H)} \mapsto (M_{(H)} \times N)_{(H)}, j^0 f|_{M_{(H)}} \pitchfork (M_{(H)} \times P)_{(H)} \tag{82}$$

*Remark.* The fact that $f \pitchfork_G P$ implies stratumwise transversality on the fixed-point sets is a fundamental conclusion derived from Prop. 6.4 of Bierstone (1977). Our proof below is a straightforward corollary of this.

*Proof.* From the discussion in Prop. 6.4 of Bierstone (1977), we have

$$j^0(f|_{M_{(H)}}) : M_{(H)} \mapsto (M \times N)_{(H)}, \quad j^0(f|_{M_{(H)}}) \pitchfork (M \times P)_{(H)}. \tag{83}$$

Note that the image of the map $j^0(f|_{M_{(H)}})$ is contained within $M_{(H)} \times N$. The transversality condition is an equality of tangent spaces:

$$\mathrm{d}(j^0(f|_{M_{(H)}}))_x(T_x M_{(H)}) + T_y((M \times P)_{(H)}) = T_y((M \times N)_{(H)}). \tag{84}$$

Intersecting both sides of this equation with $T_y((M_{(H)} \times N)_{(H)})$ yields

$$\mathrm{d}(j^0(f|_{M_{(H)}}))_x(T_x M_{(H)}) + T_y((M_{(H)} \times P)_{(H)}) = T_y((M_{(H)} \times N)_{(H)}). \tag{85}$$

Therefore, we have

$$j^0(f|_{M_{(H)}}) : M_{(H)} \to (M_{(H)} \times N)_{(H)}, \quad j^0(f|_{M_{(H)}}) \pitchfork (M_{(H)} \times P)_{(H)}. \tag{86}$$

$\square$

**Proposition D.18** (Bierstone (1977), Sec. 7). *Consider smooth $G$-manifolds $M$ and $N$. Let $P$ be a smooth $G$-submanifold of $N$. If an equivariant map $f$ satisfies $f \pitchfork_G P$ at a point $x \in f^{-1}(P)$, then there exists a neighborhood $U$ of $f$ in $C_G^\infty(M, N)$ and a $G$-invariant neighborhood $V$ of the orbit $G(x)$ such that for any map $g \in U$ and any point $y \in V$, it holds that $g \pitchfork_G P$ at $y$.*

*Remark.* The proposition above can also be derived from the properties of stratifications given in Prop. D.9 by the definition of equivariant general position in Bierstone (1977).

**Proposition D.19.** *Let $G$ be a compact Lie group, $M$ be a compact, smooth $G$-manifold, and $Y$ be a $G$-representation space. The set of smooth equivariant maps $f \in C_G^\infty(M, Y)$ that satisfy the transversality condition*

$$j^0(f|_{M_{(H)}}) \pitchfork (M_{(H)} \times Y_{(H')})_{(H)} \tag{87}$$

*for all pairs of orbit types $(H)$ and $(H')$ such that $(H)$ is present in $M$ and $(H') \geq (H)$, contains an open dense subset of $C_G^\infty(M, Y)$.*

*Remark.* The proof of density is straightforward. However, the openness of this property does not generally hold. A counterexample can be constructed following Ex. 2.1 of Bierstone (1977).

*Proof.* Since $M$ is compact, by Proposition 3.7.2 of Field (2007), the orbit types are finite. For any given symmetry type $(H)$, by Prop. D.16 the set of maps satisfying $f \pitchfork_G Y_{(H)}$ is an intersection of a finite number of residual sets, and is therefore itself a residual set. By Prop. D.17 the set of maps satisfying the transversality condition stated in the theorem is dense.

Instead of directly proving openness property, we prove a related proposition: for a map $f$ that satisfies $f \pitchfork_G Y_{(H)}$ for all orbit types $(H)$, there exists a neighborhood $U$ of $f$ such that for any $g \in U$,

$$j^0(g|_{M_{(H)}}) \pitchfork (M_{(H)} \times Y_{(H')})_{(H)} \quad \text{for all } (H') \geq (H). \tag{88}$$

The proof proceeds by induction on the dimension of the strata in the orbit type stratification of $Y$, ordered from lowest to highest (or equivalently, from highest to lowest symmetry). We discuss the fixed orbit type $(H)$ in $M$. Then we can construct neighborhoods that hold for all $(H)$ by taking the intersection of the neighborhoods corresponding to each orbit type $(H)$.

We start with the lowest-dimensional strata. Let $Y_{(H_1)}$ be a stratum of minimal dimension. Such a stratum is unique, which corresponds to a maximal symmetry type $(G)$ and is a closed $G$-submanifold $Y^G$. By Prop. D.15, the set of maps in general position to $Y_{(H_1)}$ is open in $C_G^\infty(M, Y)$. Thus, there exists a neighborhood $U_1$ of $f$ such that for any $g \in U_1$, $g \pitchfork_G Y_{(H_1)}$. By Prop. D.17, this implies that for all $(H)$,

$$j^0(g|_{M_{(H)}}) \pitchfork (M_{(H)} \times Y_{(H_1)})_{(H)}. \tag{89}$$

Assume the proposition holds for all strata of dimension up to $k-1$. Let $U_{k-1}$ be the neighborhood of $f$ found from the inductive hypothesis. For any map $g \in U_{k-1}$ and for any stratum $Y_{(H_i)}$ with $\dim Y_{(H_i)} \leq k-1$, we have

$$j^0(g|_{M_{(H)}}) \pitchfork (M_{(H)} \times Y_{(H_i)})_{(H)} \quad \text{for all } (H_i) \geq (H). \tag{90}$$

Note that by Lem. D.11 the collection $S_{(H)} = \{(M_{(H)} \times Y_{(H')})_{(H)}\}_{(H') \geq (H)}$ is a Whitney stratification and satisfies the frontier condition. Therefore, by Cor. D.10, for any $x \in M_{(H)}$ with $f(x) \in Y_{(H_i)}$, there exists a neighborhood $V_x$ of $x$ and a neighborhood $U_x$ of $j^0(f|_{M_{(H)}})$ such that for any map in $U_x$, it is transverse to the stratification $S_{(H)}$.

Next, we need to obtain an open set in $C_G^\infty(M, Y)$. We use the following sequence of maps between function spaces:

$$C_G^\infty(M, Y) \overset{\mathrm{Res}_{M_{(H)}}}{\mapsto} C_G^\infty(M_{(H)}, Y) \tag{91}$$

$$\overset{j^0}{\mapsto} C_G^\infty(M_{(H)}, (M_{(H)} \times Y)_{(H)}) \tag{92}$$

$$\hookrightarrow C^\infty(M_{(H)}, (M_{(H)} \times Y)_{(H)}). \tag{93}$$

The maps between these function spaces are continuous in $C^\infty$ topology. For example, the 0-jet map is a restriction of the continuous 0-jet map on the general function space, which can be obtained from the continuity of jet map in the Whitney $C^\infty$ topology established in Chap. 2, Prop. 3.4 of Golubitsky & Guillemin (1973). Since the topology on the equivariant function space is induced from the general function space, the restricted map is also continuous. Similarly, $\mathrm{Res}_{M_{(H)}}$ and the inclusion are continuous. From this analysis, we can pull back the neighborhood $U_x$ to obtain a neighborhood $U_x'$ of $f$ in $C_G^\infty(M, Y)$ where the transversality holds.

We deal with the global result. For any $i$ with $\dim Y_{(H_i)} \leq k-1$, the collection of neighborhoods $\{V_x\}_{x \in f^{-1}(Y_{(H_i)})}$ for all $i$ with $\dim Y_{(H_i)} \leq k-1$ forms an open cover of $f^{-1}(Y_{(H_i)})$. By the frontier condition, any stratum that intersects the closure $\overline{Y_{(H_i)}}$ has strictly smaller dimension than $Y_{(H_i)}$. Hence $\bigcup_i Y_{(H_i)}$ is closed. Since $M$ is compact and $f$ is continuous, the preimage $f^{-1}(\bigcup_i Y_{(H_i)})$ is compact, so there exists a finite subcover $\{V_{x_n}\}_{n=1}^N$ of the preimage. We construct the neighborhood $V_k := \bigcup_{n=1}^N V_{x_n}$ of the preimage in $M$, and the neighborhood $U_k^{(1)} := \bigcap_{n=1}^N U_{x_n}'$ of $f$ in the function space.

In the $C^0$ topology, by Lem. D.12, there exists a neighborhood $U_k^{(2)}$ of $f$ in $C(M, Y)$ such that for any $g \in U_k^{(2)}$,

$$g(M \setminus V_k) \cap \bigcup_i Y_{(H_i)} = \varnothing. \tag{94}$$

By pulling this back through the inclusions $C^\infty_G(M,Y) \hookrightarrow C^\infty(M,Y) \hookrightarrow C(M,Y)$, we obtain a neighborhood $(U^{(2)}_k)'$ in $C^\infty_G(M,Y)$. The neighborhood is open in the $C^0$ topology, so it also open in the $C^\infty$ topology. Let $U'_k = U_{k-1} \cap U^{(1)}_k \cap (U^{(2)}_k)'$. For any map $g \in U'_k$, we have

$$\begin{cases} j^0(g|_{M_{(H)}}) \pitchfork S_{(H)} & \text{for } x \in V_k \\ g(x) \notin Y_{(H_i)} & \text{for } x \in M \setminus V_k \end{cases} \tag{95}$$

for all $i$ with $\dim Y_{(H_i)} \leq k-1$.

Now we show by contradiction that there exists a neighborhood $U_k \subset U'_k$ of $f$ where the transversality condition holds for strata of dimension $k$. By the induction hypothesis, the transversality condition holds for all strata of dimension at most $k-1$. Therefore, it also holds for all strata of dimension at most $k$. The proof idea is from the proof of openness of equivariant general position in Sec. 7 of Bierstone (1977), where the closedness of $P$ is replaced by the condition that $f$ does not intersect low-dimensional $Y_{(H_i)}$ outside of $V_k$.

Assume openness does not hold. Then there exists a sequence $\{g_n\} \to f$ in $U'_k$ such that each $g_n$ fails the stratumwise transversality condition for some stratum of dimension $k$. Since the stratumwise transversality condition fails, it follows from Prop. D.17 that $g_n \pitchfork_G Y_{(H_j)}$ for all $j$ with $\dim Y_{(H_j)} = k$ also does not hold. The points of non-transversality for these maps, $\{y_n\}$, can only occur in $M \setminus V_k$ and $g_n(y_Y) \in \bigcup_j Y_{(H_j)}$. Since $M \setminus V_k$ is compact, there is a convergent subsequence $\{y_{n_i}\}$ with limit $x$. Then $g_n(y_Y) \to f(x)$. By construction of $U'_k$, $f(M \setminus V_k)$ does not intersect $Y_{(H_i)}$ for any $\dim Y_{(H_i)} \leq k-1$. Thus, we must have $f(x) \in Y_{(H_j)}$ for some $j$.

We claim that $g_{n_i}(y_{n_i}) \in Y_{(H_j)}$ for all sufficiently large $i$. Otherwise, there exists a further subsequence $\{g_{n_{i_\alpha}}(y_{n_{i_\alpha}})\}$ such that $g_{n_{i_\alpha}}(y_{n_{i_\alpha}}) \in Y_{(H_{j'})}$ for some $j' \neq j$ and all $\alpha$, while still $g_{n_{i_\alpha}}(y_{n_{i_\alpha}}) \to f(x) \in Y_{(H_j)}$. Hence $Y_{(H_j)} \cap \overline{Y_{(H_{j'})}} \neq \varnothing$. By the frontier condition of stratification this implies $\dim Y_{(H_j)} < \dim Y_{(H_{j'})}$. However, $Y_{(H_j)}$ and $Y_{(H_{j'})}$ have the same dimension, yielding a contradiction. Therefore, for $i$ sufficiently large, $g_{n_i}(y_{n_i}) \in Y_{(H_j)}$.

We now show that this contradicts the assumption that $f \pitchfork_G Y_{(H_j)}$ at $x$. If $f$ were to satisfy $f \pitchfork_G Y_{(H_j)}$ at $x$, then by Prop. D.18, there would exist a neighborhood $U$ of $f$ and a $G$-invariant neighborhood $V$ of $G(x)$ such that any $g \in U$ satisfies $g \pitchfork_G Y_{(H_j)}$ at any $y \in V$. This contradicts the existence of sequence $\{g_n\}$ and points $\{y_n\}$ where transversality fails. This completes the proof. $\square$

**Proposition D.20.** *Let $\mathcal{F}$ be a $C^\infty$-dense family of smooth parameterized maps in $C^\infty_G(X,Y)$ and $\{M_j\}$ be a finite collection of compact, connected and smooth $G$-submanifolds of $X$. Let*

$$S_{(H)\to(H')}(f) = \{x \in X \mid (G_x) = (H), (G_{f(x)}) = (H')\}. \tag{96}$$

*There is a $C^\infty$-dense subset $\mathcal{G} \subset \mathcal{F}$, for $g \in \mathcal{G}$, the set $S_{(H)\to(H')}(g|_{M_j})$ is a disjoint union of smooth $G$-submanifolds of $X$, and its dimension satisfies*

$$\dim S_{(H)\to(H')}(g|_{M_j}) = \dim(M_j)_{(H)} - (\dim Y^H - \dim Y^H_{(H')}) \tag{97}$$

*if the right-hand side of the equation is not smaller than $\dim G - \dim H$. Otherwise, the set $S_{(H)\to(H')}(g|_{M_j})$ is empty.*

*Proof.* By Prop. D.19, for each $M_j$, there exists an open dense set in $C^\infty_G(M_j,Y)$ such that for any map $g$ in this set,

$$j^0(g|_{(M_j)_{(H)}}) \pitchfork ((M_j)_{(H)} \times Y_{(H')})_{(H)} \quad \text{for all } (H') \geq (H). \tag{98}$$

Therefore, writing $\mathrm{codim}_X Y := \dim X - \dim Y$ for the codimension of $Y$ in $X$, the dimension theorem for transverse maps (see, e.g., Sec. 2.3 of Arnold et al. (2012)) yields

$$\mathrm{codim}_{(M_j)_{(H)}} S_{(H)\to(H')} = \mathrm{codim}_{((M_j)_{(H)} \times Y)_{(H)}} \big((M_j)_{(H)} \times Y_{(H')}\big)_{(H)}. \tag{99}$$

Furthermore, from the proof of Lem. D.11, we have

$$((M_j)_{(H)} \times Y_{(H')})_{(H)} = \big((M_j)^H_{(H)} \times Y^H_{(H')}\big) \times_{N_G(H)/H} G/H. \tag{100}$$

Thus, we obtain

$$\text{codim}_{(M_j)_{(H)}} S_{(H)\to(H')} = \text{codim}_{Y^H} Y^H_{(H')}. \tag{101}$$

Moreover, since the dimension of an orbit with orbit type $(H)$ in a $G$-manifold is $\dim G - \dim H$, if the dimension of $S_{(H)\to(H')}$ calculated above is less than $\dim G - \dim H$, then $S_{(H)\to(H')}$ is an empty set.

For each $M_j$, we take the corresponding open dense set $U_j \subset C_G^\infty(M_j, Y)$ on which the maps satisfy the dimension theorem. Then, by Cor. D.14, the set of functions $f \in C_G^\infty(X, Y)$ that satisfy $f|_{M_j} \in U_j$ contains an open dense set $U_j'$. Since $\{M_j\}$ is a finite collection, the intersection $U = \bigcap_j U_j'$ is also an open dense set. Therefore, the intersection of $U$ with the $C^\infty$-dense set $\mathcal{F}$ is a dense set $\mathcal{G}$. Thus, this intersection is the set of maps in $\mathcal{F}$ that we sought to construct, which is dense and whose elements satisfy the dimension theorem. $\qquad\square$

## D.5   PROOF OF THM. 5.2

**Theorem 5.2.** *Let $\mathcal{F}$ be a equivariant parametrization with $C^\infty$ approximation capability. If for every $(H) \in \mathcal{O}_G(M)$ we have $(p_Y(H)) \in \mathcal{O}_G(Y)$, then for any finite union of compact, smooth $G$-submanifolds $M \subset X$, any $f \in C_G^\infty(X, Y)$, any integer $r \geq 0$, and any $\epsilon > 0$, there exists a map $g \in \mathcal{F}$ such that*

$$\max_{x \in M} \|D^k f(x) - D^k g(x)\| < \epsilon, k \leq r, \tag{11}$$

*and $g|_M$ is almost isovariant relative to $Y$. Furthermore, if the feature space $Y$ contains a representation $\tilde{Y}^{\oplus r}$ for an integer $r > \max_j\{\dim M_j\}$, where $\tilde{Y}$ itself satisfies the condition $(p_{\tilde{Y}}(H)) \in \mathcal{O}_G(\tilde{Y})$, then the approximating map $g|_M$ can be chosen to be isovariant relative to $Y$.*

*Proof of Thm. 5.2.* By Lem. D.8, we consider the relation

$$\dim N_G(H, H') - \dim N_G(H) = \alpha(H, H') = \dim Y^H_{(H')} - \dim Y^{H'}. \tag{102}$$

Regarding the $G$-representation as a faithful representation of $G/\ker\rho_Y$, for orbit types satisfying $(p_Y(H')) > I(H, Y)$, the dense and open property of the minimal orbit type by Prop. B.4 in the fixed-point space implies that for each $M_j$,

$$\dim Y^H - \dim Y^H_{(H')} = \dim Y^H - \dim Y^{H'} - \alpha_G(H, H') > 0. \tag{103}$$

By Prop. D.20, it implies

$$\dim(M_j)_{(H)} > \dim S_{(H)\to(H')}(g|_{M_j}). \tag{104}$$

With respect to the Hausdorff measure $\mathcal{H}^d$, since the dimension of $S_{(H)\to(H')}(g|_{M_j})$ is strictly less than the dimension of the manifold it lies in, its measure is zero. Then, by the finiteness of the set of orbit types for a compact manifold and the finiteness of the collection $\{M_j\}$, it follows that $g|_M$ is an almost isovariant map relative to $Y$.

Furthermore, if we require $g$ to be isovariant relative to $Y$, we need

$$S_{(H)\to(H')}(g|_{M_j}) = \varnothing. \tag{105}$$

The condition for this set to be empty is related to its codimension. Since the inequality

$$\dim Y^H - \dim Y^{H'} - \alpha_G(H, H') \geq 1 \tag{106}$$

scales with multiplicity $r$ to become

$$r(\dim Y^H - \dim Y^{H'}) - \alpha_G(H, H') \geq r, \tag{107}$$

it is sufficient to choose the multiplicity for the representation $Y^{\oplus r}$ such that $r > \max_j\{\dim M_j\}$. $\qquad\square$

# E TABLES

All closed subgroups of $O(3)$ are denoted using Schoenflies notation, where it should be noted that $K = SO(3)$ and $K_h = O(3)$. When interpreting the tables, care should be taken to recognize the low-dimensional equivalences: $C_s = C_{1h} = C_{1v}$, $D_1 = C_2$, $D_{1h} = C_{2v}$, and $D_{1d} = C_{2h}$. In the tables of symmetry infimum, we list a representative subgroup from each conjugacy class and omit the class notation (e.g., $C_2$ instead of $(C_2)$) for clarity.

## E.1 MINIMAL PROPER SUPERGROUPS IN $SO(3)$ OR $O(3)$

Minimal proper supergroups table of $SO(3)$ or $O(3)$. Some of the results can be found from Fig. 3.2.1.6 of Aroyo (2016). We only present the results for $O(3)$. The discussion for $SO(3)$ can be obtained by removing the subgroups that are not subgroups of $SO(3)$ (i.e., the subgroups of the first kind), namely $C_k, D_k, T, O, I, C_\infty, D_\infty$, and $K$.

Table 6: Table of minimal proper supergroups $H$ of axial closed subgroups of $O(3)$. In the supergroup notation, $p$ denotes a prime number and $p^*$ denotes an odd prime number.

| $G$ | $H$ for General $k$ | $H$ for Special $k$ |
|---|---|---|
| $C_k$ | $C_{pk}, S_{2k}, C_{kh}, C_{kv}, D_k$ | $D_{p^*}$ $(k=2)$; $T$ $(k=3)$ |
| $S_{2k}$ | $S_{2p^*k}, C_{2k,h}, D_{kd}$ | $T_h$ $(k=3)$ |
| $C_{kh}$ | $C_{pk,h}, D_{kh}$ | $C_{pv}(k=1); D_{p^*d}$ $(k=2)$ |
| $C_{kv}$ $(k>1)$ | $C_{pk,v}, D_{kh}, D_{kd}$ | $T_d$ $(k=3); D_{ph}$ $(k=2)$ |
| $D_k$ $(k>1)$ | $D_{pk}, D_{kh}, D_{kd}$ | $T$ $(k=2); O$ $(k=3,4); I$ $(k=3,5)$ |
| $D_{kh}$ $(k>1)$ | $D_{pk,h}$ | $T_h$ $(k=2); O_h$ $(k=4)$ |
| $D_{kd}$ $(k>1)$ | $D_{p^*k,d}, D_{2k,h}$ | $T_d$ $(k=2)$; $O_h$ $(k=3)$; $I_h$ $(k=3,5)$ |

Table 7: Table of minimal proper supergroups $H$ of other closed subgroups $O(3)$.

| $G$ | $H$ |
|---|---|
| $T$ | $T_d, T_h, O, I$ |
| $T_d$ | $O_h$ |
| $T_h$ | $O_h, I_h$ |
| $O$ | $O_h, K$ |
| $O_h$ | $K_h$ |
| $I$ | $I_h, K$ |
| $I_h$ | $K_h$ |
| $C_\infty$ | $C_{\infty h}, C_{\infty,v}, D_\infty$ |
| $C_{\infty h}$ | $D_{\infty,h}$ |
| $C_{\infty v}$ | $D_{\infty,h}$ |
| $D_\infty$ | $D_{\infty,h}, K$ |
| $D_{\infty h}$ | $K_h$ |
| $K$ | $K_h$ |

### E.2 Dimensions of Fixed-point Subspaces for Subgroups of $SO(3)$ or $O(3)$

Dimensions table of fixed-point subspaces for subgroups of $SO(3)$ or $O(3)$. Some of the results can be found from Table B.1 and Table B.2 of Linehan & Stedman (2001). We only present the results for $O(3)$. The discussion for $SO(3)$ can be obtained directly from the table.

Table 8: Dimensions of fixed-point subspaces for closed subgroups of $O(3)$ acting on the irreducible representations $V_{l=l_0^\pm}$, where $a_k(l) = \lfloor l/k \rfloor$ and $b_k(l) = \lfloor (l+k)/(2k) \rfloor$.

| Subgroup | | $l = l_0^-$ | | $l = l_0^+$ | |
|---|---|---|---|---|---|
| | | $l_0$ even | $l_0$ odd | $l_0$ even | $l_0$ odd |
| $C_k$ | | \multicolumn{4}{c}{$2a_k(l_0) + 1$} | | | |
| $S_{2k}$ | $k$ even | $2b_k(l_0)$ | | $2a_{2k}(l_0) + 1$ | |
| | $k$ odd | $0$ | | $2a_k(l_0) + 1$ | |
| $C_{kh}$ | $k$ even | $0$ | | $2a_k(l_0) + 1$ | |
| | $k$ odd | $2b_k(l_0)$ | | $2a_{2k}(l_0) + 1$ | |
| $C_{kv}$ | | $a_k(l_0)$ | $a_k(l_0) + 1$ | $a_k(l_0) + 1$ | $a_k(l_0)$ |
| $D_k$ | | $a_k(l_0) + 1$ | $a_k(l_0)$ | $a_k(l_0) + 1$ | $a_k(l_0)$ |
| $D_{kh}$ | $k$ even | $0$ | | $a_k(l_0) + 1$ | $a_k(l_0)$ |
| | $k$ odd | $b_k(l_0)$ | | $a_{2k}(l_0) + 1$ | $a_{2k}(l_0)$ |
| $D_{kd}$ | $k$ even | $b_k(l_0)$ | | $a_{2k}(l_0) + 1$ | $a_{2k}(l_0)$ |
| | $k$ odd | $0$ | | $a_k(l_0) + 1$ | $a_k(l_0)$ |
| $T$ | | \multicolumn{4}{c}{$2a_3(l_0) + a_2(l_0) - l_0 + 1$} | | | |
| $T_h$ | | $0$ | | $2a_3(l_0) + a_2(l_0) - l_0 + 1$ | |
| $T_d$ | | $a_3(l_0) + b_2(l_0) + b_1(l_0) - l_0$ | | $a_4(l_0) + a_3(l_0) + a_2(l_0) - l_0 + 1$ | |
| $O$ | | \multicolumn{4}{c}{$a_4(l_0) + a_3(l_0) + a_2(l_0) - l_0 + 1$} | | | |
| $O_h$ | | $0$ | | $a_4(l_0) + a_3(l_0) + a_2(l_0) - l_0 + 1$ | |
| $I$ | | \multicolumn{4}{c}{$a_5(l_0) + a_3(l_0) + a_2(l_0) - l_0 + 1$} | | | |
| $I_h$ | | $0$ | | $a_5(l_0) + a_3(l_0) + a_2(l_0) - l_0 + 1$ | |
| $C_\infty$ | | \multicolumn{4}{c}{$1$} | | | |
| $C_{\infty h}$ | | $0$ | | $1$ | |
| $C_{\infty v}$ | | $0$ | $1$ | $1$ | $0$ |
| $D_\infty$ | | $1$ | $0$ | $1$ | $0$ |
| $D_{\infty h}$ | | $0$ | | $1$ | $0$ |

### E.3 SYMMETRY INFIMUM FOR SUBGROUPS OF $SO(3)$

Symmetry infimum table for subgroups of $SO(3)$. In the following tables, we use a color-coding scheme to classify the types of symmetry increase. An increase to the full group is termed **full degeneration** and is marked in red . An increase to a supergroup of a strictly higher dimension is termed **continuous degeneration** and is marked in blue . An increase to a supergroup of the same dimension is termed **discrete degeneration**, and is marked in yellow . No increase is termed **no degeneration**, and is marked in green . All subgroups increase to to $K$ when $l = 0$.

Table 9: Symmetry infimum of general axis subgroup of $SO(3)$ on $V_{l=l_0}^{\oplus r}$, $r > 3$, $l_0 > 0$.

|  | $l_0 < k$ | | $l_0 \geq k$ |
|---|---|---|---|
|  | $l_0$ even | $l_0$ odd | |
| $C_k$ | $D_\infty$ | $C_\infty$ | $C_k$ |
| $D_k(k > 2)$ | $D_\infty$ | $K$ | $D_k$ |

Table 10: Symmetry infimum of special axis subgroup of $SO(3)$ on $V_{l=l_0}^{\oplus r}$, $r > 3$, $l_0 > 0$.

|  | $l_0 = 1$ | $l_0 = 2$ | $l_0 = 3$ | $l_0 \geq 4$ |
|---|---|---|---|---|
| $D_2$ | $K$ | $D_2$ | $T$ | $D_2$ |

Table 11: Symmetry infimum of polyhedral subgroup of $SO(3)$ on $V_{l=l_0}^{\oplus r}$, $r > 3$, $l_0 > 0$. The "Caption" in the table takes the value $l_0 = 6, 10, 12, 15, 16, 18, 20 - 22, 24 - 28$.

| $T$ | $l_0 = 6, 9, 10$ | $l_0 = 3, 7$ | $l_0 = 4, 8$ | $l_0 \geq 12$ | other |
|---|---|---|---|---|---|
|  | $T$ | $T$ | $O$ | $T$ | $K$ |
| $O$ | $l_0 = 4, 6, 8, 9, 10$ | $l_0 \geq 12$ | other | | |
|  | $O$ | $O$ | $K$ | | |
| $I$ | Caption | $l_0 \geq 30$ | other | | |
|  | $I$ | $I$ | $K$ | | |

Table 12: Symmetry infimum of infinite subgroup of $SO(3)$ on $V_{l=l_0}^{\oplus r}$, $r > 3$, $l_0 > 0$.

|  | $l_0$ even | $l_0$ odd |
|---|---|---|
| $C_\infty$ | $D_{\infty h}$ | $C_{\infty h}$ |
| $D_\infty$ | $D_{\infty h}$ | $K_h$ |
| $K$ | $K$ | $K$ |

### E.4 SYMMETRY INFIMUM FOR SUBGROUPS OF $O(3)$

Symmetry Infimum table for Subgroups of $O(3)$. The color scheme for full, continuous, and no degeneration is the same as for $SO(3)$. For cases of **discrete degeneration** of $l_0 = l^+$, we distinguish between (a) light Green , which indicates the predictable increase to $\pi^{-1}(\pi(H))$ for the projection map $\pi : O(3) \to SO(3)$, and (b) yellow , which indicates all other exceptional cases of discrete degeneration. In the following table, the first type of subgroups degenerate to $K$ and the others to $K_h$ when $l_0 = 0^-$. All subgroups increase to to $K_h$ when $l_0 = 0^+$.

Table 13: Symmetry infimum of general axis subgroup of $O(3)$ on $V_{l=l_0^-}^{\oplus r}$, $r > 3$, $l_0 > 0$.

| Subgroup | | $0 < l_0 < k$ | | $k \leq l_0 < 2k$ | | $l_0 \geq 2k$ | |
|---|---|---|---|---|---|---|---|
| | | $l_0$ even | $l_0$ odd | $l_0$ even | $l_0$ odd | $l_0$ even | $l_0$ odd |
| $C_k$ | | $D_\infty$ | $C_{\infty v}$ | $C_k$ | $C_k$ | $C_k$ | $C_k$ |
| $S_{2k}$ | $k$ even | $K_h$ | $K_h$ | $S_{2k}$ | $S_{2k}$ | $S_{2k}$ | $S_{2k}$ |
| | $k$ odd | $K_h$ | $K_h$ | $K_h$ | $K_h$ | $K_h$ | $K_h$ |
| $C_{kh}$ | $k$ even | $K_h$ | $K_h$ | $K_h$ | $K_h$ | $K_h$ | $K_h$ |
| | $k$ odd | $K_h$ | $K_h$ | $C_{kh}$ | $C_{kh}$ | $C_{kh}$ | $C_{kh}$ |
| $C_{kv}(k>2)$ | $k$ even | $K_h$ | $C_{\infty v}$ | $D_{kd}$ | $C_{kv}$ | $C_{kv}$ | $C_{kv}$ |
| | $k$ odd | $K_h$ | $C_{\infty v}$ | $D_{kh}$ | $C_{kv}$ | $C_{kv}$ | $C_{kv}$ |
| $D_k(k>2)$ | $k$ even | $D_\infty$ | $K_h$ | $D_k$ | $D_{kd}$ | $D_k$ | $D_k$ |
| | $k$ odd | $D_\infty$ | $K_h$ | $D_k$ | $D_{kh}$ | $D_k$ | $D_k$ |
| $D_{kh}(k>2)$ | $k$ even | $K_h$ | $K_h$ | $K_h$ | $K_h$ | $K_h$ | $K_h$ |
| | $k$ odd | $K_h$ | $K_h$ | $D_{kh}$ | $D_{kh}$ | $D_{kh}$ | $D_{kh}$ |
| $D_{kd}(k>2)$ | $k$ even | $K_h$ | $K_h$ | $D_{kd}$ | $D_{kd}$ | $D_{kd}$ | $D_{kd}$ |
| | $k$ odd | $K_h$ | $K_h$ | $K_h$ | $K_h$ | $K_h$ | $K_h$ |

Table 14: Symmetry infimum of general axis subgroup of $O(3)$ on $V_{l=l_0^+}^{\oplus r}$, $r > 3$, $l_0 > 0$.

| Subgroup | | $0 < l_0 < k$ | | $k \leq l_0 < 2k$ | | $l_0 \geq 2k$ | |
|---|---|---|---|---|---|---|---|
| | | $l_0$ even | $l_0$ odd | $l_0$ even | $l_0$ odd | $l_0$ even | $l_0$ odd |
| $C_k$ | $k$ even | $D_{\infty h}$ | $C_{\infty h}$ | $C_{kh}$ | $C_{kh}$ | $C_{kh}$ | $C_{kh}$ |
| | $k$ odd | $D_{\infty h}$ | $C_{\infty h}$ | $S_{2k}$ | $S_{2k}$ | $S_{2k}$ | $S_{2k}$ |
| $S_{2k}$ | $k$ even | $D_{\infty h}$ | $C_{\infty h}$ | $D_{\infty h}$ | $C_{\infty h}$ | $C_{2kh}$ | $C_{2kh}$ |
| | $k$ odd | $D_{\infty h}$ | $C_{\infty h}$ | $S_{2k}$ | $S_{2k}$ | $S_{2k}$ | $S_{2k}$ |
| $C_{kh}$ | $k$ even | $D_{\infty h}$ | $C_{\infty h}$ | $C_{kh}$ | $C_{kh}$ | $C_{kh}$ | $C_{kh}$ |
| | $k$ odd | $D_{\infty h}$ | $C_{\infty h}$ | $D_{\infty h}$ | $C_{\infty h}$ | $C_{2kh}$ | $C_{2kh}$ |
| $C_{kv}(k>2)$ | $k$ even | $D_{\infty h}$ | $K_h$ | $D_{kh}$ | $D_{kh}$ | $D_{kh}$ | $D_{kh}$ |
| | $k$ odd | $D_{\infty h}$ | $K_h$ | $D_{kd}$ | $D_{kd}$ | $D_{kd}$ | $D_{kd}$ |
| $D_k(k>2)$ | $k$ even | $D_{\infty h}$ | $K_h$ | $D_{kh}$ | $D_{kh}$ | $D_{kh}$ | $D_{kh}$ |
| | $k$ odd | $D_{\infty h}$ | $K_h$ | $D_{kd}$ | $D_{kd}$ | $D_{kd}$ | $D_{kd}$ |
| $D_{kh}(k>2)$ | $k$ even | $D_{\infty h}$ | $K_h$ | $D_{kh}$ | $D_{kh}$ | $D_{kh}$ | $D_{kh}$ |
| | $k$ odd | $D_{\infty h}$ | $K_h$ | $D_{\infty h}$ | $K_h$ | $D_{2kh}$ | $D_{2kh}$ |
| $D_{kd}(k>2)$ | $k$ even | $D_{\infty h}$ | $K_h$ | $D_{\infty h}$ | $K_h$ | $D_{2kh}$ | $D_{2kh}$ |
| | $k$ odd | $D_{\infty h}$ | $K_h$ | $D_{kd}$ | $D_{kd}$ | $D_{kd}$ | $D_{kd}$ |

Table 15: Symmetry infimum of special axis subgroup of $O(3)$ on $V_{l=l_0^-}^{\oplus r}$, $r > 3$, $l_0 > 0$.

|          | $l_0 = 1$      | $l_0 = 2$ | $l_0 = 3$ | $l_0 \geq 4$ | |
|----------|----------------|-----------|-----------|--------------|--------------|
|          |                |           |           | $l_0$ even   | $l_0$ odd    |
| $C_{2v}$ | $C_{\infty v}$ | $D_{2d}$  | $C_{2v}$  | $C_{2v}$     | $C_{2v}$     |
| $D_2$    | $K_h$          | $D_2$     | $T_d$     | $D_2$        | $D_2$        |
| $D_{2h}$ | $K_h$          | $K_h$     | $K_h$     | $K_h$        | $K_h$        |
| $D_{2d}$ | $K_h$          | $D_{2d}$  | $T_d$     | $D_{2d}$     | $D_{2d}$     |

Table 16: Symmetry infimum of special axis subgroup of $O(3)$ on $V_{l=l_0^+}^{\oplus r}$, $r > 3$, $l_0 > 0$.

|          | $l_0 = 1$ | $l_0 = 2$      | $l_0 = 3$ | $l_0 \geq 4$ | |
|----------|-----------|----------------|-----------|--------------|--------------|
|          |           |                |           | $l_0$ even   | $l_0$ odd    |
| $C_{2v}$ | $K_h$     | $D_{2h}$       | $T_h$     | $D_{2h}$     | $D_{2h}$     |
| $D_2$    | $K_h$     | $D_{2h}$       | $T_h$     | $D_{2h}$     | $D_{2h}$     |
| $D_{2h}$ | $K_h$     | $D_{2h}$       | $T_h$     | $D_{2h}$     | $D_{2h}$     |
| $D_{2d}$ | $K_h$     | $D_{\infty h}$ | $K_h$     | $D_{4h}$     | $D_{4h}$     |

Table 17: Symmetry infimum of polyhedral subgroup of $O(3)$ on $V_{l=l_0^-}^{\oplus r}$, $r > 3$, $l_0 > 0$. The "Caption" in the table takes the value $l_0 = 6, 10, 12, 15, 16, 18, 20 - 22, 24 - 28$.

| $T$   | $l_0 = 6, 9, 10$        | $l_0 = 3, 7$  | $l_0 = 4, 8$ | $l_0 \geq 12$ | other |
|-------|-------------------------|---------------|--------------|---------------|-------|
|       | $T$                     | $T_d$         | $O$          | $T$           | $K_h$ |

| $T_d$ | $l_0 = 3, 6, 7$         | $l_0 \geq 9$  | other        | | |
|-------|-------------------------|---------------|--------------|--|--|
|       | $T_d$                   | $T_d$         | $K_h$        | | |

| $T_h$ | all $l_0$               | | | | |
|-------|-------------------------|--|--|--|--|
|       | $K_h$                   | | | | |

| $O$   | $l_0 = 4, 6, 8, 9, 10$  | $l_0 \geq 12$ | other        | | |
|-------|-------------------------|---------------|--------------|--|--|
|       | $O$                     | $O$           | $K_h$        | | |

| $O_h$ | all $l_0$               | | | | |
|-------|-------------------------|--|--|--|--|
|       | $K_h$                   | | | | |

| $I$   | Caption                 | $l_0 \geq 30$ | other        | | |
|-------|-------------------------|---------------|--------------|--|--|
|       | $I_h$                   | $I_h$         | $K_h$        | | |

| $I_h$ | all $l_0$               | | | | |
|-------|-------------------------|--|--|--|--|
|       | $K_h$                   | | | | |

Table 18: Symmetry infimum of polyhedral subgroup of $O(3)$ on $V_{l=l_0^+}^{\oplus r}$, $r > 3$, $l_0 > 0$. The "Caption" in the table takes the value $l_0 = 6, 10, 12, 15, 16, 18, 20 - 22, 24 - 28$.

| $T$ | $l_0 = 3, 6, 7, 9, 10$ | $l_0 = 4, 8$ | $l_0 \geq 12$ | other |
|---|---|---|---|---|
| | $T_h$ | $O_h$ | $T_h$ | $K_h$ |

| $T_d$ | $l_0 = 4, 6, 8, 9, 10$ | $l_0 \geq 12$ | other |
|---|---|---|---|
| | $O_h$ | $O_h$ | $K_h$ |

| $T_h$ | $l_0 = 3, 6, 7, 9, 10$ | $l_0 = 4, 8$ | $l_0 \geq 12$ | other |
|---|---|---|---|---|
| | $T_h$ | $O_h$ | $T_h$ | $K_h$ |

| $O$ | $l_0 = 4, 6, 8, 9, 10$ | $l_0 \geq 12$ | other |
|---|---|---|---|
| | $O_h$ | $O_h$ | $K_h$ |

| $O_h$ | $l_0 = 4, 6, 8, 9, 10$ | $l_0 \geq 12$ | other |
|---|---|---|---|
| | $O_h$ | $O_h$ | $K_h$ |

| $I$ | Caption | $l_0 \geq 12$ | other |
|---|---|---|---|
| | $I_h$ | $I_h$ | $K_h$ |

| $I_h$ | Caption | $l_0 \geq 12$ | other |
|---|---|---|---|
| | $I_h$ | $I_h$ | $K_h$ |

Table 19: Symmetry infimum of infinite subgroup of $O(3)$ on $V_{l=l_0^-}^{\oplus r}$, $r > 3$, $l_0 > 0$.

| | $l_0$ even | $l_0$ odd |
|---|---|---|
| $C_\infty$ | $D_\infty$ | $C_{\infty v}$ |
| $C_{\infty h} = S_{2\infty}$ | $K_h$ | $K_h$ |
| $C_{\infty v}$ | $K_h$ | $C_{\infty v}$ |
| $D_\infty$ | $D_\infty$ | $K_h$ |
| $D_{\infty h} = D_{\infty d}$ | $K_h$ | $K_h$ |
| $K$ | $K_h$ | $K_h$ |
| $K_h$ | $K_h$ | $K_h$ |

Table 20: Symmetry infimum of infinite subgroup of $O(3)$ on $V_{l=l_0^+}^{\oplus r}$, $r > 3$, $l_0 > 0$.

| | $l_0$ even | $l_0$ odd |
|---|---|---|
| $C_\infty$ | $D_{\infty h}$ | $C_{\infty h}$ |
| $C_{\infty h} = S_{2\infty}$ | $D_{\infty h}$ | $C_{\infty h}$ |
| $C_{\infty v}$ | $D_{\infty h}$ | $K_h$ |
| $D_\infty$ | $D_{\infty h}$ | $K_h$ |
| $D_{\infty h} = D_{\infty d}$ | $D_{\infty h}$ | $K_h$ |
| $K$ | $K_h$ | $K_h$ |
| $K_h$ | $K_h$ | $K_h$ |

# F    Detailed Experiment

Experiments in § 6.1 and 6.2 are conducted with randomly initialized TFN (Thomas et al., 2018) and HEGNN (Cen et al., 2024) architectures as the underlying equivariant models, whereas the experiments in § 6.3 first pretrain a HEGNN encoder to obtain molecular embeddings and then fine-tune separate MLP heads for the final prediction task under different settings.

For TFN (Thomas et al., 2018), we use the implementation from GWL-test (Joshi et al., 2023), which relies on irreducible-representation features with alternating parity. For HEGNN, we extend the original design to a multi-channel variant. In particular, the spherical scalarized features computed for nodes $j$ and $i$ are defined as

$$z_{ij,c}^{(l)} = 1/\sqrt{C} \cdot \langle \tilde{v}_{i,c}^{(l)}, \tilde{v}_{j,c}^{(l)} \rangle, \tag{108}$$

where $\tilde{v}_{i,c}^{(l)}$ and $\tilde{v}_{j,c}^{(l)}$ denote the $l$-th degree steerable features of channel $c$ among a total of $C$ channels. To obtain the degree-$l_0$ graph-level representation, we apply global mean pooling for each graph $\mathcal{G}(\mathcal{V}, \mathcal{E})$:

$$\tilde{v}_{\mathcal{G},c}^{(l)} = 1/|\mathcal{V}| \cdot \sum_i \tilde{v}_{i,c}^{(l)}. \tag{109}$$

We will detail, in this section, how each experimental task processes these graph-level features.

## F.1    Visualization of Representation Space

We use a TFN with single-layer to calculate the embedding. After that, to extract features of degree $l_0$, we append an `o3.Linear` layer from `e3nn` (Geiger & Smidt, 2022), denoting this setup as $\text{TFN}_{l=l_0}$ in our experiments.

## F.2    Expressivity on Symmetric Graphs

This section first introduces the detailed settings of § 6.2 in § F.2.1, and then introduces the reproduction of the original experiment of GWL-test (Joshi et al., 2023) in § F.2.2.

### F.2.1    Embedding Difference Norm Experiment

We employed both TFN (Thomas et al., 2018) and HEGNN (Cen et al., 2024) to compute the norm of the embedding difference across 12 configurations for each model, varying the number of channels (1, 4, 16) and layers (1, 2, 3, 4). The degree-$l$ discrepancy between two graphs $\mathcal{G}_0$ and $\mathcal{G}_1$ is defined as

$$\Delta^{(l)} = 1/|C| \cdot \sum_{c=1}^{C} \|\tilde{v}_{\mathcal{G}_0,c}^{(l)} - \tilde{v}_{\mathcal{G}_1,c}^{(l)}\|. \tag{110}$$

These choices give rise to $2 \times 3 \times 4 = 24$ distinct $\Delta^{(l)}$. In Fig. 5, we report the maximum norm across all $\Delta^{(l)}$. Since every norm is strictly positive, a maximal value below $10^{-6}$ indicates that all corresponding norms fall below $10^{-6}$, meaning none of them can distinguish $\mathcal{G}_0$ from $\mathcal{G}_1$.

### F.2.2    Original GWL-Test on Symmetric Graphs

**Dataset.**  Same as the setting in (Joshi et al., 2023), we construct four symmetric $k$-fold structures ($k \in \{2, 3, 4, 6\}$), each centered at the origin. For each structure $\mathcal{G}_0$ we apply a random rotation to produce $\mathcal{G}_1$ which ensures $\mathcal{G}_1$ does not coincide with the original $\mathcal{G}_0$. The goal is to evaluate whether different equivariant neural network architectures can distinguish $\mathcal{G}_0$ from $\mathcal{G}_1$. To validate distinct aspects of our theory, we consider rotations separately in 2D and 3D; in the 3D experiments we additionally ensure that $\mathcal{G}_1$ is not coplanar with $\mathcal{G}_0$.

**Embeddings.**  The extracted $l_0$-degree embeddings from TFN are fed into a vanilla classifier for the classification task. The model was trained for 300 epochs to ensure the classifier had sufficient capacity to discriminate the classes.

**Results.**  The detailed experimental results are presented in Table 21, which can be observed that the color blocks in this table are completely consistent with Fig. 5. It demonstrates that our theoretical predictions in Table 1 are in complete agreement with the empirical findings obtained from the model.

Such remarkable consistency not only confirms the correctness of our theoretical analysis, but also highlights the importance of constructing mappings with appropriately chosen features.

Table 21: Results of distinguishing $k$-fold structures rotated in 2D/3D space.

| GNN Layer | 2D Rotational Symmetry | | | | 3D Rotational Symmetry | | | |
|---|---|---|---|---|---|---|---|---|
| | 2 fold | 3 fold | 4 fold | 6 fold | 2 fold | 3 fold | 4 fold | 6 fold |
| $\text{TFN}_{l=0}$ | $50.0 \pm 0.0$ | $50.0 \pm 0.0$ | $50.0 \pm 0.0$ | $50.0 \pm 0.0$ | $50.0 \pm 0.0$ | $50.0 \pm 0.0$ | $50.0 \pm 0.0$ | $50.0 \pm 0.0$ |
| $\text{TFN}_{l=1}$ | $50.0 \pm 0.0$ | $50.0 \pm 0.0$ | $50.0 \pm 0.0$ | $50.0 \pm 0.0$ | $50.0 \pm 0.0$ | $50.0 \pm 0.0$ | $50.0 \pm 0.0$ | $50.0 \pm 0.0$ |
| $\text{TFN}_{l=2}$ | $\mathbf{100.0 \pm 0.0}$ | $50.0 \pm 0.0$ | $50.0 \pm 0.0$ | $50.0 \pm 0.0$ | $\mathbf{100.0 \pm 0.0}$ | $\mathbf{100.0 \pm 0.0}$ | $\mathbf{100.0 \pm 0.0}$ | $\mathbf{100.0 \pm 0.0}$ |
| $\text{TFN}_{l=3}$ | $50.0 \pm 0.0$ | $\mathbf{100.0 \pm 0.0}$ | $50.0 \pm 0.0$ | $50.0 \pm 0.0$ | $50.0 \pm 0.0$ | $\mathbf{100.0 \pm 0.0}$ | $50.0 \pm 0.0$ | $50.0 \pm 0.0$ |
| $\text{TFN}_{l=4}$ | $\mathbf{100.0 \pm 0.0}$ | $50.0 \pm 0.0$ | $\mathbf{100.0 \pm 0.0}$ | $50.0 \pm 0.0$ | $\mathbf{100.0 \pm 0.0}$ | $\mathbf{100.0 \pm 0.0}$ | $\mathbf{100.0 \pm 0.0}$ | $\mathbf{100.0 \pm 0.0}$ |
| $\text{TFN}_{l=5}$ | $50.0 \pm 0.0$ | $\mathbf{100.0 \pm 0.0}$ | $50.0 \pm 0.0$ | $50.0 \pm 0.0$ | $50.0 \pm 0.0$ | $\mathbf{100.0 \pm 0.0}$ | $50.0 \pm 0.0$ | $50.0 \pm 0.0$ |
| $\text{TFN}_{l=6}$ | $\mathbf{100.0 \pm 0.0}$ | $\mathbf{100.0 \pm 0.0}$ | $\mathbf{100.0 \pm 0.0}$ | $\mathbf{100.0 \pm 0.0}$ | $\mathbf{100.0 \pm 0.0}$ | $\mathbf{100.0 \pm 0.0}$ | $\mathbf{100.0 \pm 0.0}$ | $\mathbf{100.0 \pm 0.0}$ |
| $\text{TFN}_{l=7}$ | $50.0 \pm 0.0$ | $\mathbf{100.0 \pm 0.0}$ | $50.0 \pm 0.0$ | $50.0 \pm 0.0$ | $50.0 \pm 0.0$ | $\mathbf{100.0 \pm 0.0}$ | $50.0 \pm 0.0$ | $50.0 \pm 0.0$ |
| $\text{TFN}_{l=8}$ | $\mathbf{100.0 \pm 0.0}$ | $\mathbf{100.0 \pm 0.0}$ | $\mathbf{100.0 \pm 0.0}$ | $\mathbf{100.0 \pm 0.0}$ | $\mathbf{100.0 \pm 0.0}$ | $\mathbf{100.0 \pm 0.0}$ | $\mathbf{100.0 \pm 0.0}$ | $\mathbf{100.0 \pm 0.0}$ |
| $\text{TFN}_{l=9}$ | $50.0 \pm 0.0$ | $\mathbf{100.0 \pm 0.0}$ | $50.0 \pm 0.0$ | $50.0 \pm 0.0$ | $50.0 \pm 0.0$ | $\mathbf{100.0 \pm 0.0}$ | $50.0 \pm 0.0$ | $50.0 \pm 0.0$ |
| $\text{TFN}_{l=10}$ | $\mathbf{100.0 \pm 0.0}$ | $\mathbf{100.0 \pm 0.0}$ | $\mathbf{100.0 \pm 0.0}$ | $\mathbf{100.0 \pm 0.0}$ | $\mathbf{100.0 \pm 0.0}$ | $\mathbf{100.0 \pm 0.0}$ | $\mathbf{100.0 \pm 0.0}$ | $\mathbf{100.0 \pm 0.0}$ |
| $\text{TFN}_{l=11}$ | $50.0 \pm 0.0$ | $\mathbf{100.0 \pm 0.0}$ | $50.0 \pm 0.0$ | $50.0 \pm 0.0$ | $50.0 \pm 0.0$ | $\mathbf{100.0 \pm 0.0}$ | $50.0 \pm 0.0$ | $50.0 \pm 0.0$ |

## F.3 MOLECULE PROPERTY PREDICTION WITH PRETRAINED EQUIVARIANT FEATURES

### F.3.1 DETAILED EXPERIMENTAL SETUP

We employ HEGNN as the encoder in our experiments. TFN is computationally prohibitive in this setting: its tensor-product operator has a complexity of $\mathcal{O}(L^6)$, and when the maximum degree is set to $L = 11$ (the upper limit supported by e3nn), training a four-layer TFN with only four irreducible-representation channels on QM9 requires roughly 10 A100 GPU-hours per epoch. This far exceeds any practical budget. In contrast, HEGNN adopts spherical scalarization, where interactions across different degrees are mediated solely through scalars, reducing the complexity to $\mathcal{O}(L^2)$. For this reason, we choose HEGNN as our encoder.

Concretely, we use a four-layer HEGNN with 16 irreducible-representation channels and a hidden dimension of 64. The resulting features are passed through a scalarization layer and then into a two-layer MLP to predict the molecular isotropic polarizability $\alpha$ on QM9. During fine-tuning, we freeze the encoder, apply a mask to selectively remove information, and train a separate two-layer MLP for each setting. Specifically, our designed mask multiplies the features of the unselected degrees by 0, followed by scalarization calculation, that is:

$$\tilde{\boldsymbol{v}}_c^{(l)} = \text{HEGNN}(\mathcal{G}), \tag{111}$$

$$\boldsymbol{s}^{(l)} = \text{vec}([\langle \tilde{\boldsymbol{v}}_{c_1}^{(l)}, \tilde{\boldsymbol{v}}_{c_2}^{(l)} \rangle])_{C \times C}, l = 1, \dots, 11, \tag{112}$$

$$\hat{\alpha} = \text{MLP}_{\text{finetune}}(\tilde{\boldsymbol{v}}^{(0)}, \boldsymbol{s}^{(1)}, \dots, \boldsymbol{s}^{(11)}), \tag{113}$$

Following the standard protocol, we train on the first 110k molecules for 300 epochs and fine-tune for an additional 30 epochs. Notably, although the final visualizations are produced by running the trained models on the entire QM9 dataset[5], this does not affect our theoretical conclusions, as our analysis relies solely on horizontal comparisons rather than absolute predictive performance.

### F.3.2 CASE STUDIES

To further validate our theory, we analyze three prominent point groups $C_{2h}$, $C_{3h}$, and $T_d$ (see Fig. 7). Each exhibiting a representative pattern of symmetry increase. These groups display a symmetry increase to the full group $O(3)$ under specific conditions: for $C_{2h}$ when $l_0$ is odd, for $C_{3h}$ when $l_0 = 1$, and for $T_d$ when $l_0 = 1, 2, 5$. This increase corresponds to full degeneration, making the features non-discriminative. The impact of this degeneration is evident in our experiments. Due to the full degeneration of 1-degree features, introducing them paradoxically decreases predictive performance for molecules with $C_{2h}$, $C_{3h}$, and $T_d$ symmetry. Following the introduction of 2-degree features, a marked improvement in performance is observed for $C_{2h}$ and $C_{3h}$. Conversely, for $T_d$, performance again decreases, a result directly attributable to the fact that its 2-degree features also undergo full degeneration.

---

[5]Minor discrepancies between $l = 0$ in Fig. 6 and $l \leq 0$ in Fig. 7 result from not fixing the random seed and do not affect the conclusions.

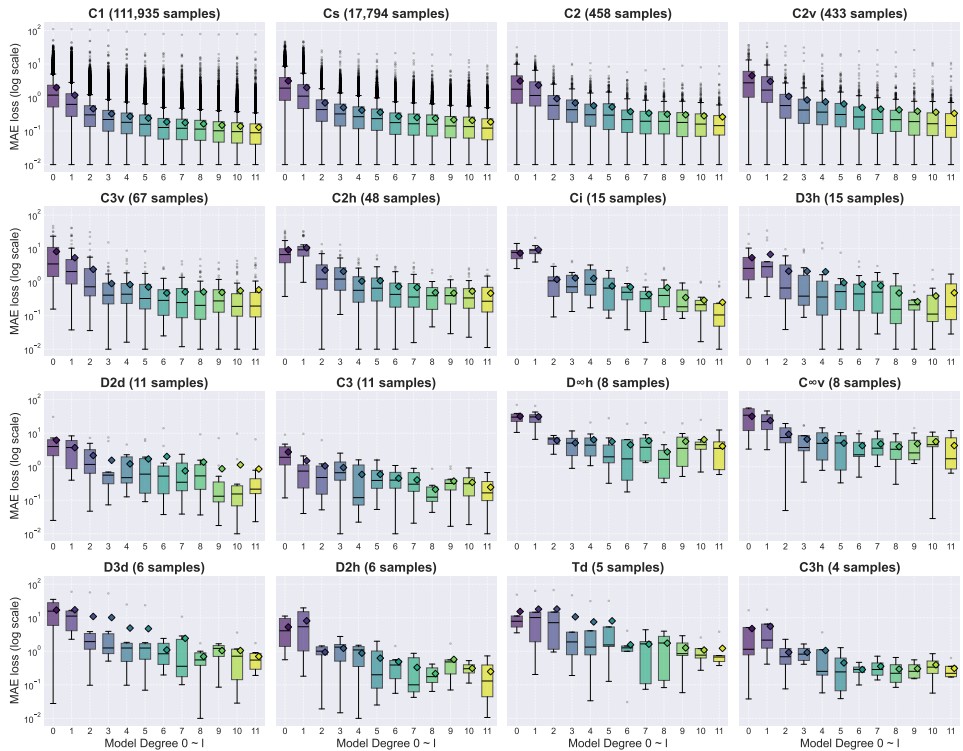

Figure 7: MAE loss (in units of $a_0^3$) for isotropic polarizability prediction with degree $l \leq l_0$ across molecules from the top-16 point groups by molecular count. Each boxplot shows the distribution of errors at a given degree, while diamond markers denote the corresponding mean MAE.

The experiment shows strong dependence between model performance and feature choice on a real-world dataset validates our discussion on symmetry increase and feature space dimension. It demonstrates a critical model design principle: not only should one avoid building a model entirely from fully degenerate features, but one should also avoid including individual feature components that undergo full degeneration, as they can be actively detrimental to predictive performance.

Table 22: Symmetry infimum of point group symmetry on the QM9 dataset.

| | 0 | 1 | 2 | 3 | 4 | 5 | 6 | 7 | 8 | 9 | 10 | 11 |
|---|---|---|---|---|---|---|---|---|---|---|---|---|
| $C_1$ | $K$ | $C_1$ | $C_s$ | $C_1$ | $C_s$ | $C_1$ | $C_s$ | $C_1$ | $C_s$ | $C_1$ | $C_s$ | $C_1$ |
| $C_s$ | $K_h$ | $C_s$ | $C_{2h}$ | $C_s$ | $C_{2h}$ | $C_s$ | $C_{2h}$ | $C_s$ | $C_{2h}$ | $C_s$ | $C_{2h}$ | $C_s$ |
| $C_2$ | $K$ | $C_{\infty v}$ | $C_{2h}$ | $C_2$ | $C_{2h}$ | $C_2$ | $C_{2h}$ | $C_2$ | $C_{2h}$ | $C_2$ | $C_{2h}$ | $C_2$ |
| $C_{2v}$ | $K_h$ | $C_{\infty v}$ | $D_{2h}$ | $C_{2v}$ | $D_{2h}$ | $C_{2v}$ | $D_{2h}$ | $C_{2v}$ | $D_{2h}$ | $C_{2v}$ | $D_{2h}$ | $C_{2v}$ |
| $C_{3v}$ | $K_h$ | $D_{\infty h}$ | $C_{\infty v}$ | $C_{3v}$ | $D_{3d}$ | $C_{3v}$ | $D_{3d}$ | $C_{3v}$ | $D_{3d}$ | $C_{3v}$ | $D_{3d}$ | $C_{3v}$ |
| $C_{2h}$ | $K_h$ | $K_h$ | $C_{2h}$ | $K_h$ | $C_{2h}$ | $K_h$ | $C_{2h}$ | $K_h$ | $C_{2h}$ | $K_h$ | $C_{2h}$ | $K_h$ |
| $C_i = S_2$ | $K_h$ | $K_h$ | $C_i$ | $K_h$ | $C_i$ | $K_h$ | $C_i$ | $K_h$ | $C_i$ | $K_h$ | $C_i$ | $K_h$ |
| $D_{3h}$ | $K_h$ | $K_h$ | $D_{\infty h}$ | $D_{3h}$ | $D_{\infty h}$ | $D_{3h}$ | $D_{6h}$ | $D_{3h}$ | $D_{6h}$ | $D_{3h}$ | $D_{6h}$ | $D_{3h}$ |
| $D_{2d}$ | $K_h$ | $K_h$ | $D_{\infty h}$ | $T_d$ | $D_{4h}$ | $D_{2d}$ | $D_{4h}$ | $D_{2d}$ | $D_{4h}$ | $D_{2d}$ | $D_{4h}$ | $D_{2d}$ |
| $C_3$ | $K$ | $C_{\infty v}$ | $D_{\infty h}$ | $C_3$ | $C_{3h}$ | $C_3$ | $C_{3h}$ | $C_3$ | $C_{3h}$ | $C_3$ | $C_{3h}$ | $C_3$ |
| $D_{\infty h}$ | $K_h$ | $K_h$ | $D_{\infty h}$ | $K_h$ | $D_{\infty h}$ | $K_h$ | $D_{\infty h}$ | $K_h$ | $D_{\infty h}$ | $K_h$ | $D_{\infty h}$ | $K_h$ |
| $C_{\infty v}$ | $K_h$ | $C_{\infty v}$ | $D_{\infty h}$ | $C_{\infty v}$ | $D_{\infty h}$ | $C_{\infty v}$ | $D_{\infty h}$ | $C_{\infty v}$ | $D_{\infty h}$ | $C_{\infty v}$ | $D_{\infty h}$ | $C_{\infty v}$ |
| $D_{3d}$ | $K_h$ | $K_h$ | $D_{\infty h}$ | $K_h$ | $D_{3d}$ | $K_h$ | $D_{3d}$ | $K_h$ | $D_{3d}$ | $K_h$ | $D_{3d}$ | $K_h$ |
| $D_{2h}$ | $K_h$ | $K_h$ | $D_{2h}$ | $K_h$ | $D_{2h}$ | $K_h$ | $D_{2h}$ | $K_h$ | $D_{2h}$ | $K_h$ | $D_{2h}$ | $K_h$ |
| $T_d$ | $K_h$ | $K_h$ | $K_h$ | $T_d$ | $O_h$ | $K_h$ | $O_h$ | $T_d$ | $O_h$ | $T_d$ | $O_h$ | $T_d$ |
| $C_{3h}$ | $K_h$ | $K_h$ | $D_{\infty h}$ | $C_{3h}$ | $D_{\infty h}$ | $C_{3h}$ | $C_{6h}$ | $C_{3h}$ | $C_{6h}$ | $C_{3h}$ | $C_{6h}$ | $C_{3h}$ |

