# OpenReview forum: "Reducing Symmetry Increase in Equivariant Neural Networks"
_ICLR.cc/2026/Conference — ICLR 2026 Poster_

### Official Review · Reviewer_gQcX · 2025-11-01

**Soundness:** 4
**Presentation:** 2
**Contribution:** 4
**Rating:** 8
**Confidence:** 3

**Summary:**

The paper investigates why equivariant models can unintentionally increase symmetry in their outputs, causing distinct inputs to become indistinguishable. It introduces a practical notion of a symmetry lower bound determined solely by the model feature space: if the feature space cannot realize the input symmetry, the output is guaranteed to gain extra symmetry. Building on this idea, the authors provide a computable test that predicts when symmetry increase will happen and how severe it will be. They also show that, under the manifold hypothesis, generic equivariant models can avoid unnecessary symmetry increase. Experiments on controlled setups closely match the predictions.

**Strengths:**

- The paper is well structured and uses clear notation.
- Although I did not check the proofs in depth, the results appear sound and internally consistent.
- The paper positions itself as a relevant contribution to the literature on symmetry-breaking inputs in equivariant machine learning. It formalizes the notion of a symmetry infimum, providing a simple, checkable target for assessing symmetry increase.
- The authors provide all the necessary machinery to derive this object.

**Weaknesses:**

- Despite the paper being well organized, the exposition is overly compact and hard to follow, as strong backgrounds in differential geometry and Lie groups are uncommon in the ML community. While the paper need not be self-contained, a more explicit notation guide in the appendix would help. In the eventual camera-ready, the extra page could be used to introduce key concepts in plain language with brief examples.
- The (almost)-isovariance results rely on the manifold hypothesis and smooth parameterizations. Non-smooth and distributional regimes, common in practice, are not addressed. Consider clarifying the scope of the claims and discussing whether any results extend beyond the smooth setting.

**Questions:**

1. Can you discuss the practical feasibility of using higher-dimensional irreducible representations in the output representation, or of increasing their multiplicities? Does this lead to computational or memory bottlenecks during training or inference?
2. Is it possible to adapt Theorem 5.2 to a tubular-neighborhood hypothesis, namely that the data distribution is supported in a tubular neighborhood of, rather than exactly in, a smooth low-dimensional submanifold?

---

> ### Author Response · Authors · 2025-11-22
>
> > **R1 (W1)** Explicit notation and key concepts guide in the appendix.
> > - **W1** ("a more explicit notation guide in the appendix would help ... the extra page could be used to introduce key concepts in plain language with brief examples.")
>
> Thank you for your suggestion. It will be very helpful in making our theory more accessible to the machine learning community. We have begun to incorporate relevant background material on group theory and differential geometry, and will further expand this part in the revised manuscript.
>
> > **R2 (W2)** Dissussion whether any results extend beyond the smooth setting.
> > - **W2** ("The (almost)-isovariance results rely on the manifold hypothesis and smooth parameterizations. Non-smooth and distributional regimes, common in practice, are not addressed. Consider clarifying the scope of the claims and discussing whether any results extend beyond the smooth setting")
>
> That is a keen observation. For the smoothness setting mentioned by the reviewer, we divide the discussion into two parts: **parameterization smoothness** and **data smoothness**. For parameterization smoothness, the smooth parameterization is a critical assumption for our conclusions, and relaxing this assumption poses substantial challenges. In contrast, regarding the relaxation of data smoothness, we have in fact made some efforts toward this direction in the paper.
>
> **Difficulty with Non-smooth Parameterization:**
>
> Our proof relies on the openness and density of maps that satisfy Bierstone's equivariant general position [A]. It is by combining this property with the density of the family of model parameter maps that we derive the dimensional properties of the manifold consisting of points where the symmetry of a generic model map changes.
>
> The primary difficulty lies in the fact that the definition of Bierstone's equivariant general position in [A] depends on the module of equivariant functions $C_{G}^{\infty}(U, V)$ being representable as a $C_{G}^{\infty}(U)$-module generated by basic equivariant polynomials. However, as pointed out in Remark 6.6.1 of [B], similar conclusion does not necessarily hold for $C_{G}^{k}(U, V)$ when $k < \infty$. Given the complexity of the discussion for a finite $k$, our efforts in this work are confined to the $k = \infty$ case.
>
> **Effort to Handle Non-smooth Data Structures:**
>
> While our analysis assumes smooth parameterization, we have made efforts to handle certain non-smooth data structures. As noted in [C] (**Sec. 5.11.3**), a "manifold" in machine learning differs from a mathematical one, as it may not account for abrupt topological changes like self-intersections, where data degrees of freedom can suddenly increase. Our "union of manifolds" assumption partially circumvents the restriction to smooth submanifolds by covering classic cases of self-intersection (e.g., the image of a smooth immersion of a compact smooth manifold can be represented as a finite union of compact smooth manifolds). This assumption has also received some experimental validation [D].
>
> Nevertheless, a full treatment of non-smoothness remains a key challenge. Our future work will be centered on extending this analysis to the more general case of $C^k$ continuity.
>
> > **R3 (Q1)** Practical feasibility of using higher-dimensional irreducible representation and time and space complexity analysis.
> > - **Q1** ("Can you discuss the practical feasibility of using higher-dimensional irreducible representations in the output representation, or of increasing their multiplicities? Does this lead to computational or memory bottlenecks during training or inference?")
>
>
> Thank you for your insightful comment. Increasing the dimensionality (degree) or the multiplicity (channels) of the irreducible representations is entirely feasible. Several studies have noted that using higher-degree features can be crucial for improving model performance [E, F, G]. As an applied example, in the QM9 molecular property prediction task (**Tabs. S2, S3** & **Figs. 6, 7**), we already make use of 16-channel features up to $l_0=11$, which is the highest degree currently supported by the `e3nn` library. Regarding multiplicity, the baseline model employs three variants with 1, 4, and 16 channels, while our setting uses 16 channels. The corresponding experimental details can be found in the general response **G1**.
>
> The practical limitation is time complexity. For degree $L$ and multiplicity $C$, TFN requires tensor-product operations with a cost of $\mathcal{O}(C^{2}L^{6})$. In our QM9 setting, even a modest 4-channel TFN takes roughly 10 A100 GPU-hours per epoch, making higher-order configurations prohibitively slow. For this reason, we use HEGNN [E], whose spherical-scalarization mechanism reduces the complexity to $\mathcal{O}(CL^{2})$.
>
> Moreover, efficient implementations such as eSCN [H] further accelerate tensor-product computations, making the use of even higher-degree features increasingly practical.

---

> > ### Author Response · Authors · 2025-11-22
> >
> > > **R4 (Q2)** Adapting **Thm. 5.2** to a tubular-neighborhood hypothesis.
> > > - **Q2** ("Is it possible to adapt Theorem 5.2 to a tubular-neighborhood hypothesis, namely that the data distribution is supported in a tubular neighborhood of, rather than exactly in, a smooth low-dimensional submanifold?")
> >
> > That's an excellent question. When defining almost equivariance, we chose to base our discussion on the Hausdorff measure of submanifolds. Extending this to a probability measure would undoubtedly be a more realistic assumption, especially given the presence of observational noise. However, this generalization presents the following difficulties.
> >
> > **Difficulty with Non-compact Structure:**
> >
> > If the data is supported on an open set, that set is not compact. Our proofs related to density rely on the openness of the set of maps satisfying equivariant general position, and this openness is guaranteed by the compactness of the data manifold (see **Prop. E.8** or [A]).
> >
> > A possible solution is to extend our conclusions to compact manifolds with boundaries and then consider such manifolds encompassing tubular neighborhoods. This would require generalizing the concept of equivariant general position to manifolds with boundaries. However, this is a highly non-trivial task, as the existing mathematical tools are almost exclusively developed for the simpler case without boundary.
> >
> > Future work will focus on scenarios where the data is supported on non-compact spaces, and we will attempt to relax the topological constraints on the data manifold.
> >
> > > **Reference**
> >
> > - [A] Bierstone E. General Position of Equivariant Maps.
> > - [B] Field M. Dynamics and Symmetry.
> > - [C] Goodfellow I, Bengio Y, Courville A. Deep learning.
> > - [D] Brown B C, Caterini A L, Ross B L et al. Verifying the Union of Manifolds Hypothesis for Image Data. ICLR'23.
> > - [E] Cen J, Li A, Lin N, et al. Are high-degree representations really unnecessary in equivariant graph neural networks? NeurIPS'24.
> > - [F] Frank T, Unke O, Müller K R. So3krates: Equivariant attention for interactions on arbitrary length-scales in molecular systems. NeurIPS'22
> > - [G] Xie Y Q, Daigavane A, Kotak M, et al. The Price of Freedom: Exploring Expressivity and Runtime Tradeoffs in Equivariant Tensor Products. ICML'25.
> > - [H] Passaro S, Zitnick C L. Reducing SO (3) convolutions to SO (2) for efficient equivariant GNNs. ICML'23.

---

> ### Author Response · Authors · 2025-11-26
> **Kindly request your feedback before the end of the discussion period**
>
> Dear Reviewer gQcX,
>
> As the author–reviewer discussion period is approaching its end, we would greatly appreciate it if you could kindly take a moment to review our responses at your convenience. If there are any remaining questions or points that would benefit from further clarification, we would be more than happy to address them before the discussion window closes.
>
> Thank you very much for your time and valuable efforts.
>
> Sincerely,
>
> The Authors

---

> ### Author Response · Authors · 2025-11-29
>
> Dear Reviewer gQcX,
>
> We truly appreciate your professional feedback and constructive suggestions, which have significantly helped to improve the quality and readability of our manuscript.
> Due to the temporary policy introduced after the recent incident at ICLR, we are no longer able to continue the discussion in this review cycle. We regret that we are unable to engage in further discussion.
>
> We would like to express our gratitude once again. Your sincere recognition and valuable advice are a great encouragement and inspiration to us.
>
> Best regards,
>
> The Authors

---

### Official Review · Reviewer_cFwq · 2025-11-01

**Soundness:** 3
**Presentation:** 3
**Contribution:** 2
**Rating:** 4
**Confidence:** 3

**Summary:**

The paper presents a rigorous theoretical framework for analyzing symmetry increase in equivariant neural networks, when outputs become more symmetric than inputs, reducing expressivity. It introduces the notion of a symmetry infimum, a computable lower bound determined by the feature space, and provides an algorithm for computing it. Experiments on synthetic symmetric structures validate the theoretical predictions. However, the work remains highly abstract and focuses mainly on characterizing rather than reducing symmetry increase, offering limited practical guidance for model design.

**Strengths:**

The paper addresses an underexplored but conceptually important issue in equivariant learning, the tendency of models to exhibit unwanted symmetry increase. It introduces an original and rigorous theoretical framework based on the notion of a symmetry infimum, supported by solid mathematical analysis and consistent experimental validation. Although abstract, it represents a meaningful theoretical advance in understanding the limitations of equivariant neural networks.

**Weaknesses:**

While the paper offers a rigorous theoretical framework for characterizing symmetry increase, it lacks a clear and consolidated discussion of how to mitigate it in practice. The ideas for reduction, such as adjusting representation degree or multiplicity, are scattered across sections and never formulated as concrete design guidelines, making the work hard to apply in real settings. The paper also feels rather abstract and narrowly focused; the topic is niche within equivariant learning, and the presentation does little to connect theory to practice. Moreover, the title’s promise of “reducing” symmetry increase is not fully realized, as the paper mainly analyzes rather than mitigates the phenomenon.

**Questions:**

The title and framing suggest that the paper provides methods for reducing symmetry increase, yet most of the work focuses on characterizing it. Could the authors clarify what concrete design principles or architectural adjustments actually help reduce symmetry increase in practice?

Could the authors provide more intuition or visual explanation of the symmetry infimum concept to make the theoretical framework more accessible to practitioners? What about simpler examples? (Consider a 2D shape such as an equilateral triangle. The triangle itself has 3-fold rotational symmetry, it looks the same every 120° rotation. Now, imagine an equivariant network that processes this triangle and produces a feature representation...)

Beyond synthetic examples, do the authors foresee domains (e.g., molecular modeling, materials science) where symmetry increase significantly impacts performance and where their theory could guide practical model improvements?

The phenomenon of symmetry increase seems loosely analogous to aliasing in the Nyquist–Shannon theorem, where insufficient sampling along the translation group causes distinct frequencies to become indistinguishable. In the present context, a restricted feature space (e.g., limited degrees of irreducible representations) similarly prevents the network from distinguishing between different orientations, leading to a kind of symmetry aliasing over the rotation group. Could the authors comment on whether this analogy is meaningful, and whether the symmetry infimum can be seen as a group-theoretic counterpart to the Nyquist limit for translational symmetry?

---

> ### Author Response · Authors · 2025-11-22
>
> > **R1 (W1 & Q3).**  Formulation of concrete design guidelines leading to practical model improvement.
> > - **W1** ("While the paper offers a rigorous theoretical framework for characterizing symmetry increase, it lacks a clear and consolidated discussion of how to mitigate it in practice. The ideas for reduction, such as adjusting representation degree or multiplicity, are scattered across sections and never formulated as concrete design guidelines, making the work hard to apply in real settings. The paper also feels rather abstract and narrowly focused; the topic is niche within equivariant learning, and the presentation does little to connect theory to practice.")
> > - **Q3** ("Beyond synthetic examples, do the authors foresee domains (e.g., molecular modeling, materials science) where symmetry increase significantly impacts performance and where their theory could guide practical model improvements?")
>
> Thank you for your valuable suggestions. In our initial version, we presented a theoretical method for mitigating symmetry increase This method requires ensuring the output feature space contains the orbit type of the input symmetry (as detailed in **Thms 3.2, 5.2**). However, we acknowledge that the lack of direct model design guidelines and experiments on real-world datasets may have led to doubts about the theory's practical utility. We have therefore made the following significant additions to the paper.
>
> **Guidelines for Orientation-dependent Tasks:**
>
> We have revised **Sec 4.2**, replacing the previous discussion on the properties of orbit types for general representations with a focused guide on equivariant model design. In the new **Sec. 4.2**, we explain that **one can control the overall symmetry increase of a model by itemizing the possible input symmetries and then carefully controlling the symmetry infimum of each feature component through precomputed results**.
>
> As a special case, **if the objective is to eliminate symmetry increase as much as possible, the symmetry infimum of the feature components must match the orbit type of the input symmetry**. This is critical for tasks that depend on orientation, such as the experiment in **Appendix G.2** on distinguishing rotated geometric graphs. In such cases, symmetry increase can be detrimental to model performance because it causes a loss of orientational information, rendering the equivariant features unable to distinguish between an input and its rotated versions. Our experimental results in **Sec. 6.3** reflect this phenomenon.
>
> **Guidelines for General Tasks:**
>
> Even in orientation-independent tasks, certain types of symmetry increase should be avoided. This is because the symmetry of a feature is linked to the dimensionality of its fixed-point subspace. For example, for any non-trivial representation, if a feature's symmetry increases to that of the full group, the dimension of its fixed-point subspace collapses to zero. Therefore, **when designing for equivariant learning tasks with non-trivial input symmetries, one should avoid selecting feature components where the predicted symmetry increase implies a severe compression of the feature dimension**. For the O(3) case, the tables in **Appendix F.4** show that for certain non-trivial, high-multiplicity feature components $V_{l=l_0^{\pm}}^{\oplus r}, r > 3$, the symmetry increases to O(3). These components should generally be avoided, as they lack the capacity to equivariantly encode inputs with the given symmetry.
>
> **Experimental Results on a Real-World Dataset:**
>
> As outlined in general response **G1**, we have added a supplementary orientation-independent QM9 experiment to further illustrate the practical implications of our theoretical framework. Briefly, we have evaluated isotropic polarizability on QM9, which contains many highly symmetric molecules across 22 point groups (**Tab. S1**). We have pretrained an equivariant encoder (degrees 0–11) and fine-tuned masked MLP heads that selectively activate individual degrees, allowing us to directly examine how symmetry-induced degeneracies affect prediction accuracy. The observed trends (**Tabs. S2, S3**; **Figs. 6, 7**) align closely with our theoretical predictions.
>
> In the **Case Study** of **G1**, we provide molecular examples exhibiting $C_{2h}$, $D_{3h}$, and $T_d$  symmetries. The experimental results demonstrate that following the feature-selection principle leads to improved model performance on symmetric datasets.
>
> This strong dependence between model performance and feature choice on a real-world dataset validates our discussion on symmetry increase and feature space dimension. It demonstrates a critical design principle: **not only should one avoid building a model entirely from fully degenerate features, but one should also avoid including individual feature components that undergo full degeneration, as they can be actively detrimental to predictive performance**.

---

> > ### Author Response · Authors · 2025-11-22
> >
> > > **R2 (W2 & Q1).** Discussion of mitigating symmetry increase.
> > > - **W2** ("the title’s promise of “reducing” symmetry increase is not fully realized, as the paper mainly analyzes rather than mitigates the phenomenon."")
> > > - **Q1** ("The title and framing suggest that the paper provides methods for reducing symmetry increase, yet most of the work focuses on characterizing it. Could the authors clarify what concrete design principles or architectural adjustments actually help reduce symmetry increase in practice?")
> >
> > Thank you for your comments on our paper. We would like to clarify a potential misunderstanding that our paper only characterizes symmetry increase without offering a way to mitigate it. In fact, the entire theoretical framework of our paper is constructed around the very goal of mitigating symmetry increase.
> >
> > After defining the symmetry infimum in **Sec. 3**, we introduced the concept of an **isovariant mapping** in **Sec. 3.1** precisely to formalize the condition where an equivariant map's input does not undergo symmetry increase. **Therefore, to eliminate symmetry increase, we require such isovariant mappings to exist**. In **Thm. 3.2**, we provide a necessary condition for their existence: **the output space must contain the orbit type corresponding to the input symmetry**. Subsequently, in **Sec. 5**, we show that this condition is, in a sense, also sufficient by defining the concept of an **almost isovariant** map. We demonstrate in **Thm. 5.2** that for an appropriately chosen output space, a mapping being almost isovariant is a generic property (i.e., it is the typical case). This theoretical development leads to a direct strategy for preventing symmetry increase: ensure the model's feature space contains the orbit type of the input symmetry. **Sec. 4** of our paper is dedicated to providing the tools for calculating these orbit types.
> >
> > We believe this misunderstanding may have arisen because our initial manuscript lacked a clear connection to practical scenarios. We have now addressed this by adding new content to **Sec. 4.2**. In **Sec. 4.2**, we explicitly outline a strategy for controlling the model's output symmetry increase by **carefully selecting features to control the symmetry infimum** (for a detailed discussion, please see our response **R1**). Following this strategy, if the goal is to eliminate symmetry increase as much as possible (e.g., in orientation-dependent tasks such as **Appendix G.2**), **one can design the output feature components to include the orbit type of the input symmetry**. This lowers the symmetry infimum to the level of the input symmetry itself. In this case, **Thm. 5.2** provides a guarantee that a parameterized map will "almost never" exhibit symmetry increase. In addition, to help readers better understand the algorithms in Sec. 4, we have also included the algorithmic workflows (**Algos. 1, 2**) and a simple worked example in the revised version.
> >
> > > **R3 (Q2).** Intuition or visual explanation of the symmetry infimum concept.
> > > - **Q2** ("Could the authors provide more intuition or visual explanation of the symmetry infimum concept to make the theoretical framework more accessible to practitioners? What about simpler examples? (Consider a 2D shape such as an equilateral triangle...)")
> >
> > Thank you for your feedback. We would like to clarify that our paper already includes visualizations to explain the concept of the symmetry infimum. For instance, to illustrate the three types of symmetry degeneration derived from the symmetry infimum of the group $D_{kh}$, we provide a schematic in **Fig. 2**. This figure explicitly shows how symmetry increase, as determined by the infimum, leads to the degeneration of group action orbits. Furthermore, to demonstrate the practical impact of the infimum, we present an experiment involving the equivariant encoding and rotation of k-fold point clouds. **Figs. 3, 4** illustrate how this infimum makes it impossible for an equivariant model to distinguish certain rotations that lie outside the k-fold geometric symmetry group. This directly demonstrates an increase in the symmetry of the k-fold equivariant features.
> >
> > Regarding your suggestion for an equilateral triangle example, we note that it is nearly identical to our 3-fold example, differing only by a central node at the origin. As such, it does not offer a significant simplification. Moreover, our choice to use the k-fold example was deliberate, as it allows us to directly build upon and continue the discussions from previous works [A] and [B]. For these reasons, we prefer not to add a simpler example.
> >
> > However, we recognize that readers might find it confusing to encounter the visualizations in Section 4.1 before reaching the main experimental section. Acknowledging this, and in line with the suggestion, we have now added a brief description of these visualization experiments directly in Section 4.1, immediately after introducing the three types of degeneration.

---

> > > ### Author Response · Authors · 2025-11-22
> > >
> > > > **R4 (Q4).** Comments on connection with Nyquist–Shannon theorem.
> > > > - **Q4** ("The phenomenon of symmetry increase seems loosely analogous to aliasing in the Nyquist–Shannon theorem, where insufficient sampling along the translation group causes distinct frequencies to become indistinguishable. In the present context, a restricted feature space (e.g., limited degrees of irreducible representations) similarly prevents the network from distinguishing between different orientations, leading to a kind of symmetry aliasing over the rotation group. Could the authors comment on whether this analogy is meaningful, and whether the symmetry infimum can be seen as a group-theoretic counterpart to the Nyquist limit for translational symmetry?")
> > >
> > > Thank you for your response. This is an interesting observation. The connection between the Fourier transform and group theory. Aliasing stems from the non-injective nature of time-domain sampling and frequency-domain truncation. The different frequency components obtained from the Fourier transform of a sampled signal correspond to different irreducible representations of the translation group, determined by the signal's frequency. Components of different frequencies as signals with different symmetries and suggest that the sampling frequency must be high enough for the frequency domain to contain irreducible representations that capture the symmetries of these components. This perspective may suggest a natural connection between the Nyquist frequency and the symmetry infimum.
> > >
> > > **Non-equivariance of Low-frequency Sampling:**
> > >
> > > While this observation seems plausible, the connection is subtle. It is important to note that aliasing is a non-injective phenomenon resulting from sampling and Fourier transformation. In the frequency domain, the time-domain sampling operation is equivalent to periodic extension, and aliasing manifests as the overlap of spectra during this extension. However, when aliasing occurs, this process is not equivariant. When the signal's spectrum overlaps during extension, a translation in the time domain no longer corresponds to a simple phase shift in the aliased regions due to phase cancellation. Therefore, for a signal with high-frequency components, low-frequency sampling is a non-equivariant operation. Equivariance is only guaranteed when the sampling rate exceeds the Nyquist frequency. In summary, **due to lack of equivariance in low frequency sampling, symmetry increase is not the cause of aliasing during low-frequency sampling**, though it does tangentially reflect the fact that frequency information can be preserved with high-frequency sampling.
> > >
> > > **Symmetry increases in Signal Processing:**
> > >
> > > In the field of signal processing, a phenomenon that can be perfectly explained by symmetry increase is the application of a high-frequency truncation filter after a Fourier transform, as this truncation is an equivariant linear map. In literature related to protein crystallography and diffraction [C], this type of symmetry that appears at low resolution but vanishes at high resolution is referred to as pseudosymmetry.
> > >
> > > > **Reference**
> > >
> > > - [A] Joshi, C K et al. On the expressive power of geometric graph neural networks. ICML'23.
> > > - [B] Cen J, Li A, Lin N, et al. Are high-degree representations really unnecessary in equivariant graph neural networks? NeurIPS'24.
> > > - [C] Kleywegt G J. Validation of protein crystal structures.

---

> > > > ### Author Response · Authors · 2025-11-26
> > > > **Kindly request your feedback before the end of the discussion period**
> > > >
> > > > Dear Reviewer cFwq,
> > > >
> > > > As the author–reviewer discussion period is approaching its end, we would greatly appreciate it if you could kindly take a moment to review our responses at your convenience. If there are any remaining questions or points that would benefit from further clarification, we would be more than happy to address them before the discussion window closes.
> > > >
> > > > Thank you very much for your time and valuable efforts.
> > > >
> > > > Sincerely,
> > > >
> > > > The Authors

---

> ### Author Response · Authors · 2025-11-29
>
> Dear Reviewer cFwq,
>
> We sincerely appreciate your updated review on November 27, 2025, at 04:20 (EST). We noticed with gratitude that you revised the Contribution score from "2: fair" to "3: good" and the Overall Rating from "4: weak reject" to "6: weak accept."
>
> We apologize that this note comes later than intended. We had been waiting for the opportunity to respond after you completed any revisions or comments on the initial review. Unfortunately, due to an unexpected incident beyond the control of either party, further direct exchanges were no longer possible. We therefore take this opportunity to express our sincere thanks, albeit belatedly.
>
> We truly appreciate the time and effort you devoted to reviewing our paper and reconsidering it. Due to the temporary policy introduced after the recent incident at ICLR, we are no longer able to continue the discussion in this review cycle. We believe that our rebuttal has fully clarified all previous misunderstandings and has clearly conveyed the theoretical depth and practical guidance of our work.
>
> Best regards,
>
> The Authors

---

### Official Review · Reviewer_Xv3y · 2025-11-05

**Soundness:** 3
**Presentation:** 3
**Contribution:** 3
**Rating:** 8
**Confidence:** 4

**Summary:**

The paper studies symmetry increase in equivariant neural networks. When symmetric inputs are passed through a G-equivariant map, they can acquire more symmetry in their outputs than what the input possesses, which should degrade expressivity. There has been some work on the issue but a unified formulation and treatment has been lacking thus far. The paper proposes (a) a symmetry infimum determined by the feature space, (b) a computable procedure via orbit types to predict that infimum, and (c) under manifold-type regularity and sufficient approximation capacity, it is shown that most equivariant maps achieve this infimum (i.e. can take care of unwanted increases).

The paper motivates the problem with degradation with k-fold symmetric structures: rotated copies can collapse in feature space despite not being equivalent physically. Several empirical works (Joshi et al. 2023) and some theory works (Cen et al. 2024) are cited, and then the problem is framed via orbit types and Curie's principle (which in this case would amount to saying that equivariant maps can not decrease symmetry). To summarize the main ideas and their development. If the input orbit type $G_X$ does not occur in $Y$ (not in $O_{G(Y)}$, then the equality $G_X = G_{f(x)}$ is impossible. This means that strict increase (degeneration) is forced. These are illustrated by three flavours (full, axial, half), qualitatively shown in figure 2. Then a partial order on orbit types is defined by subgroup inclusion (a standard move), then in a fixed point space $X^H$ the minimal orbit type exists is unique (this is stated in theorem 3.1). The symmetry infimum $I_{G(X,H)}$ is that minimal orbit type. The symmetry increase is said to be "unexpected" if $G_{f(x)}$ exceeds the infimum. Isovariance is $G_X = G_{f(x)}$, and the necessary condition for it is $O_{G(X)} \subseteq O_{G(Y)}$. Equivalently, $I_{G(Y,H) = (H)} $ for all $(H)$ in $O_{G(X)}$. If the kernel rho_Y is not equal to the identity, every isotropy in $Y$ contains the kernel, so some increase is forced. Then a projection operator $p_{Y(H)}$ (closure under the kernel) is introduced and the relative isovariance defined. The necessary condition becomes $I_{G(Y,H)} = (p_Y(H))$. Using decomposition of representations (and for high multiplicity using Michel's criterion), a union or min of infima type behaviour is obtained for direct sums (corollary 4.3). Later the details are provided for k-fold geometry (D_kh symmetry), the minimal supergroups among candidates identifies and summarized in table 1. The analysis then discusses three degenerations that matches the visualization figures. Other results at the end work out the " most equivariant maps achieve this infimum" bit.

**Strengths:**

I think the core contribution is original and provides a crisp, representation-aware lower bound that predicts degeneration modes and unifies disparate observations (collapse to zero is seen as a special full case).

Main theorems look good and provide useful machinery and results, which formalize quite a few intuitions that had been floating around in the literature.

The paper is also generally clear, and fairly easy to follow. The k-fold example and three degeneration regimes are intuitive.

The main message of the paper can also lead to actionable design guidance: choose feature spaces whose orbit-type sets include the required (p_Y(H)); increase multiplicity to guarantee realizability; expect almost-isovariance generically.

**Weaknesses:**

There is a heavy focus on SO(3)/O(3) and dihedral cases, it would be good to add a discussion for other groups (like SE(3), E(2)), products groups.

The synthetic experiments are fine and align with the theory, but see some suggestions in 'questions.' Despite the theoretical nature of the paper, I think it would be strengthened if the authors include some more experimental setups.

**Questions:**

There seem to be a lot of typos in the paper that should be correct. For example, there are several just in the abstract ("enpowerred", "scienctific", "indepth").

Can the authors add pseudo-code for the orbit-type test and infimum selection in the main text, and a small worked example? This should make the theory easy to follow.

It would be good to have ablations for r, for different architecture families and depth. It would also be good to include one or two real, even if simple, domain-based sanity checks (perhaps from molecular or crystal contexts).

---

> ### Author Response · Authors · 2025-11-22
>
> > **R1 (W1).** Discussion for more cases (e.g. SE(3) / SE(2) and product groups).
> > - **W1** ("There is a heavy focus on SO(3)/O(3) and dihedral cases, it would be good to add a discussion for other groups (like SE(3), E(2)), products groups.")
>
> Thank you for the suggestions on our paper; they are instrumental in making our work more complete. Our theory is primarily discussed within the framework of compact Lie groups as the system symmetry group. For analysis, we mainly focus on SO(3)/O(3) as the **system symmetry group** $G$, with a core example being the case where the **input symmetry group** $H$ is one of the dihedral groups $D_{kh}$ (k-fold geometric symmetry groups) introduced in **Ex. 2.2**. The discussion of SO(3)/O(3) as a system symmetry group is sufficient to address many real-world problems, but the discussion of input symmetries in practical applications has been insufficient. Before we elaborate on the additional content related to input symmetry groups, we will first respond to your suggestions and then discuss the new examples we have added.
>
> **Regarding SE(3)/E(2):**
>
> Our fundamental assumption in this paper is the compact Lie group assumption, which includes orthogonal groups and finite groups. The **system symmetry group** $G$ we consider is a compact Lie group, and the **input symmetry group** $H$, being a closed subgroup of $G$, is therefore also a compact Lie group.
>
> The groups mentioned, SE(3) and E(2), are non-compact because they possess non-compact translation subgroups, placing them outside the scope of our discussion. This compactness assumption is crucial for our theoretical proofs. For example, it guarantees that any finite-dimensional representation space can be uniquely decomposed into a direct sum of irreducible representations (up to equivariant linear isomorphism).
>
> Furthermore, in the field of equivariant graph learning, when referring to SE(3) and E(2) equivariant features, one is typically considering features that are invariant to translation. This approach is similar to our discussion in **Ex. 2.1** on the equivariant encoding of point clouds, where the problem is effectively addressed by considering the input space quotiented by the translation group, thereby reducing the analysis to its rotational components, SO(3) and O(2).
>
> **Regarding product groups:**
>
> We divide the discussion into two cases: one where the system symmetry group $G$ is a direct product group, and the other where the input symmetry group $H$ is a direct product group.
>
>
> For cases where the **system symmetry group** $G$ is a direct product, in fact, we have discussed the handling of product groups as system symmetry groups, that is $O(3) \times S_n$ in **Ex. 2.1**. However, because our focus is on graph-level features, we must select $S_n$-invariant features. As a result, the action of $S_n$ on the feature space becomes trivial, and the problem can be reduced to one with O(3) as the symmetry group. The method for handling the kernel of a representation on the feature space is precisely the topic of **Sec. 3.2**.
>
> For cases where the **input symmetry group** $H$ is a direct product, obtaining practical conclusions requires certain assumptions about the structure of the system symmetry group and the representation. For instance, if we consider a system group $G_1 \times G_2$ acting on a direct sum $V_1 \oplus V_2$, for an input symmetry $H_1 \times H_2$, we can conclude that $H_1 \times H_2$ is an orbit type of $V_1 \oplus V_2$ if and only if $H_1$ is an orbit type of $V_1$ and $H_2$ is an orbit type of $V_2$. That is
> $$\mathcal{O}\_{G\_1 \times G\_2}(V\_1 \oplus V\_2) = \mathcal{O}\_{G\_1}(V\_1) \times \mathcal{O}\_{G\_2}(V\_2)$$
> therefore
> $$I\_{G\_1 \times G\_2}(V\_1 \oplus V\_2, H\_1 \times H\_2) = I\_{G\_1}(V\_1, H\_1) \times I\_{G\_2}(V\_2, H\_2)$$.
>
> **More cases of point groups:**
>
> In many scientific tasks, such as molecular property prediction on the QM9 dataset, **system symmetry group** $G$ is typically SO(3)/O(3). Since the QM9 dataset consists of small molecules, **input symmetry groups** $H$ described by a large number of the closed subgroups of SO(3)/O(3) appear, with the dihedral  groups $D_{kh}$ being a minority in this context (see **Tab. S1**). This necessitates a discussion of the other closed subgroups of SO(3)/O(3). We have added to the paper **the computed results for the symmetry infima of all closed subgroups of SO(3) and O(3)** under the representations $V_{l = l_0}^{\oplus r}$ and $V_{l = l_0^{\pm}}^{\oplus r}$ for $r > 3$. We list the calculation results related to the axial subgroups of O(3) in **Tab. S5** (where we use $K_h$ to represent O(3)). The complete results can be found in **Appendix F.3** and **Appendix F.4**. For calculation details, please see **Appendix C.5** and our response to **R4**.

---

> > ### Author Response · Authors · 2025-11-22
> >
> > **Table S5: Symmetry infimum of $V_{l=l_0^{\pm}}^{\oplus r}, r > 3$ for axial subrgoups of O(3).**
> > |Subgroup||$l_0<k$|$l_0<k$|$l_0<k$|$l_0<k$|$k\leq l_0<2k$|$k\leq l_0<2k$|$k\leq l_0<2k$|$k\leq l_0<2k$|$l_0\geq 2k$|$l_0\geq2k$|$l_0\geq 2k$|$l_0\geq 2k$|
> > |-|-|-|-|-|-|-|-|-|-|-|-|-|-|
> > |||$l_0^-$even|$l_0^-$odd|$l_0^+$even|$l_0^+$odd|$l_0^-$even|$l_0^-$odd|$l_0^+$even|$l_0^+$odd|$l_0^-$even|$l_0^-$odd|$l_0^+$even|$l_0^+$odd|
> > |$C_k$|$k$even|$D_{\infty }$|$C_{\infty v}$|$D_{\infty h}$|$C_{\infty h}$|$C_k$|$C_k$|$C_{kh}$|$C_{kh}$|$C_k$|$C_k$|$C_{kh}$|$C_{kh}$|
> > |$C_k$|$k$odd|$D_{\infty }$|$C_{\infty v}$|$D_{\infty h}$|$C_{\infty h}$|$C_k$|$C_k$|$S_{2k}$|$S_{2k}$|$C_k$|$C_k$|$S_{2k}$|$S_{2k}$|
> > |$S_{2k}$|$k$even|$K_h$|$K_h$|$D_{\infty h}$|$C_{\infty h}$|$S_{2k}$|$S_{2k}$|$D_{\infty h}$|$C_{\infty h}$|$S_{2k}$|$S_{2k}$|$C_{2kh}$|$C_{2kh}$|
> > |$S_{2k}$|$k$odd|$K_h$|$K_h$|$D_{\infty h}$|$C_{\infty h}$|$K_h$|$K_h$|$S_{2k}$|$S_{2k}$|$K_h$|$K_h$|$S_{2k}$|$S_{2k}$|
> > |$C_{kh}$|$k$even|$K_h$|$K_h$|$D_{\infty h}$|$C_{\infty h}$|$K_h$|$K_h$|$C_{kh}$|$C_{kh}$|$K_h$|$K_h$|$C_{kh}$|$C_{kh}$|
> > |$C_{kh}$|$k$odd|$K_h$|$K_h$|$D_{\infty h}$|$C_{\infty h}$|$C_{kh}$|$C_{kh}$|$D_{\infty h}$|$C_{\infty h}$|$C_{kh}$|$C_{kh}$|$C_{2kh}$|$C_{2kh}$|
> > |$C_{kv}(k>2)$|$k$even|$K_h$|$C_{\infty v}$|$D_{\infty h}$|$K_h$|$D_{kd}$|$C_{kv}$|$D_{kh}$|$D_{kh}$|$C_{kv}$|$C_{kv}$|$D_{kh}$|$D_{kh}$|
> > |$C_{kv}(k>2)$|$k$odd|$K_h$|$C_{\infty v}$|$D_{\infty h}$|$K_h$|$D_{kh}$|$C_{kv}$|$D_{kd}$|$D_{kd}$|$C_{kv}$|$C_{kv}$|$D_{kd}$|$D_{kd}$|
> > |$D_{k}(k>2)$|$k$even|$D_{\infty }$|$K_h$|$D_{\infty h}$|$K_h$|$D_k$|$D_{kd}$|$D_{kh}$|$D_{kh}$|$D_k$|$D_k$|$D_{kh}$|$D_{kh}$|
> > |$D_{k}(k>2)$|$k$odd|$D_{\infty }$|$K_h$|$D_{\infty h}$|$K_h$|$D_k$|$D_{kh}$|$D_{kd}$|$D_{kd}$|$D_k$|$D_k$|$D_{kd}$|$D_{kd}$|
> > |$D_{kh}(k>2)$|$k$even|$K_h$|$K_h$|$D_{\infty h}$|$K_h$|$K_h$|$K_h$|$D_{kh}$|$D_{kh}$|$K_h$|$K_h$|$D_{kh}$|$D_{kh}$|
> > |$D_{kh}(k>2)$|$k$odd|$K_h$|$K_h$|$D_{\infty h}$|$K_h$|$D_{kh}$|$D_{kh}$|$D_{\infty h}$|$K_h$|$D_{kh}$|$D_{kh}$|$D_{2kh}$|$D_{2kh}$|
> > |$D_{kd}(k>2)$|$k$even|$K_h$|$K_h$|$D_{\infty h}$|$K_h$|$D_{kd}$|$D_{kd}$|$D_{\infty h}$|$K_h$|$K_h$|$D_{kd}$|$D_{2kh}$|$D_{2kh}$|
> > |$D_{kd}(k>2)$|$k$odd|$K_h$|$K_h$|$D_{\infty h}$|$K_h$|$K_h$|$K_h$|$D_{kd}$|$D_{kd}$|$K_h$|$K_h$|$D_{kd}$|$D_{kd}$|
> >
> > > **R2 (W2 & Q3).** Including real checks and experimental setups.
> > > - **W2** ("I think it would be strengthened if the authors include some more experimental setups.")
> > > - **Q3** ("It would also be good to include one or two real, even if simple, domain-based sanity checks")
> >
> > We sincerely appreciate the thoughtful comments. As outlined in general response **G1**, we have added a supplementary QM9 experiment to further illustrate the practical implications of our theoretical framework. Briefly, we have evaluated isotropic polarizability on QM9, which contains many highly symmetric molecules across 22 point groups (**Tab. S1**). We have pretrained an equivariant encoder (degrees 0–11) and fine-tuned masked MLP heads that selectively activate individual degrees, allowing us to directly examine how symmetry-induced degeneracies affect prediction accuracy. The observed trends (**Tabs. S2, S3**; **Figs. 6, 7**) align closely with our theoretical predictions.
> >
> > In the **Case Study** of **G1**, we have provided molecular examples exhibiting $C_{2h}$, $D_{3h}$, and $T_d$  symmetries. The experimental results demonstrate that following the feature-selection principle introduced at the beginning of **G1** leads to improved model performance on symmetric datasets.
> >
> > > **R3 (Q1).** Correction of typos.
> > > - **Q1** ("There seem to be a lot of typos in the paper that should be correct.")
> >
> > Thank you for your careful reading. We have thoroughly checked the paper and corrected the typos.

---

> > > ### Author Response · Authors · 2025-11-22
> > >
> > > > **R4 (Q2).** Providing pseudo-code for the orbit-type test and infimum and selecting a small worked example
> > > > - **Q2** ("Can the authors add pseudo-code for the orbit-type test and infimum selection in the main text, and a small worked example?")
> > >
> > > Thank you for your suggestions. The previous version overlooked a detailed description of the algorithm's workflow, and the advice will certainly help readers become more familiar with our process. We have added pseudocode and a simple example to the main text, as well as an improved algorithm in the appendix for the complete calculation involving all closed subgroups.
> > >
> > > **Pseudocode and Example:**
> > >
> > > In **Sec. 4.1**, **Prop. 4.2** of the paper, we have added pseudocode for the orbit type test and the symmetry infimum calculation, designated as **Algo. 1** and **Algo. 2**, respectively. To better acquaint readers with the algorithmic flow, we have revised the original **Ex. 4.3** to serve as a demonstration for both algorithms. Due to space constraints in the main text, we provide only a high-level overview of the process; the specific details are elaborated in **Appendix C.3**.
> > >
> > > **Fast Algorithm:**
> > >
> > > The procedure for calculating the symmetry infima for all closed subgroups of SO(3)/O(3) is similar to that in **Ex. 4.3**.
> > >
> > > In Appendix C.4, we first use **Algo. 1** to compute all orbit types for the representations $V_{l = l_0}^{\oplus r}$ and $V_{l = l_0^{\pm}}^{\oplus r}$ where $r > 3$. We note that according to **Prop. 6.2** from [A], the correction term in the Ihrig-Golubitsky criterion is non-zero only for $C_k$ in the case of SO(3), and for $C_k, S_{2k}, C_{kh}$ in the case of O(3). Consequently, for all other cases, the results from the Michel criterion are identical to those for $r = 1$. Therefore, we only performed computations for the cases where the correction term is non-zero, and for all other results, we refer to the $r = 1$ results from **Tab. B.1** and **Tab. B.2** in [B].
> > >
> > > In **Appendix C.5**, we use **Algo. 3**, an improved version of **Algo. 2**, to compute the symmetry infima. Compared to **Algo. 2**, **Algo. 3** can leverage the pre-computed symmetry infima of supergroups to reduce the computational load. We adopted a top-down computational strategy, first calculating for the groups of higher order and then using these computed infima to determine the results for their subgroups.

---

> > > > ### Author Response · Authors · 2025-11-22
> > > >
> > > > > **R5 (Q3).** Ablations for r, for different model architecture families and depth
> > > > > - **Q3** ("It would be good to have ablations for r, for different architecture families and depth.")
> > > >
> > > > Thank you for your suggestion. Our theory is primarily based on high-multiplicity representations, making an ablation study on multiplicity essential. Furthermore, to demonstrate the applicability of our theory to real-world models, it is necessary to conduct ablations on different model architectures and network depths.
> > > >
> > > > **Adjustment of Experimental Setup:**
> > > >
> > > > To demonstrate the indistinguishability of features, the approach in [C] involved feeding the features into a classifier. However, we have noted that this method is not fundamental, as the results can be influenced by numerical errors during training and the adequacy of the classifier's training. Therefore, we have adjusted our experimental setup to directly calculate the distance between the embedded features. The original experimental results have been moved to **Appendix G.2**. We conducted tests across two model architectures (TFN and HEGNN), three multiplicities for the irreducible representations (1, 4, and 16), and four network depths (1, 2, 3, and 4 layers). The results are presented in **Tab. S6**.
> > > >
> > > > **Experimental Results:**
> > > >
> > > > The outcomes were consistent across all experimental settings. For features exhibiting full degeneracy, the distance between the features before and after a 3D rotation was less than $10^{-6}$. For features with axial degeneracy, the distance between features before and after a 2D rotation was also less than $10^{-6}$. These conclusions align with the results we calculated in **Ex. 4.3**.
> > > >
> > > > **Table S6:  Max embeddding differenece norm among all 24 configurations.**
> > > >
> > > > |**GNN Layer**|**2-fold (2D)**|**3-fold (2D)**|**4-fold (2D)**|**6-fold (2D)**|**2-fold (3D)**|**3-fold (3D)**|**4-fold (3D)**|**6-fold (3D)**|
> > > > |-|-|-|-|-|-|-|-|-|
> > > > |$l=0$|>1e-3|>1e-3|>1e-3|>1e-3|>1e-3|>1e-3|>1e-3|>1e-3|
> > > > |$l=1$|>1e-3|>1e-3|>1e-3|>1e-3|>1e-3|>1e-3|>1e-3|>1e-3|
> > > > |$l=2$|**<1e-6**|>1e-3|>1e-3|>1e-3|**<1e-6**|**<1e-6**|**<1e-6**|**<1e-6**|
> > > > |$l=3$|>1e-3|**<1e-6**|>1e-3|>1e-3|>1e-3|**<1e-6**|>1e-3|>1e-3|
> > > > |$l=4$|**<1e-6**|>1e-3|**<1e-6**|>1e-3|**<1e-6**|**<1e-6**|**<1e-6**|**<1e-6**|
> > > > |$l=5$|>1e-3|**<1e-6**|>1e-3|>1e-3|>1e-3|**<1e-6**|>1e-3|>1e-3|
> > > > |$l=6$|**<1e-6**|**<1e-6**|**<1e-6**|**<1e-6**|**<1e-6**|**<1e-6**|**<1e-6**|**<1e-6**|
> > > > |$l=7$|>1e-3|**<1e-6**|>1e-3|>1e-3|>1e-3|**<1e-6**|>1e-3|>1e-3|
> > > > |$l=8$|**<1e-6**|**<1e-6**|**<1e-6**|**<1e-6**|**<1e-6**|**<1e-6**|**<1e-6**|**<1e-6**|
> > > > |$l=9$|>1e-3|**<1e-6**|>1e-3|>1e-3|>1e-3|**<1e-6**|>1e-3|>1e-3|
> > > > |$l=10$|**<1e-6**|**<1e-6**|**<1e-6**|**<1e-6**|**<1e-6**|**<1e-6**|**<1e-6**|**<1e-6**|
> > > > |$l=11$|>1e-3|**<1e-6**|>1e-3|>1e-3|>1e-3|**<1e-6**|>1e-3|>1e-3|
> > > >
> > > > > **Reference**
> > > >
> > > > - [A] Ihrig E, Golubitsky M. Pattern selection with O(3) symmetry.
> > > > - [B] Linehan M J, Stedman G E. Little groups of irreps of O(3), SO(3), and the infinite axial subgroups.
> > > > - [C] Joshi, C K et al. On the expressive power of geometric graph neural networks. ICML'23.

---

> > > > > ### Author Response · Authors · 2025-11-26
> > > > > **Kindly request your feedback before the end of the discussion period**
> > > > >
> > > > > Dear Reviewer Xv3y,
> > > > >
> > > > > As the author–reviewer discussion period is approaching its end, we would greatly appreciate it if you could kindly take a moment to review our responses at your convenience. If there are any remaining questions or points that would benefit from further clarification, we would be more than happy to address them before the discussion window closes.
> > > > >
> > > > > Thank you very much for your time and valuable efforts.
> > > > >
> > > > > Sincerely,
> > > > >
> > > > > The Authors

---

> > > > > > ### Comment · Reviewer_Xv3y · 2025-11-27
> > > > > >
> > > > > > Thank you for your very detailed response -- much appreciated. They have clarified several of my questions. I will keep my current rating (which is already at a clear accept).

---

> ### Author Response · Authors · 2025-11-27
>
> Dear Reviewer Xv3y,
>
> Thank you very much for your thoughtful and timely response. We sincerely appreciate the constructive suggestions you have provided throughout the discussion. Your feedback has significantly improved the quality and clarity of our manuscript. Your support is truly encouraging to us, and we are grateful for the time and effort you devoted to reviewing our work.
>
> Best regards,
>
> The Authors

---

### Author Response · Authors · 2025-11-22
**General Response**

We sincerely thank the reviewers and ACs for their time and thoughtful evaluations. We are encouraged that the proposed symmetry infimum is regarded as clear, important, and valuable (Reviewers Xv3y & gQcX), and that the theoretical framework is viewed as original and rigorous (Reviewers Xv3y & cFwq).

We also appreciate constructive feedback across all reviews. In response to these concerns, we have added additional experiments and analyses, highlighted these updates in the revised PDF, and uploaded code and model weights to the supplementary material anonymous GitHub repository.

> **G1: Molecule property prediction with pretrained equivariant features (in Sec. 6.3).**
> - **Reviewer Xv3y: W2** ("some more experimental setups") & **Q3** ("domain-based sanity checks (perhaps from molecular or crystal contexts)")
> - **Reviewer cFwq: Q3** ("foresee domains (e.g., molecular modeling, materials science) where symmetry increase significantly impacts performance and where their theory could guide practical model improvements?")
> - **Reviewer gQcX: Q1** ("the practical feasibility of using higher-dimensional irreducible representations in the output representation, or of increasing their multiplicities")

We have added an additional experiment to provide a more **domain-grounded validation** of our theoretical framework. Specifically, we have included an evaluation on the QM9 dataset to **examine how features of different degrees contribute to a practical molecular property prediction task**. The experiment shows strong dependence between model performance and feature choice on a real-world dataset, validating our discussion on symmetry increase and feature space dimension. It demonstrates a critical model design principle: **not only should one avoid building a model entirely from fully degenerate features, but one should also avoid including individual feature components that undergo full degeneration, as they can be actively detrimental to predictive performance**.

**Dataset:**

We have used the `PointGroup` library to compute point groups of all QM9 molecules, revealing **22 distinct symmetry classes** as shown in **Tab. S1**. This rich symmetry structure makes QM9 an appropriate setting for assessing  degeneration in real data.

**Table S1:  Point group and number of corresponding molecules.**

|**$C_1$**|**$C_s$**|**$C_2$**|**$C_{2v}$**|**$C_{3v}$**|**$C_{2h}$**|**$C_i$**|**$D_{3h}$**|
|-|-|-|-|-|-|-|-|
|111,935|17,794|458|433|67|48|15|15|
|**$D_{2d}$**|**$C_3$**|**$D_{\infty h}$**|**$C_{\infty v}$**|**$D_{3d}$**|**$D_{2h}$**|**$T_d$**|**$C_{3h}$**|
|11|11|8|8|6|6|5|4|
|**$D_{3}$**|**$C_4$**|**$D_2$**|**$D_{6h}$**|**$O_h$**|**$S_4$**|||
|4|1|1|1|1|1|||

**Embeddings:**

The purpose of this experiment is to evaluate how embeddings of different degrees contribute to a practical downstream task. We have followed a pretrain–finetune strategy: an equivariant model is first trained as a shared encoder to produce embeddings of all degrees, and feature masks are then applied during finetuning to isolate the degrees we aim to assess. Concretely, we have adopted HEGNN [A] as the encoder. TFN becomes computationally prohibitive here—when $l_0 = 11$, training a four-layer TFN with only four irrep channels demands roughly 10 A100-hours per epoch on QM9. In contrast, HEGNN employs a spherical-scalarization mechanism that reduces the computational cost from $\mathcal{O}(L^6)$ to $\mathcal{O}(L^2)$, making both pretraining and finetuning tractable.

Our pipeline first pretrains an equivariant encoder covering degrees 0–11, after which the encoder is entirely frozen, and lightweight MLP heads are finetuned under different masking configurations. This setup enables us to activate specific degrees on demand and directly test how symmetry-induced degeneracies affect predictive performance.

**Results:**

The prediction results for the top-16 point groups are reported in **Tabs. S2, S3**, and also **Figs. 6** (in **Sec. 6.3**)**, 7** (in **Appendix G.3**) , the empirical behavior aligns closely with our theoretical predictions.  When the symmetry of a molecule forces certain feature degrees to degenerate, the corresponding prediction heads exhibit significantly elevated error, while degrees that remain non-degenerate maintain low error. The transition between degenerate and non-degenerate cases aligns precisely with the symmetry-increase conditions analyzed in our theoretical framework. Together, these results confirm that our theory provides accurate and actionable guidance for designing and understanding equivariant models in realistic molecular property-prediction tasks.

---

> ### Author Response · Authors · 2025-11-22
>
> **Case Study:**
>
> We have further verified our theory using three notable point groups: $C_{2h}$, $D_{3h}$, and $T_d$. The results have been listed in **Tab. S4**. Each of which presents a representative symmetry-increase pattern. According to the calculated results in **Tab. 22 (in Appendix G.3)**, we also report the symmetry lower bounds in **Tab. S4**. For cases where $l_0 > 0$, the symmetries of $C_{2h}$, $C_{3h}$, and $T_d$ increase to $K_h$ (i.e. O(3)) when $l_0$ is odd, $l_0 = 1$, and $l_0 = 1, 2, 5$, respectively. This corresponds to the case of full degeneration. Because the features lose their discriminative power in this scenario, we observe in that the model's predictive ability, when using only $l_0$-degree features, is significantly lower than its performance when using features of other degrees. Furthermore, due to the full degeneration of the $1$-degree features, we observe that after introducing them, for molecules with $C_{2h}$, $C_{3h}$, and $T_d$ symmetry, the predictive performance does not increase but instead decreases. Following the introduction of $2$-degree features, there is a marked improvement in performance for $C_{2h}$ and $C_{3h}$. However, for $T_d$, the performance again fails to increase and instead decreases because the full degeneration of $2$-degree features.
>
> **Table S2:  MAE loss ($a_0^3$) for isotropic polarizability prediction with degree $\color{red}{l=l_0}$.**
>
> |Point Group|$l_0=0$|$l_0=1$|$l_0=2$|$l_0=3$|$l_0=4$|$l_0=5$|$l_0=6$|$l_0=7$|$l_0=8$|$l_0=9$|$l_0=10$|$l_0=11$|
> |-|-|-|-|-|-|-|-|-|-|-|-|-|
> |$C_1$|2.05|3.25|1.95|2.78|2.08|2.78|1.37|2.67|2.08|2.56|1.93|2.57|
> |$C_s$|3.25|4.31|1.93|4.19|2.70|4.19|1.77|4.28|2.80|3.94|2.62|4.12|
> |$C_2$|3.33|7.83|3.13|4.98|3.36|4.76|2.36|4.79|3.74|4.65|3.33|5.06|
> |$C_{2v}$|4.87|7.20|3.04|6.92|3.97|6.50|2.57|6.75|4.27|6.43|4.25|6.44|
> |$C_{3v}$|8.97|9.38|5.85|7.69|6.47|10.25|4.00|8.70|8.92|9.81|7.94|10.41|
> |$C_{2h}$|9.30|26.32|3.23|21.20|4.44|28.04|2.20|25.46|7.97|28.29|6.51|29.03|
> |$C_i$|7.37|19.81|2.48|14.70|3.34|21.66|1.81|18.88|9.00|21.88|7.60|22.65|
> |$D_{3h}$|6.59|28.59|8.95|13.65|9.82|10.09|4.83|15.63|9.11|14.69|11.18|15.85|
> |$C_3$|2.94|8.53|5.58|6.21|5.69|6.86|1.41|6.81|5.45|5.44|3.18|5.30|
> |$D_{2d}$|6.14|22.61|6.67|8.74|6.40|12.37|4.74|10.70|11.55|9.56|9.39|13.14|
> |$C_{\infty v}$|32.86|22.57|12.47|26.99|13.68|22.18|10.42|23.90|10.47|23.28|10.86|34.41|
> |$D_{\infty h}$|32.72|37.24|6.95|35.79|13.25|37.70|6.37|37.00|10.60|37.76|9.59|37.96|
> |$D_{2h}$|6.42|26.34|1.66|20.88|4.33|27.65|2.95|25.98|8.46|28.96|8.97|29.58|
> |$D_{3d}$|19.20|34.94|14.73|31.26|23.61|36.25|8.83|33.33|33.61|36.38|23.30|37.08|
> |$T_d$|17.07|44.73|42.07|18.12|18.05|46.58|8.14|13.30|34.87|12.70|23.57|25.97|
> |$C_{3h}$|5.11|15.21|1.52|6.02|5.92|2.77|3.86|8.89|6.57|4.61|9.35|14.28|
>
>
> **Table S3:  MAE loss ($a_0^3$) for isotropic polarizability prediction with degree $\color{red}{l\leq l_0}$.**
>
> |Point Group|$l_0=0$|$l_0=1$|$l_0=2$|$l_0=3$|$l_0=4$|$l_0=5$|$l_0=6$|$l_0=7$|$l_0=8$|$l_0=9$|$l_0=10$|$l_0=11$|
> |-|-|-|-|-|-|-|-|-|-|-|-|-|
> |$C_1$|1.99|1.18|0.46|0.33|0.28|0.24|0.19|0.18|0.16|0.15|0.14|0.13|
> |$C_s$|3.13|1.99|0.69|0.50|0.42|0.36|0.28|0.25|0.24|0.22|0.21|0.19|
> |$C_2$|3.12|2.33|0.93|0.69|0.57|0.52|0.38|0.34|0.32|0.31|0.28|0.27|
> |$C_{2v}$|4.48|3.03|1.09|0.85|0.75|0.65|0.50|0.45|0.43|0.39|0.36|0.34|
> |$C_{3v}$|8.15|5.32|2.41|0.91|0.85|0.71|0.47|0.51|0.51|0.49|0.55|0.59|
> |$C_{2h}$|9.17|10.56|2.28|2.09|1.07|1.10|0.74|0.69|0.49|0.47|0.54|0.47|
> |$C_i$|7.34|9.25|1.20|1.33|1.29|0.76|0.71|0.43|0.69|0.35|0.29|0.25|
> |$D_{3h}$|5.35|6.72|2.12|2.11|2.05|0.96|0.86|0.79|0.47|0.26|0.39|0.48|
> |$C_3$|2.74|1.50|1.06|0.94|0.59|0.59|0.45|0.40|0.21|0.37|0.34|0.24|
> |$D_{2d}$|6.18|3.69|2.17|1.55|1.21|1.70|2.02|0.75|1.38|0.87|1.14|0.85|
> |$C_{\infty v}$|31.86|23.85|9.34|6.62|6.13|5.04|4.25|4.78|3.95|4.84|5.59|4.27|
> |$D_{\infty h}$|32.01|30.99|6.04|5.13|6.35|5.61|4.50|5.95|2.77|5.69|6.43|4.09|
> |$D_{2h}$|5.30|8.11|0.94|1.23|0.88|0.63|0.49|0.33|0.22|0.57|0.31|0.25|
> |$D_{3d}$|16.96|17.24|10.92|10.16|4.93|4.73|1.10|2.44|0.70|1.05|1.05|0.70|
> |$T_d$|15.36|18.06|18.22|10.66|7.53|8.06|1.57|1.63|1.75|1.27|1.09|1.23|
> |$C_{3h}$|4.76|5.55|0.93|0.93|1.06|0.45|0.29|0.36|0.30|0.30|0.39|0.31|

---

> > ### Author Response · Authors · 2025-11-22
> >
> > **Table S4:  Special point group cases for analysis.**
> >
> > |**Point Group**|**$l=0$**|**$l=1$**|**$l=2$**|**$l=3$**|**$l=4$**|**$l=5$**|**$l=6$**|**$l=7$**|**$l=8$**|**$l=9$**|**$l=10$**|**$l=11$**|
> > |-|-|-|-|-|-|-|-|-|-|-|-|-|
> > |$C_{2h}$|9.30|**26.32**|3.23|**21.20**|4.44|**28.04**|2.20|**25.46**|7.97|**28.29**|6.51|**29.03**|
> > |$D_{3h}$|6.59|**28.59**|8.95|13.65|9.82|10.09|4.83|15.63|9.11|14.69|11.18|15.85|
> > |$T_d$|17.07|**44.73**|**42.07**|18.12|18.05|**46.58**|8.14|13.30|34.87|12.70|23.57|25.97|
> > |**PointGroup**|**$l\leq0$**|**$l\leq1$**|**$l\leq2$**|**$l\leq3$**|**$l\leq4$**|**$l\leq5$**|**$l\leq6$**|**$l\leq7$**|**$l\leq8$**|**$l\leq9$**|**$l\leq10$**|**$l\leq11$**|
> > |$C_{2h}$|9.17|**10.56**|2.28|**2.09**|1.07|**1.10**|0.74|**0.69**|0.49|**0.47**|0.54|**0.47**|
> > |$D_{3h}$|5.35|**6.72**|2.12|2.11|2.05|0.96|0.86|0.79|0.47|0.26|0.39|0.48|
> > |$T_d$|15.36|**18.06**|**18.22**|10.66|7.53|**8.06**|1.57|1.63|1.75|1.27|1.09|1.23|
> > |**PointGroup**|**$l=0$**|**$l=1$**|**$l=2$**|**$l=3$**|**$l=4$**|**$l=5$**|**$l=6$**|**$l=7$**|**$l=8$**|**$l=9$**|**$l=10$**|**$l=11$**|
> > |$C_{2h}$|$K_h$|$\color{red}{K_h}$|$C_{2h}$|$\color{red}{K_h}$|$C_{2h}$|$\color{red}{K_h}$|$C_{2h}$|$\color{red}{K_h}$|$C_{2h}$|$\color{red}{K_h}$|$C_{2h}$|$\color{red}{K_h}$|
> > |$D_{3h}$|$K_h$|$\color{red}{K_h}$|$D_{\infty h}$|$D_{3h}$|$D_{\infty h}$|$D_{3h}$|$D_{6h}$|$D_{3h}$|$D_{6h}$|$D_{3h}$|$D_{6h}$|$D_{3h}$|
> > |$T_d$|$K_h$|$\color{red}{K_h}$|$\color{red}{K_h}$|$T_d$|$O_h$|$\color{red}{K_h}$|$O_h$|$T_d$|$O_h$|$T_d$|$O_h$|$T_d$|
> >
> >
> >
> > > **Note 1.** A condition missing in **Cor. 4.3**.
> >
> > We apologize for an omission in the original **Cor. 4.3**. The properties we stated for the orbit types and infimum of direct-sum representations require a key assumption: that all closed subgroups of $G$ satisfy the bottleneck condition. Without this condition, only the weaker inclusions
> > $$\mathcal{O}\_G(V\_{1}) \cup \mathcal{O}\_G(V\_{2}) \subseteq \mathcal{O}\_G(V\_{1} \oplus V\_{2})$$
> > and inequalities
> > $$I\_G(V\_{1} \oplus V\_{2},H) \le I\_{G}(V\_{i},H)$$
> > for $i=1,2$ are generally guaranteed to hold. **However, this omission does not affect the practical application of our theory**. Our primary goal is to mitigate undesirable symmetry increase, and to achieve this, we only need to control the upper bound of the symmetry infimum.
> >
> > To ensure our theory is complete, we have added a detailed discussion of the bottleneck condition in **Appendix C.6**. This new section clarifies that while all closed subgroups of SO(3) satisfy this condition, only a subset of the closed subgroups of O(3) do.
> >
> > > **Note 2.** Removal of discussion related to the Linehan-Stedman criterion.
> >
> > The discussion of the Linehan-Stedman criterion in the original manuscript was presented in an intuitive manner and lacked rigorous mathematical justification. Formalizing this theory would require significant additional effort. Although we attempted to provide a rigorous definition in **Prop. C.3**, some further formalization is still required to clarify certain concepts used in this paper. For the sake of rigor, we have removed the parts of our discussion related to the Linehan-Stedman criterion. This removed content was primarily concerned with determining the orbit types of general representations. However, in the context of machine learning, our primary focus is on the high-multiplicity case. **Therefore, the removal of this section does not affect the application of our theory in the vast majority of relevant scenarios.**
> >
> > > **Reference**
> >
> > - [A] Cen J, Li A, Lin N, et al. Are high-degree representations really unnecessary in equivariant graph neural networks? NeurIPS'24.

---

### Author Response · Authors · 2025-12-03
**Rebuttal Acknowledgement**

Dear AC and Reviewers,

Thank you very much for the time and effort you devoted during the rebuttal period. We sincerely appreciate your recognition of our work. Your thoughtful feedback has been invaluable in improving the quality and clarity of our manuscript.

---

Our work provides a systematic theoretical framework, a computable algorithm, and a validated practical guideline for understanding and mitigating symmetry increases in ENNs:

- **Theory**: We formalize the notion of the **symmetry infimum** of inputs and feature spaces, and prove that the output symmetry of any equivariant map cannot decrease and is always constrained by a unique **symmetry infimum**.

- **Algorithm & Practical Guideline**: We develop a general algorithm to compute the **symmetry infimum** and provide complete results for **all closed subgroups of SO(3)/O(3)**. Based on these results, we propose practical guidelines to prevent harmful symmetry increases in feature design, supported by theoretical guarantees.

- **Experiment**: On both synthetic datasets and the real-world QM9 dataset, feature choices that violate our guideline lead to performance degradation, consistent with our theoretical predictions.

Overall, our work offers principled and directly applicable guidance for ENN design and, to the best of our knowledge, is the first to quantitatively characterize the effect of symmetry increases.

---

During the discussion phase, we provided additional theoretical clarifications and new experimental results to address the reviewers' key concerns. We were greatly encouraged by the **uniformly positive feedback** from all four reviewers. In particular, Reviewer cFwq revised the contribution score from "2: fair" to **"3: good"**, and the overall rating from "4: weak reject" to **"6: weak accept"**. Reviewer Xv3y expressed strong support for our rebuttal and maintained a **clear accept** recommendation. Although further interaction with Reviewer gQcX was restricted by policy, their initial review already provided a **clear accept** recommendation.

Our theoretical framework was described as "original" and "rigorous" (Reviewers Xv3y & cFwq), and as offering a "simple, checkable target" (Reviewer gQcX). The main results were characterized as "useful machinery" (Reviewer Xv3y) and "sound and internally consistent" (Reviewer gQcX). Reviewers also highlighted the clarity and practical value of our work, noting that it is "clear and intuitive" and provides "actionable design guidance" (Reviewer Xv3y), with experiments described as "consistent" and "closely matching the predictions" (Reviewers cFwq & gQcX).

---

Finally, we outline the timeline of this rebuttal process (all times in EST):

- Reviewer Xv3y: assigned a score of 8 ("clear accept") from the outset. After our rebuttal, on Nov 27, 2025 at **00:25**, the reviewer stated that our response had "clarified several of my questions" and maintained the "clear accept" recommendation.
- Reviewer cFwq: initially assigned a score of 4 ("weak reject"). On Nov 27, 2025 at **04:20**, the reviewer revised the contribution score from "2: fair" to "3: good" and the overall rating from "4: weak reject" to "6: weak accept".
- Reviewer gQcX: assigned a score of 8 ("clear accept") from the beginning. Unfortunately, we were unable to continue discussion with this reviewer.

For completeness, we note that the OpenReview reviewer identity leak for ICLR 2026 was officially reported at **10:09** on Nov 27, 2025, which occurred **after all of the updates described above**.

Best regards,

The Authors

---

### Meta-Review · Area_Chair_pq3M · 2026-01-06

**Summary:**

This paper attempts to provide a systematic theoretical treatment of the phenomenon known as symmetry amplification in equivariant neural networks (ENNs), and to clarify both the conditions under which it arises and principled ways to mitigate it. In ENNs, when the input exhibits a high degree of symmetry, the output representation may acquire symmetry beyond that of the input itself, leading to a loss of expressivity in which distinct inputs become indistinguishable. The paper formalizes this phenomenon based on the group-theoretic constraint that equivariant maps cannot reduce symmetry, and introduces the notion of a symmetry lower bound, which is determined by the structure of the feature space.

Reviewer Xv3y gave a highly positive assessment, while raising questions about extensibility to other symmetry groups such as SE(3) and product groups, as well as about expanding the experimental setup and improving the presentation. In response, the authors explained that their theory is formulated under the assumption of compact groups, and added clarification that, in many practical applications, the problem can be reduced to the SO(3) component. Following these clarifications, Reviewer Xv3y stated that their concerns had been resolved and maintained the original “clear accept” recommendation.

Reviewer cFwq also rated the work highly in terms of theoretical rigor and originality, but expressed concern that the paper appeared overly abstract and seemed to focus more on analyzing the phenomenon than on actually mitigating symmetry amplification. The authors responded by reiterating that the entire paper is structured around the goal of suppressing undesirable symmetry increases. In particular, they substantially revised Section 4.2 to articulate concrete design principles for output feature spaces, and added new experiments demonstrating that the theory translates into tangible performance improvements. Although Reviewer cFwq was unable to respond further, it is reasonable to expect that they would appreciate the clearer alignment between theory and empirical results.

Reviewer gQcX evaluated the paper very positively, while pointing out that the presentation might be difficult to follow for readers outside the immediate field and that the scope of the claims should be made explicit given certain technical constraints. The authors addressed these points by outlining plans to improve exposition in the appendix and by explicitly acknowledging the theoretical difficulties involved in extending the results beyond smooth settings, thereby clarifying the scope and identifying directions for future work. These responses were largely accepted, and Reviewer gQcX’s strong positive evaluation remained unchanged.

In summary, the paper was consistently recognized for its theoretical contributions, and the authors made a sincere and effective effort during the rebuttal to address methodological questions and reviewer concerns within reasonable limits. This effort, and the resulting clarifications, merit a fair and positive assessment of the work.

**Reviewer Concerns:**

See above.

**Reviewer Scores:**

See above.

---

### Decision · Program_Chairs · 2026-01-26

Accept (Poster)